



# Trends and variability of snowmelt in China under climate change

Yong Yang[1], Rensheng Chen[1,2], Guohua Liu[1,3], Zhangwen Liu[1], Xiqiang Wang[1]

[1]Qilian Alpine Ecology and Hydrology Research Station, Northwest Institute of Eco-Environment and Resources, Chinese Academy of Sciences, Lanzhou 730000, China

[2]College of Urban and Environmental Sciences,Northwestern University, Xi'an 710127, China
[3]College of Resources and Environment, University of Chinese Academy of Sciences, Beijing 100049, China

*Correspondence to*: Rensheng Chen (crs2008@lzb.ac.cn)

**Abstract.** Snowmelt is a major fresh water resource, and quantifying snowmelt and its variability under climate change is necessary for planning and management of water resources. Spatiotemporal changes in snow properties in China have drawn

wide attention in recent decades; however, country-wide assessments of snowmelt are lacking. Using precipitation and temperature data with a high spatial resolution (0.5 seconds, approximately 1 km), this study calculated the monthly snowmelt in China for the 1951-2017 period using a simple temperature index model, and the model outputs were validated using snowfall, snow depth, snow cover extent and snow water equivalent. Precipitation and temperature scenarios developed from five CMIP5 models were used to predict future snowmelt in China under three different representative

concentration pathways (RCP) scenarios (RCP2.6, RCP4.5 and RCP8.5). The results showed that the mean annual snowmelt in China from 1951 to 2017 was $2.41 \times 10^{11}$ m$^3$. The mean annual snowmelts in Northern Xinjiang, Northeast China, and the Tibetan Plateau – China's three main stable snow cover regions – were $0.18 \times 10^{11}$ m$^3$, $0.42 \times 10^{11}$ m$^3$ and $1.15 \times 10^{11}$ m$^3$, respectively. From 1951 to 2017, the snowmelt increased significantly in the Tibetan Plateau and decreased significantly in North, Central and Southeast China. In the whole of China, there was a decreasing trend in snowmelt, but this was not

statistically significant. The mean annual snowmelt runoff ratios were generally more than 10% in almost all third-level basins in West China, more than 5% in third-level basins in North and Northeast China, and less than 2% in third-level basins in South China. From 1951 to 2017, the annual snowmelt runoff ratios decreased in most third-level basins in China. Under RCP2.6, RCP4.5 and RCP8.5, the projected snowmelt in China in 2030s (2050s, 2090s) may decrease by 13.4% (16.3%, 13.8%), 19.1% (19.8%, 22.5%), 17.1% (24.7%, 42.8%) compared with the historical period (1951-2017),

respectively. Most of the projected mean annual snowmelt runoff ratios in third-level basins in different decades (2030s, 2050s and 2090s) were lower than those in the historical period. Low temperature regions can tolerate more warming, and the snowmelt change in these regions is mainly influenced by precipitation; however, the snowmelt change in warm regions is more sensitive to temperature increases. The spatial variability of snowmelt changes may lead to regional differences in the impact of snowmelt on water supply.



## 1 Introduction

Snow properties have changed significantly under the ongoing warming of the global climate, and variations in snow cover exert strong feedbacks on the climate system due to its high albedo and low thermal conductivity as well as the high latent

heat of phase change (Zhang and Ma, 2018; Pulliainen et al., 2020; Vano, 2020; You et al., 2020). Additionally, snow is a critical component of the hydrological system and water cycle, and snowmelt is a major fresh water resource in many regions (Mankin et al., 2015). More than one-sixth of the earth's population relies on snowmelt for their water supply, and snowmelt-dominated regions contributes roughly one-quarter of the global gross domestic product (Barnett et al., 2005). Climate warming has resulted in smaller snowfall/precipitation ratios, earlier snowmelt times and slower snowmelt rates

(Berghuijs et al., 2014; Musselman et al., 2017; Barnhart et al., 2020). This has not only changed seasonal runoff distributions, but has also affected the total annual runoff (Bloschl et al., 2019; Jenicek and Ledvinka, 2020). Consequently, determining the amount of snowmelt and its variability under climate change is important for the planning and management of water resources, such as agricultural water management, flood forecasting, reservoir operation, and the design of hydraulic structures (Barnhart et al., 2020; Qin et al., 2020).

There are many models for calculating snowmelt, and these can be basically divided into two types: physically based snowmelt models and simpler temperature index models (also known as degree-day methods) (Skaugen et al., 2018; Li et al., 2019). In theory, physically based snowmelt models can provide more accurate predictions by considering the coupled interaction between energy components in complex snowmelt processes (Li et al., 2019). However, many studies have shown that due to the mathematical complexities and massive data requirements of physically based models, they do not

necessarily perform better than temperature index models (Hock, 2003; Jost et al., 2012; Skaugen et al., 2018; Lopez et al., 2020). Temperature index models are based on the assumptions that the temporal variability of incoming solar radiation is adequately represented by the variations of air temperature that the snowmelt during a time interval is proportional to positive air temperatures (Hock, 2003; Jost et al., 2012; Lopez et al., 2020). Because of the wide availability of air temperature data, their computational simplicity, and their generally good model performance, temperature index models are

the most common approach for calculating snowmelt around the world (Hock, 2003; Immerzeel et al., 2010; Lopez et al., 2020).

China covers a vast area and a variety of climate regions , and its snow covered regions are widely distributed geographically with evident spatial differences (Tan et al., 2019). Northern Xinjiang, Northeast China-Inner Mongolia (hereafter referred to as the Northeast China) and the Tibetan Plateau are the three main regions with stable snow cover in China (Ke et al., 2016)

(Fig. 1a). As a typical arid and semi-arid region of Central Asia with a significant lack of freshwater resources, the surface runoff in the Xinjiang Uygur Autonomous Region (hereafter, Xinjiang) is mainly supplied by snow meltwater (Chen et al., 2020; Wu et al., 2021). In Northeast China, snow plays an important role as a natural reservoir in winter and a source of water in spring, with snowmelt contributing more than half of the runoff during the main crop planting months (April and May) (Qi et al., 2020). Snow melting is aslo an important hydrological process in the Tibetan Plateau, which is the source





region of many major Asian rivers and considered as the asian water towers (Immerzeel et al., 2010). Snowmelt is also an important water source in other parts of China, especially the arid and semi-arid areas in North China (Zhang et al., 2015; Wu et al., 2021). Spatiotemporal changes in China's snow properties (e.g. snow cover extent, snow cover phenology, snow depth, snow density and snow water equivalent) have drawn wide attention in recent decades (Dai and Chen, 2010; Wang and Li, 2012; Chen et al., 2016; Ke et al., 2016; Tan et al., 2019; Ma et al., 2020; Yang et al., 2020); however, country-wide

assessments of snowmelt are lacking. Projected increases in air temperature and associated precipitation regime shifts are expected to have significant consequences for snowmelt and water resources (Ficklin et al., 2016). Although many studies have investigated snowmelt in single or multiple basins (e.g. Zhang et al., 2015; Chen et al., 2019; Li et al., 2019; Zhang et al., 2020; Li et al., 2021), the spatiotemporal variability of snowmelt in China and its response to climate warming remain unclear.

Under this background, we developed a simple monthly temperature-index model to calculate the snowmelt in China using a high spatial resolution (0.5 seconds, approximately 1 km) dataset of monthly air temperatures and precipitation. The model considered complex snow processes such as melting, accumulation and sublimation, and was validated using snowfall, snow depth, snow cover extent and snow water equivalent. The objectives of this study were to (1) quantify the snowmelt across China and in its three main stable snow cover distribution regions; (2) analyse the spatiotemporal variability of snowmelt in

China in the 1951-2017 period; (3) elucidate the spatiotemporal variability of snowmelt runoff ratio in third-level basins in China; (4) assess the impacts of projected future climate change on snowmelt in China.

## 2 Study region and data sets

### 2.1 Study region

In general, China can be divided into five main climatic zones: the mountain plateau zone (MPZ), the temperate monsoon

zone (TMZ), the temperate continental zone (TCZ), the subtropical monsoon zone (SMZ) and the tropical monsoon zone (Yang et al., 2021) (Fig. 1b). Because the land area of the tropical monsoon zone is significantly smaller than that of the other climatic zones and it has extremely low snowfall, it was incorporated into the SMZ for parameter assignment. Additionally, because of the lack of meteorological data and the fact that it has very little snowfall, Taiwan was not considered in this study. The snow cover types in China can be divided into five types: prairie, taiga, tundra, mountain and

ephemeral types (Li et al., 2020) (Fig. 1c).



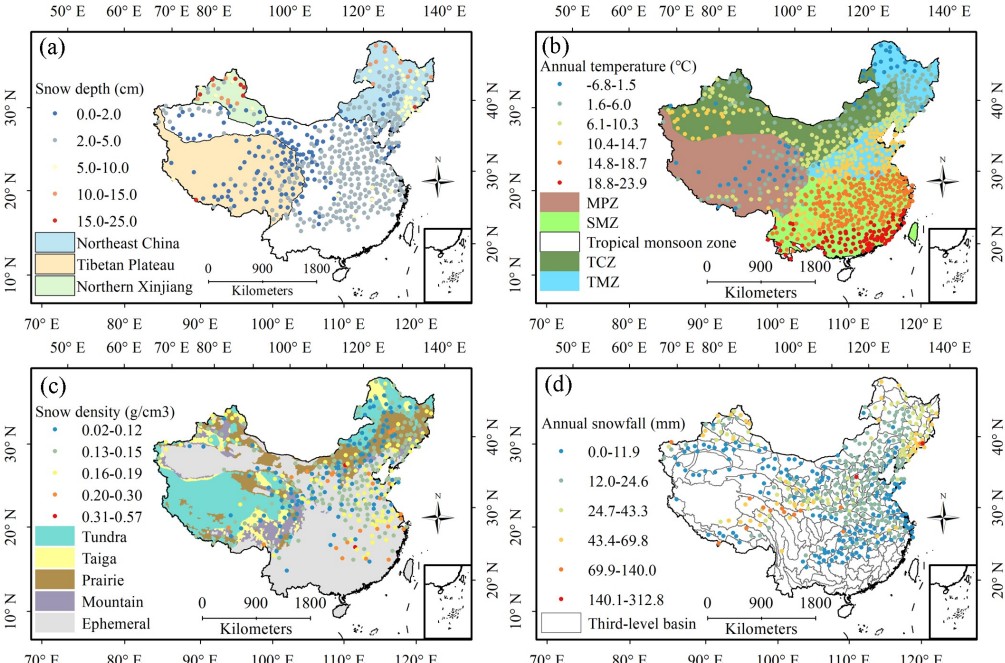

**Figure 1. The three main stable snow cover regions and the mean snow depth in China (1951-2009) (a); China's five climatic zones (MPZ, mountain plateau zone; TMZ, temperate monsoon zone; TCZ, temperate continental zone; SMZ, subtropical monsoon zone) and mean annual air temperature (1951-2017) (b); the snow cover classification and mean snow density in China (1999-2008) (c); the third-level basins and mean annual snowfall in China (1961-1979) (d).**

## 2.2 Data collection

### 2.2.1 High spatial resolution dataset of monthly air temperatures and precipitation

Data with a high spatial resolution of 0.5 seconds (approximately 1 km), including the monthly minimum, maximum, and mean temperatures ($T_{min}$, $T_{max}$, and $T_a$, respectively) and precipitation, were obtained from the Network Common Data Form (NetCDF)(https://doi.org/10.5281/zenodo.3114194 for precipitation; https://doi.org/10.5281/zenodo.3185722 for air temperatures). The data were spatially downscaled from the 30 seconds Climatic Research Unit time series data with the climatology dataset of WorldClim using delta spatial downscaling, and the data set was evaluated using observations collected from 1951 to 2016 by 496 meteorological stations across China. Detailed information on the dataset was given by Peng et al. (2019). Although the original data set covers the 1901 to 2017 period, we selected 1951 to 2017 as the study period, as only data from after 1951 have been validated by observations from meteorological stations.



### 2.2.2 Data sets for model parameterization

The observational air temperature data used to determine positive degree-day (*PDD*) parameters were collected from 824 meteorological stations (Fig. 1b) in the 1951-2017 period and were provided by the China Meteorological Administration

(http://data.cma.cn/). The positive degree-day is defined as the cumulative temperature above 0°C over a given period (Wake and Marshall, 2017), and in this study, it was the monthly positive accumulated temperature.

The observational snow density data used to determine degree-day factors (*DDF*) were collected from 417 meteorological stations in China (Fig. 1c) during 1999-2008 period and were provided by the China Meteorological Administration.

The threshold temperature parameters for determining the precipitation types at meteorological stations were obtained from

Han et al. (2010), who derived threshold temperature parameters using air temperature and precipitation data from 643 meteorological stations in China from 1961 to 1979 (precipitation types are not labelled from 1980). The threshold temperatures of each calculated cell were interpolated using the parameters from the meteorological stations.

### 2.2.3 Data sets for model evaluation

The observational snowfall data used to validate the model were collected from 475 meteorological stations in China (Fig.

1d) during the 1961-1979 period and were provided by the China Meteorological Administration.

The observational snow depth data used to validate the model were collected from 557 meteorological stations in China (Fig. 1a) during the 1951-2009 period and were provided by the China Meteorological Administration.

A long time series of daily snow depth derived from passive microwave remote sensing data (1979-2018) was obtained from by the National Tibetan Plateau Data Center (http://data.tpdc.ac.cn). The data set covers the entire land surface of China with

a spatial resolution of 0.25 degree. Detailed information on the dataset sources and product processes can be found in Che et al. (2008) and Dai et al. (2012). This data set has been widely utilized in climatic and hydrological research in China (e.g. Liu et al., 2020; Wu et al., 2021; Zhu et al., 2021). The spatial resolution of this dataset is significantly different from that of the air temperature and precipitation data used in this study. The snow cover extent measures generated from this dataset was used to validate the model.

Additionally, a long time series of daily snow water equivalent dataset derived from passive microwave remote sensing data (1980-2020) was provided by the National Cryosphere Desert Data Center (https://www.ncdc.ac.cn). The snow water equivalent dataset was produced from the passive microwave remote sensing data using the mixed-pixel method. The dataset covers the entire land surface of China with a spatial resolution of 25 km. Detailed information on the dataset sources and product processes are given by Jiang et al. (2014) and Yang et al. (2019).

### 135 2.2.4 Climate projections and downscaling

Five CIMP5 models under three different representative concentration pathways scenarios (RCP2.6, RCP4.5 and RCP8.5), namely GFDL-ESM2M, HadGEM2-ES, IPSL-CM5A-LR, MIROC-ESM-CHEM, and NorESM1-M, were selected to predict





the future snowmelt changes in China. Aforementioned climate projections have been bias-corrected downscaled to a grid with a resolution of 0.5 degrees (Hempel et al., 2013). The delta method was a statistically downscaled method corrected not

only against the average observed climate but also for the observed variance, and was used to determine the monthly future meteorological data (2006-2099) based on the high-spatial-resolution temperature and precipitation dataset and the simulations of the five CIMP5 models during the historical period (1951-2005). Detailed information on the delta method was given in Immerzeel et al. (2012) and Zhao et al. (2019).

## 3 Method

### 3.1 Snowmelt model

Temperature index models are based on an assumed relationship between snowmelt and air temperature, which is usually expressed in the form of positive temperature sums. In this study, the snowmelt was calculated as follows:

$$M_m = min(DDF_m \cdot PDD_m \cdot D_m, S_{acc,m}) \tag{1}$$

$$S_{acc,m} = S_{acc,m-1} + P_{snow,m} - S_{sub,m} - M_m \tag{2}$$

where $M$, $S_{acc}$, $P_{snow}$ and $S_{sub}$ are the snowmelt, snow accumulation, snowfall and snow sublimation (mm), respectively. $DDF$ is the degree-day factor (mm °C$^{-1}$ day$^{-1}$), an empirical factor that relates the rate of snowmelt to air temperature. $PDD$ is the accumulated positive air temperature (°C). The subscript $m$ indicates the month, and $D$ is the number of days in the month $m$.

### 3.1.1 Snowfall

The determination of the precipitation types is the first crucial step, and the distinction between rainfall and snowfall is based

on the assumption that precipitation falls either as rain, as snow or as mix, depending on two threshold temperature parameters:

$$P_{snow} = \begin{cases} 0 & T_a \geq T_{rain} \\ P \cdot (T_{rain} - T_a)/(T_{rain} - T_{snow}) & T_{snow} < T_a < T_{rain} \\ P & T_a \leq T_{snow} \end{cases} \tag{3}$$

where $P_{snow}$ and $P$ are the monthly snowfall and precipitation (mm), respectively. $T_a$ is the monthly mean temperature (°C). $T_{rain}$ and $T_{snow}$ are the threshold temperature (°C) for rainfall and snowfall, respectively.

### 3.1.2 DDF

$DDF$ is the key parameter of the temperature index snowmelt model, which depends on the snow density and varies with space and time. We used the following the empirical equation to determine the $DDF$ after Rajkumari et al. (2019):

$$DDF = 1.1(\rho_s/\rho_w) \tag{4}$$

where $\rho_s$ and $\rho_w$ are the density of snow and water (g cm$^{-3}$) , respectively, and $DDF$ is the degree-day factor (cm °C$^{-1}$ day$^{-1}$

). For taiga regions (Fig. 1c), the equation was corrected as follows:



$$DDF = 1.04(\rho_s/\rho_w) - 0.07 \tag{5}$$

The *DDF* values at the meteorological stations were calculated for each month, and were then interpolated to each calculated cell.

### 3.1.3 *PDD*

Based on the daily mean air temperatures from 1960 to 2018 at meteorological stations in China, Liu et al. (2021) proposed a simple method to calculate the monthly *PDD* according to the monthly mean temperature:

$$PDD = \begin{cases} 0 & T_a \leq T_1 \\ a \cdot T_a^2 + b \cdot T_a + c & T_1 < T_a < T_2 \\ T_a \cdot D_m & T_a \geq T_2 \end{cases} \tag{6}$$

where $T_1$ and $T_2$ are threshold temperatures (°C); $a$, $b$ and $c$ are empirical parameters.

First, $T_1$, $T_2$, $a$, $b$ and $c$ were calculated according to the measured data from meteorological stations in the four different

climate zones (Fig. 1b), and were then interpolated to each calculated cell. Table 1 shows the different threshold temperatures and empirical parameters in the four climate zones, and Fig. 2, shows the relationship between the measured monthly *PDD* and the mean temperature at meteorological stations, and the relationship between the calculated and measured monthly *PDD*, taking the MPZ as an example. Statistical analysis shows that the equation (6) can adequately calculate the monthly *PDD* (Table 1).

**Table 1. The parameters required for the calculation of the monthly accumulated positive air temperature (*PDD*) and the statistical analysis between the calculated and measured monthly *PDD* in four different climatic zones of China.**

|       | $T_1$ | $T_2$ | $a$ | $b$ | $c$ | $R^2$ | *MAE* | *RMSE* | *NSE* |
|-------|-------|-------|------|-------|-------|-------|-------|--------|-------|
| MPZ   | -7.99 | 5.79  | 0.79 | 15.37 | 56.38 | 1.00  | 5.87  | 10.85  | 1.00  |
| TCZ   | -10.85| 9.89  | 0.52 | 15.29 | 85.38 | 1.00  | 7.96  | 15.32  | 1.00  |
| TMZ   | -10.41| 9.51  | 0.52 | 15.45 | 81.43 | 1.00  | 8.45  | 16.56  | 1.00  |
| SMZ   | -4.05 | 8.56  | 0.22 | 23.12 | 49.63 | 1.00  | 2.67  | 7.63   | 1.00  |

**Note. MPZ, mountain plateau zone; TMZ, temperate monsoon zone; TCZ, temperate continental zone; SMZ, subtropical monsoon zone; $T_1$, $T_2$, $a$, $b$ and $c$, parameters in the Eq. 6); $R^2$, coefficient of determination; *MAE*, mean absolute error (°C); *RMSE*, root mean square error (°C); *NSE*, Nash-Sutcliffe efficiency.**




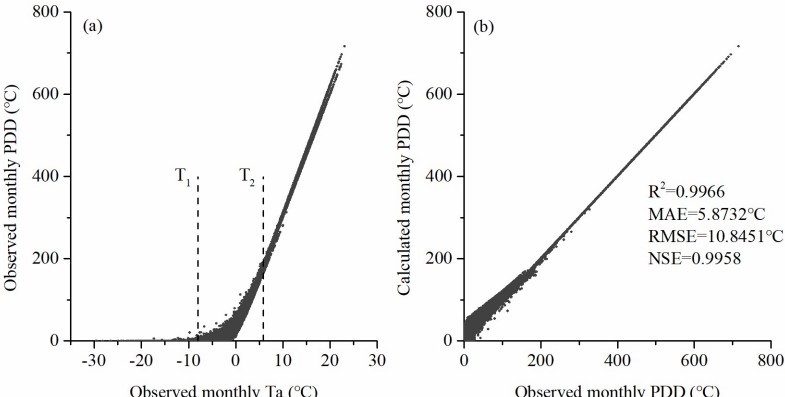

**Figure 2.** The relationship between the observed monthly accumulated positive air temperature (*PDD*) and mean air temperature (*$T_a$*) (a), and the scatter plot between the calculated and observed monthly *PDD* (b) at meteorological stations in the mountain plateau zone of China. *$R^2$*, coefficient of determination; *MAE*, mean absolute error (°C); *RMSE*, root mean square error (°C); *NSE*, Nash-Sutcliffe efficiency.

### 3.1.4 Snow sublimation

As a loss of water from the snowpack to the atmosphere, snow sublimation is difficult to quantify by measurement or modelling (Stigter et al., 2018). The bulk aerodynamic method and aerodynamic profile method are the common methods to calculate snow sublimation (Svoma, 2016). Some researchers used evapotranspiration equations (e.g. Penman-Monteith method) to estimate sublimation (Stigter et al., 2018). In general, these methods require accurate meteorological data and are very difficult to scale up from the microscale to the macroscale (Svoma, 2016). Many studies have reported the ratio of snow sublimation to snowfall (e.g. Zhang et al., 2008; Zhu et al., 2014; Sexstone et al., 2018). Therefore, considering the limited data availability and monthly scale of this study, the following simple equation was used to estimate the monthly snow sublimation.

$$S_{sub} = min(k \cdot S_{acc}, PET) \tag{7}$$

where $k \cdot S_{acc}$ and *PET* are the amount of snow accumulation available for sublimation and the potential sublimation (mm), respectively. The empirical parameter $k$ was set according to studies reporting the ratio of snow sublimation to snowfall in China and surrounding areas (Table S2 and Fig. S1), as shown in Table 2.

*PET* was calculated by the Hargreaves-Samani method (Hargreaves and Samani, 1985):

$$PET = 0.0023 \cdot R_a(T_a + 17.8)(T_{max} - T_{min})^{0.5}/\lambda \tag{8}$$

where $T_a$, $T_{max}$ and $T_{min}$ are the monthly average temperature, maximum temperature and minimum temperature (°C), respectively; $R_a$, is the extraterrestrial radiation (MJ m$^{-2}$ month$^{-1}$), which is a function of the latitude and the Julian day; $\lambda$ is the latent heat of vaporisation (MJ kg$^{-1}$).





**Table 2. The values of the empirical parameter $k$ used to estimate snow sublimation (Eq. 7) for different climate zones and different snow types**

| Snow cover type | MPZ | TCZ | TMZ | SMZ |
|---|---|---|---|---|
| Tundra | 0.68 | 0.43 | 0.37 | - |
| Taiga | 0.63 | 0.42 | 0.33 | - |
| Prairie | 0.41 | 0.22 | 0.15 | - |
| Mountain | 0.55 | 0.35 | 0.31 | - |
| Ephemeral | 0.23 | 0.10 | 0.09 | 0.08 |

Note. MPZ, mountain plateau zone; TMZ, temperate monsoon zone; TCZ, temperate continental zone; SMZ, subtropical monsoon zone.

**3.2 Snowmelt runoff ratio**

The snowmelt runoff ratio, which represents the contribution of snowmelt to river discharge, is defined as the percentage of runoff derived from snowmelt to the total runoff. Hydrological models are generally effective for determining the snowmelt runoff ratio (Immerzeel et al., 2010; Jenicek and Ledvinka, 2020), however, they are difficult to implement over large regions due to data limitations and large computational requirements. Therefore, many studies have used indirect indicators to estimate the snowmelt runoff ratio, such as the ratios of total snowfall to total precipitation, total snowfall to total runoff,

or melt season runoff to total annual runoff (Li et al., 2017). Snowmelt and rainfall are the sources of runoff generation (Vormoor et al., 2016). In this study, the snowmelt runoff ratio was calculated as the ratio of snowmelt to the sum of snowmelt and rainfall:

$$Snow_r = M/(M + P_{rain}) \times 100 \tag{9}$$

where $M$ and $P_{rain}$ are the snowmelt and rainfall (mm) , respectively, and $Snow_r$ is the percentage of runoff derived from

snowmelt to the total runoff.

**3.3 Trend analysis**

The non-parametric Mann-Kendall test (Mann, 1945; Kendall, 1975) was used to analyse the trend and significance level of the snowmelt and other variables:

$$Z = \begin{cases} (S-1)/\sqrt{var(S)} & S > 0 \\ 0 & S = 0 \\ (S+1)/\sqrt{var(S)} & S < 0 \end{cases} \tag{10}$$

$$S = \sum_{k=1}^{n-1} \sum_{j=k+1}^{n} sgn(x_j - x_k) \tag{11}$$

$$sgn(x_j - x_k) = \begin{cases} 1 & (x_j - x_k) > 0 \\ 0 & (x_j - x_k) = 0 \\ -1 & (x_j - x_k) < 0 \end{cases} \tag{12}$$

$$var(S) = \frac{n(n-1)(2n+5) - \sum_{i=1}^{m} t_i(t_i-1)(2t_i+5)}{18} \tag{13}$$





where $n$ is the number of data set, $x_j$ and $x_k$ are the data values in time series $j$ and $k$, $m$ is the number of tied groups and $t_i$ denotes the number of ties of extent $i$. A tied group is a set of sample data having the same value. Positive and negative

values of $Z$ indicate increasing and decreasing trends, respectively. Testing trends is done at the specific $\alpha$ significance level. If $|Z| > Z_{1-\alpha/2}$, the trend is statistically significant; otherwise the trend is not statistically significant. In this study, significance levels of $\alpha = 0.05$ (95% confidence level) were applied.

Additionally, Sen's slope method (Sen, 1968) was used to analyse the slope of the variation:

$$\beta = median\left(\frac{x_j - x_k}{j - k}\right), j > k \tag{14}$$

where $\beta$ sign reflects data trend reflection, while its value indicates the steepness of the trend.

**3.4 Evaluation criteria**

Statistical indices were used for quantitative analysis of the snowmelt model performance, and a series of statistical criteria used in this study as follows:

$$R^2 = [cov(X_s, X_o)/\sigma X_s \, \sigma X_o]^2 \tag{15}$$

$$MAE = \frac{1}{n}\sum_{i=1}^{n}|X_{si} - X_{oi}| \tag{16}$$

$$RMSE = \sqrt{\frac{\sum_{i=1}^{n}(X_{si} - X_{oi})^2}{n}} \tag{17}$$

$$NSE = 1 - \frac{\sum_{i=1}^{n}(X_{si} - X_{oi})^2}{\sum_{i=1}^{n}(X_{oi} - \overline{X_o})^2} \tag{18}$$

where $R^2$, $MAE$, $RMSE$ and $NSE$ are the coefficient of determination, mean absolute error, root mean square error, and Nash-Sutcliffe efficiency, respectively. $X_{si}$ and $X_{oi}$ represent simulated and observed data at time $i$, respectively. $n$ is the number of

data points, and $cov$ and $\sigma$ are the covariance and standard deviation, respectively.

**4 Results**

**4.1 Model performance**

**4.1.1 Snowfall validation using observational data from meteorological stations**

Although observational snowfall data from 475 meteorological stations in China were collected, due to data scarcity or the

very low snowfall at some stations, snowfall data from 457 stations were used to verify the snowfall from the model. Fig. 3 shows the statistical criteria between the calculated and observed snowfall at meteorological stations in China from 1961 to 1979. The *MAE* and *RMSE* varied greatly between stations due to the huge difference in the amount of snowfall at each station. The ratios of *MAE* and *RMSE* to monthly mean snowfall (*MAE/Mean* and *RMSE/Mean*, respectively) were selected to analyse the model performance. Of the 457 stations, 315 (68.9%) had $R^2 > 0.4$; 263 (5.7%) had *MAE/Mean* < 1; 274

(60.0%) had *RMSE/Mean* < 1, accounting for 60.0%; and 296 (64.7%) had *NSE* > 0.2. Among China's three main stable





snow cover regions, the most accurate snowfall simulation was obtained for Northeast China, followed by North Xinjiang and then the Tibetan Plateau.

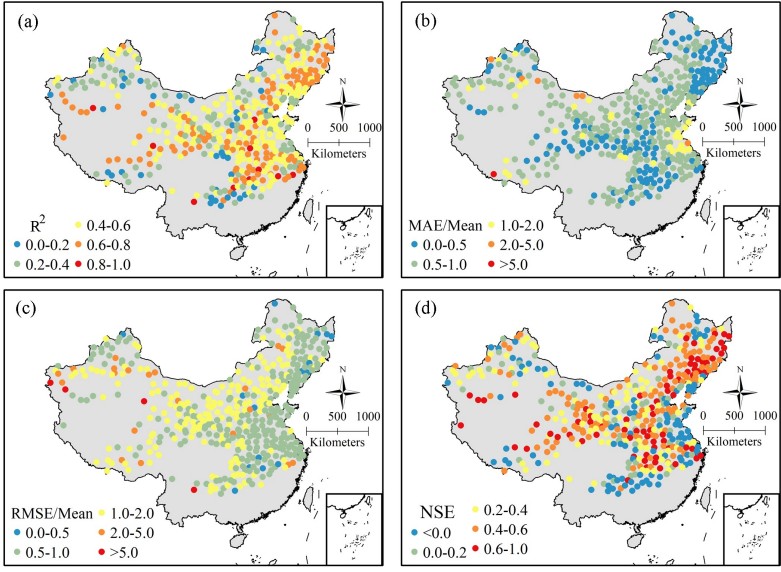

**Figure 3. Statistical criteria of the calculated snowfall against observed snowfall at 457 meteorological stations in China (a, $R^2$, coefficient of determination; b, *MAE/Mean*, *MAE*, mean absolute error; *Mean*, monthly mean snowfall; c, *RMSE/Mean*, *RMSE*, root mean square error; d, *NSE*, Nash-Sutcliffe efficiency).**

### 4.1.2 Snow depth validation using observational data from meteorological stations

The monthly snow depth in each grid was calculated from the snow accumulation and snow density. As the snow depth was observed at meteorological stations daily, the snow depth observation on the last day of each month was selected to verify the model's snow depth output. Because of the scarce data or the very shallow snow at some stations, the snow depth observations from only 264 stations were selected to verify the snow depth from the model. Fig. 4 shows the statistical analyses of the calculated and observed snow depths in China from 1951 to 2009. Similar to snowfall validation, besides $R^2$ and *NSE*, the ratio of *MAE* and *RMSE* to the mean snow depth (*MAE/Mean* and *RMSE/Mean,* respectively) were selected to analyse the model performance. Of the 264 stations, 108 (40.9%) had $R^2$ > 0.2; 221 (83.7%) had *MAE/Mean* < 2; 52 (19.7%) had *RMSE/Mean* < 2; and 105 (39.8%) had *NSE* > 0. Better snow depth simulations were obtained in regions with larger snow depths, such as Northeast China and North Xinjiang.

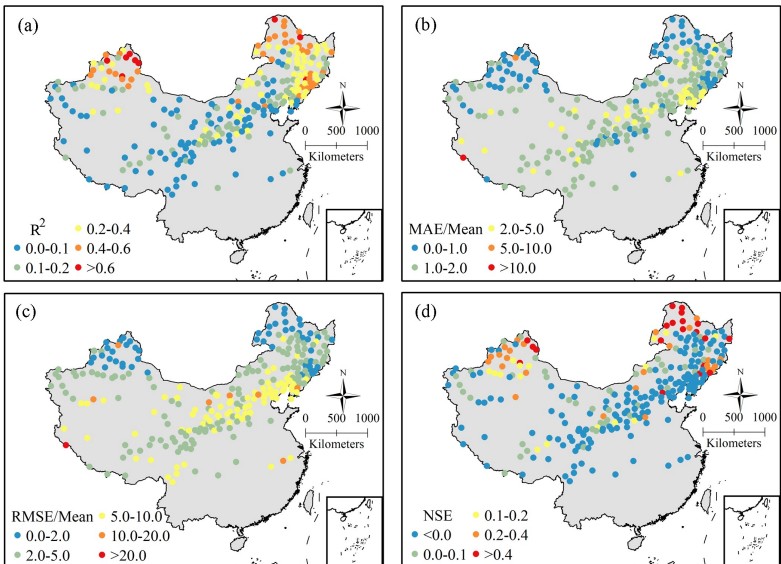

**Figure 4. Statistical criteria of the calculated snow depth against observed snow depth at 264 meteorological stations in China (a, *R²*, coefficient of determination; b, *MAE/Mean*, *MAE*, mean absolute error; *Mean*, mean snow depth; c, *RMSE/Mean*, *RMSE*, root mean square error; d, *NSE*, Nash-Sutcliffe efficiency).**

### 4.1.3 Snow cover extent validation using the dataset derived from passive microwave remote sensing

As the extend of the snow cover derived from remote sensing has a daily time scale and the snowmelt model has a monthly time scale, the remote sensing data on the last day of each month were selected to verify the model's snow cover extent output. The $R^2$, $MAE$, $RMSE$ and $NSE$ between the snow cover extent output by the model against dataset derived from remote sensing in China from 1979 to 2017, were 0.93, $0.45 \times 10^6$ km², $0.64 \times 10^6$ km² and 0.89, respectively. Among the three main stable snow cover regions, the $R^2$, $MAE$, $RMSE$ and $NSE$ were, respectively 0.81, $0.06 \times 10^6$ km², $0.09 \times 10^6$ km² and 0.76, for Northern Xinjiang, 0.93, $0.13 \times 10^6$ km², $0.21 \times 10^6$ km² and 0.87 for Northeast China, and 0.90, $0.32 \times 10^6$ km², $0.40 \times 10^6$ km² and 0.81 for the Tibetan Plateau (Fig. 5).



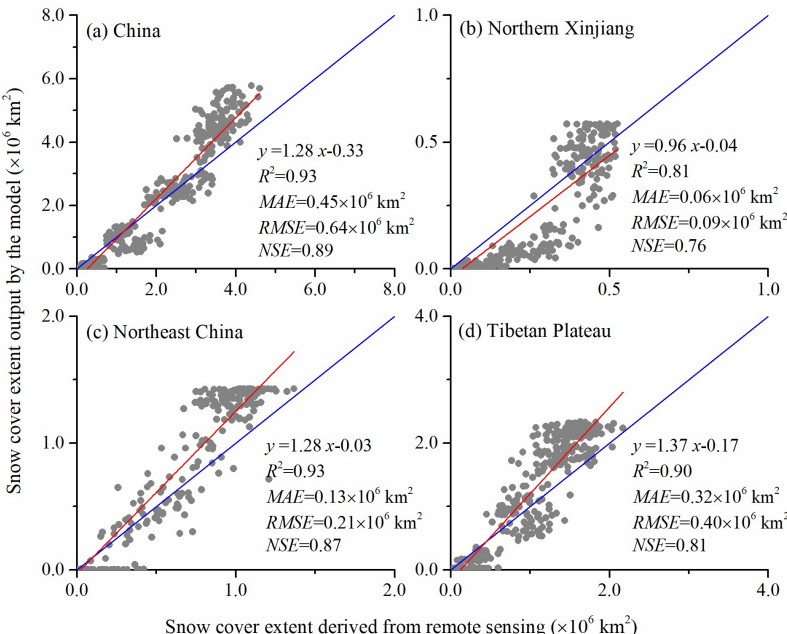

**Figure 5. Scatterplots of the snow cover extent output by the model and derived from remote sensing in China and its three main stable snow cover regions. The red and blue solid lines are the linear fit and the 1:1 line, respectively.**

**4.1.4 Snow water equivalent validation using the dataset derived from passive microwave remote sensing**

The snow water equivalent derived from passive microwave remote sensing on the last day of each month was used to verify the snow accumulation output by the model. As the spatial resolution of the snow water equivalent dataset was 25 km, and the grid scale of the model was about 1 km, to facilitate comparison, they were uniformly converted into water equivalent units of $m^3$ for different regions. The $R^2$, $MAE$, $RMSE$ and $NSE$ between the snow accumulation output by the model against snow water equivalent dataset in China from 1980 to 2017, were 0.62, $1.27 \times 10^{10}$ $m^3$, $1.27 \times 10^{10}$ $m^3$ and 0.80, respectively. The $R^2$, $MAE$, $RMSE$ and $NSE$ for the three main stable snow cover regions are shown in Fig. 6. A large number of glaciers are distributed in the Tibetan Plateau, many of which may not be recorded in the remote sensing-based snow water equivalent dataset because its spatial resolution of about 25 km is larger than the width of most glaciers. This can explain the fact that a snow water equivalent of 0 is observed in some months in this region. However, the spatial scale of the model is about 1 km, meaning that the snow accumulation in the glacier areas is always detected. This can explain the observation that in the Tibetan Plateau the modelled snow water equivalent is higher than the remote sensing-based values. When the snow accumulation in the glacier areas was not considered, the model performance improved in both the Tibetan Plateau and the whole of China ($MAE$ and $RMSE$ decreased and $NSE$ increased; see Fig. S2).

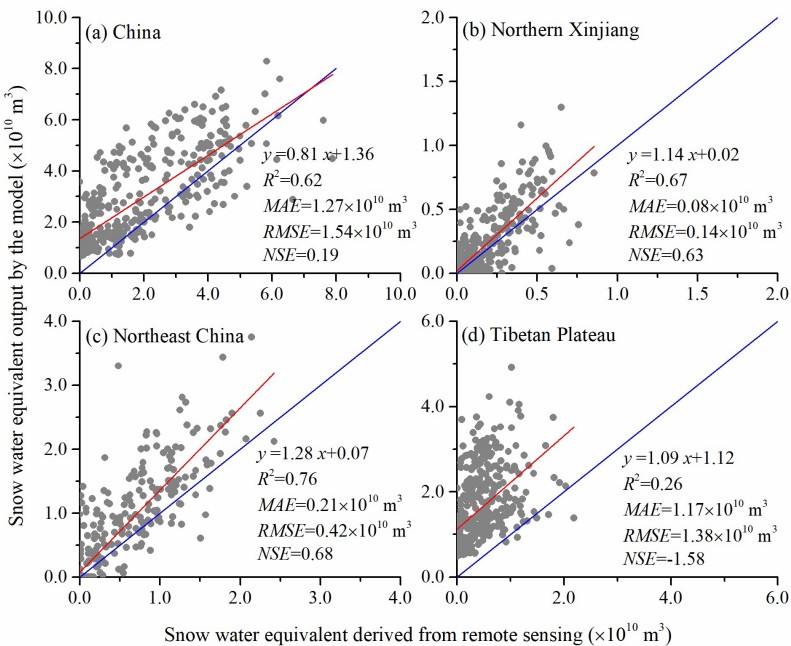

**Figure 6. Scatterplots of the snow water equivalent output by the model and derived from remote sensing in China and its three main stable snow cover regions. The red and blue solid lines are the linear fit and the 1:1 line, respectively.**

310 **4.2 Spatial and temporal variability of snowmelt**

**4.2.1 Spatial distribution**

The mean annual snowmelt in China from 1951 to 2017 was about $2.41 \times 10^{11}$ m$^3$. The three main stable snow cover regions accounted for about 72.62% of the total snowmelt in China, with the mean annual snowmelt in Northern Xinjiang, Northeast China and Tibetan Plateau being about $0.18 \times 10^{11}$ m$^3$, $0.42 \times 10^{11}$ m$^3$ and $1.15 \times 10^{11}$ m$^3$, respectively. Southeast China (mainly

315 the Huaihe River basin and the lower reaches of the Yangtze River) also had high snowmelt due to heavy snowfall (Fig. 7a). The areas with with the lowest snowmelt were mainly distributed in the arid region of Northwest China, because of this region's low precipitation, and in the humid region of South China, because of this region's high air temperature and low snowfall.

The spatial pattern of snowmelt differed considerably during the year (Fig. 8). In the three stable snow cover regions, in

320 winter, the air temperature was low and there was little snowmelt. The winter snowmelt mainly occurred in South China. In spring, North Xinjiang and Northeast China were the main regions of snowmelt in China. With further warming, the Tibetan



Plateau became the main region of snowmelt until May. In summer, there was no snowfall in most of China and snowmelt occurred only in the high mountains of the Tibetan Plateau and the Tianshan Mountains.

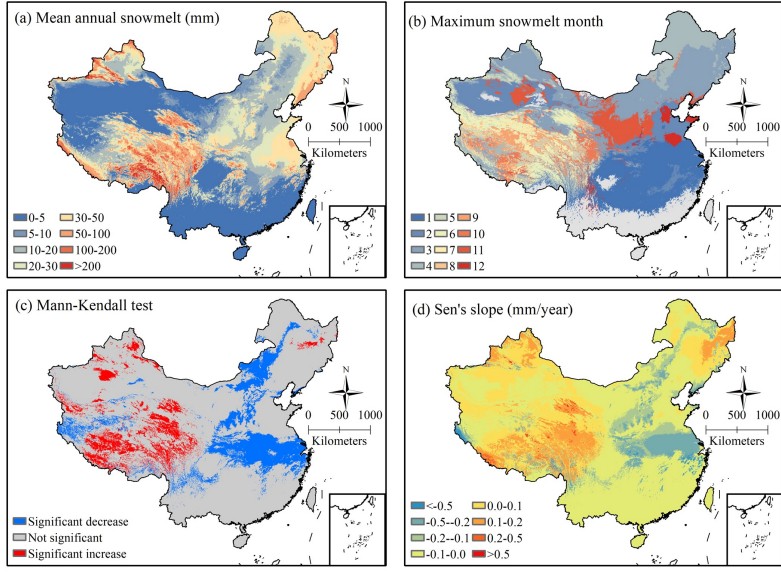

**Figure 7. Spatial distribution of mean annual snowmelt (a), the month of maximum snowmelt (b); trends of annual snowmelt based on the Mann-Kendall method (c); Sen's slope of the annual snowmelt in China during the 1951-2017 period.**

The month of maximum snowmelt differed between regions (Fig. 7b). In the Northern Xinjiang and Northeast China, due to the warming in spring, the maximum monthly snowmelt generally occurred in March, April or May. Meanwhile, in North and Southeast China, which are ephemeral snow regions where the snow melts quickly after falling, the maximum snowmelt generally occurred in the month with the largest snowfall (which occurred in winter). Because of the Tibetan Plateau's complex terrain and varied climate, the months of maximum snowmelt varied greatly across this region. In the southeastern part of the Tibetan Plateau, the month of maximum snowmelt occurred during the winter snowfall period because of the warm and humid climate. In the Gangdise Mountains, the Caidamu Basin and other colder regions, the month of maximum snowmelt generally occurred in spring. Meanwhile, in the Qiangtang Plateau, which has a drier climate, most snowfall occurred in summer, and the month of maximum snowmelt also occurred in this season. There were many high elevations mountains in the Tibetan Plateau, with air temperatures above 0 ℃ for only a few days in the summer, and the month of maximum snowmelt in those high elevations areas also occurred in summer.



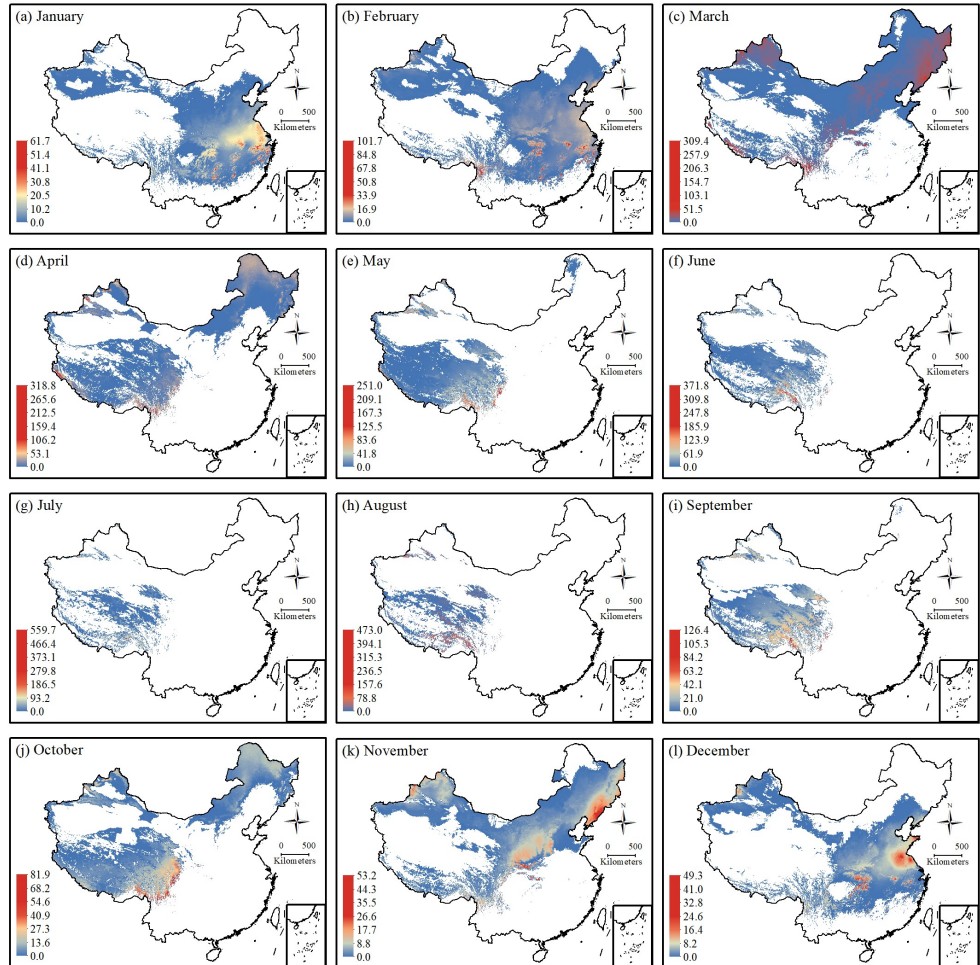

**Figure 8. Spatial distribution of the mean monthly snowmelt (mm) in 12 months in China during the 1951-2017 period.**

### 4.2.2 Temporal variations

From 1951 to 2017, the annual snowmelt increased significantly in some areas of Northern Xinjiang, although it did not increase significantly in Northern Xinjiang as a whole. In Northeast China, although the central and eastern regions showed a significant increasing trend, the southwestern regions showed a decreasing trend, leading to a slight decreasing trend (although not a significant trend) in the whole of Northeast China. Annual snowmelt increased significantly in most parts of





the Tibetan Plateau, leading to a significant increasing trend in the Tibetan Plateau as a whole. Southeast China, the annual snowfall decreased due to climate warming, and snowmelt also decreased significantly in this region , leading to a decreasing trend (although not a significant trend) of annual snowmelt in China (Fig. 7c, Fig. 7d and Fig. 9). From 1951 to 2017, the linear trends of annual snowmelt in China, Northern Xinjiang, Northeast China, and Tibetan Plateau were $-2.7 \times 10^9$

$m^3$ decade$^{-1}$, $0.2 \times 10^9$ $m^3$ decade$^{-1}$, $-0.3 \times 10^9$ $m^3$ decade$^{-1}$ and $1.4 \times 10^9$ $m^3$ decade$^{-1}$, respectively.

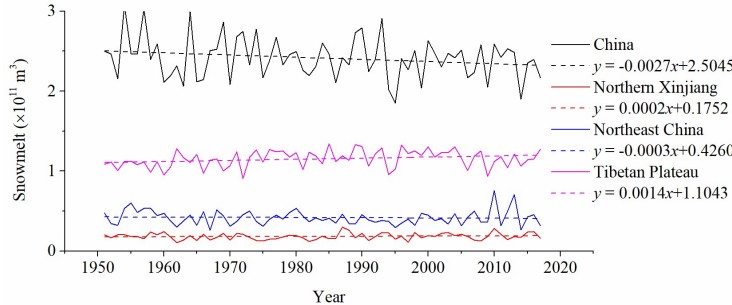

**Figure 9. Interannual variability of the mean annual snowmelt in China and its three main stable snow cover regions from 1951 to 2017.**

The temporal trend of snowmelt in each month from 1951 to 2017 varied across China (Fig. S3, Fig. S4). In Northern

Xinjiang and Northeast China, snowmelt increased significantly in March, while decreased significantly in April. This implied that the warming of these two regions led to the earlier snowmelt time. Regarding seasonal changes between 1951 and 2017, in the Tibetan Plateau, spring snowmelt increased significantly, whereas summer snowmelt decreased significantly. This may be due to the warming in spring and the reduction of snowfall in summer. In Southeast China, snowmelt experienced a decreasing trend in almost all months because of the reduction in snowfall due to climate warming.

**4.3 Spatial and temporal variability of the snowmelt runoff ratio**

**4.3.1 Spatial distribution**

Of the 210 third-level basins in China, only nine small basins near the tropical monsoon region had no snowmelt runoff from 1951 to 2017 (as snowmelt was not calculated in Taiwan, it was assumed that there was no snowmelt there). In West China, which contains two main stable snow cover region – Northern Xinjiang and the Tibetan Plateau – the mean annual snowmelt

runoff ratios were more than 10% in all basins, except those in the the Taklimakan Desert. In the basins in North and Northeast China, the snowmelt runoff ratios were generally more than 5%, whereas, due to heavy rainfall, the snowmelt runoff ratios in basins in South China were generally less than 2% (Fig. 10a).

The monthly snowmelt runoff ratio also showed large spatial differences (Fig. 11). In the cold months of November, December, January, and February, the snowmelt runoff ratios in the basins of Central and North China were over 30% due to

the precipitation being dominated by snowfall. Because of the extremely low temperatures in cold months, and there was no

snowmelt or rainfall in some basins of the three stable snow cover regions. In March, although the rainfall increased, the snowmelt from the snow that accumulated in winter also increased, and the snowmelt runoff ratios were still relatively high in most basins of North China. In April, the snowmelt runoff ratios began to decline in most basins, and basins with snowmelt runoff ratios greater than 30% were mainly located in the three stable snow cover regions. In May and June, the

snowmelt runoff ratios further decreased, dropping to zero in most of South, East, and Northeast China; however, they were still more than 30% in some basins in Northern Xinjiang and the Tibetan Plateau. In July and August, there was no snowmelt runoff in any basins except for some in Xinjiang and the Tibetan Plateau, and almost no basins had a snowmelt runoff ratio of more than 30%. In September and October, snowfall and snowmelt gradually increased, leading to a gradual increase in the snowmelt runoff ratio in many basins.

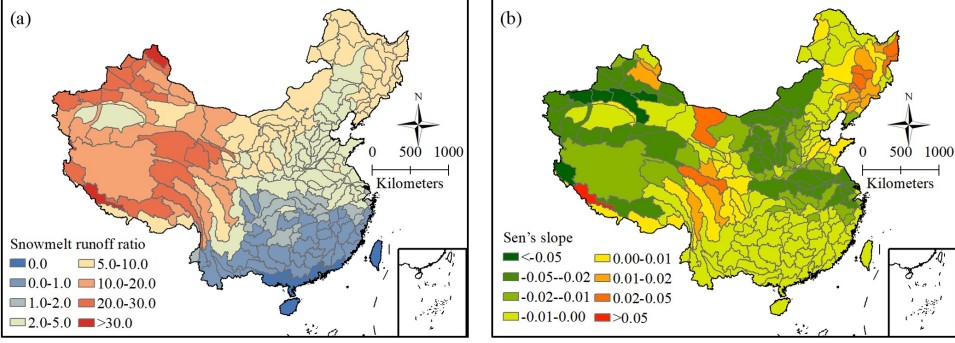


**Figure 10. Spatial distribution of the mean annual snowmelt runoff ratio (%) (a), and Sen's slope of the annual snowmelt runoff ratio (% year⁻¹) (b) in third-level basins in China during the 1951-2017 period.**

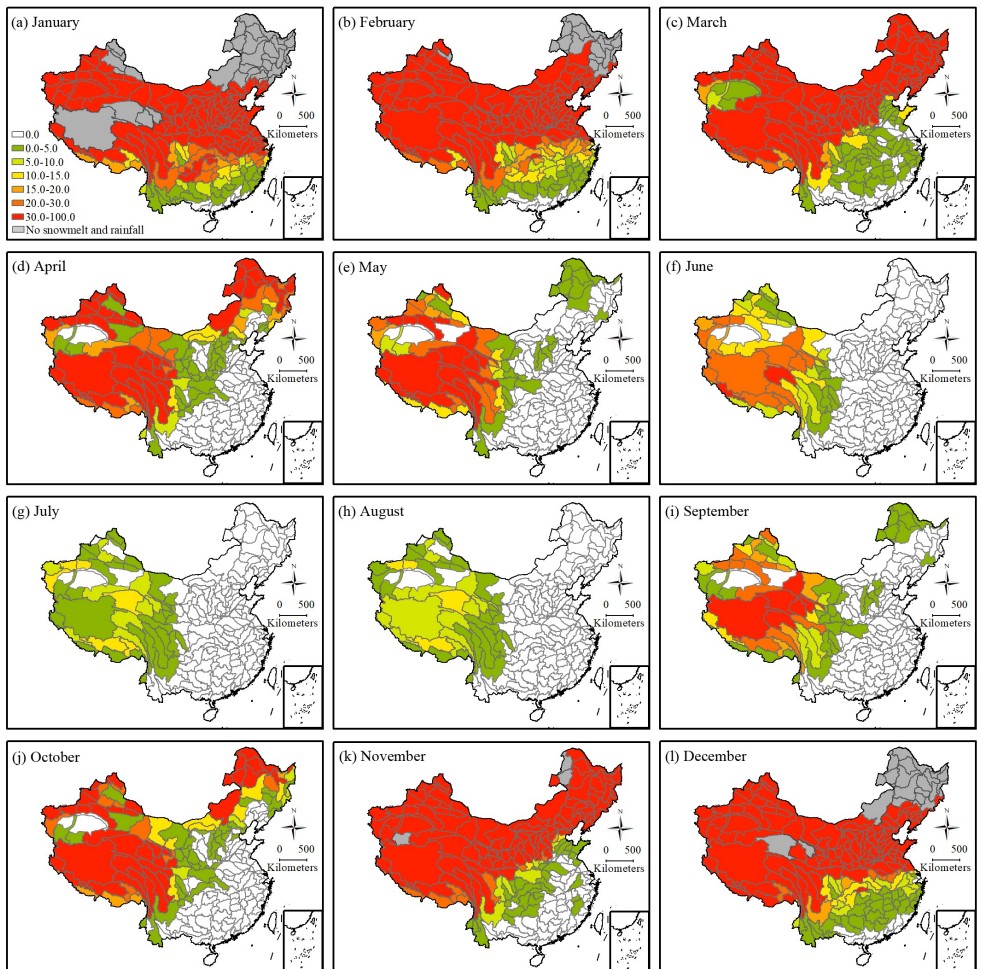

**Figure 11. Spatial distribution of the monthly mean snowmelt runoff ratio (%) in 12 months in third-level basins in China during the 1951-2017 period.**

### 4.2.2 Temporal variations

From 1951 to 2017, the Sen's slope in third-level basins showed that the annual snowmelt runoff ratio decreased in most basins (Fig. 10b). Additionally, the Mann-Kendall test showed the basins with a significant decreasing trend of the snowmelt runoff ratio were mainly distributed in central Inner Mongolia, the southern slope of the Tianshan Mountains and South China (Fig. S5). The basins with an increased snowmelt runoff ratio were mainly distributed in the southeastern the Tibetan





Plateau, the Heihe River basin in the Qilian Mountains, the Gurbantunggut Desert and Wulungu River in Northern Xinjiang, and the Songhua River basin in Northeast China. Among these basins, the Mann-Kendall test showed that only 4 basins had significant increasing trends: the source region of the Yellow River and three sub-basins of the Songhua River.

The temporal trend in the snowmelt runoff ratio from 1951 to 2017 showed spatial variations in China in every month (Fig. S6, Fig. S7). In the third-level basins in Central and East China, the snowmelt runoff ratio decreased in almost every month. In December, January and February, the snowmelt runoff ratio decreased significantly in the sub-basins of the Huaihe River and the middle and lower reaches of the Yangtze River. In March, the snowmelt runoff ratio decreased mainly in the middle reaches of the Yellow River and in the Northwest China. In April and May, the snowmelt runoff ratios decreased

significantly in Northern Xinjiang and Northeast China, and in June, July and August, they decreased significantly in the Tibetan Plateau, Tianshan Mountains and Altai Mountains. In September, the snowmelt runoff ratios decreased in only a few basins, mainly in the Tianshan Mountain and the edge of the Tibetan Plateau. In October, the snowmelt runoff ratios decreased significantly mainly in Northern Xinjiang and central Inner Mongolia in October, and in November, they decreased significantly in Southern Xinjiang and the middle reaches of the Yellow River. In some months, the snowmelt

runoff ratio increased in a few basins, mainly in the Tibetan Plateau and nearby areas. For example, in May, the snowmelt runoff ratios showed a large increase in the upper reaches of four rivers: the Yangtze River, the Nu River, the Lancang River, and the Yarlung Zangbo River; however, the Mann-Kendall test showed that few of these monthly increasing trends in third-level basins were significant (Fig. S7).

**4.4 Future changes of snowmelt under different climate scenarios**

**4.4.1 Snowmelt**

The snowmelt in Northern Xinjiang and Northeast China showed a significant decreasing trend from 2006 to 2099 under RCP8.5, but showed no significant changes under RCP2.6 or RCP4.5. The snowmelt in the Tibetan Plateau and the whole of China showed significant decreasing trends from 2006 to 2099 under all three RCPs, with the most drastic decline being under RCP8.5 (Fig. 12). The changes in snowmelt in China and its three main stable snow cover regions in the 2030s, 2050s

and 2090s under the three RCPs were shown in Table 3. In Northern Xinjiang, the total projected snowmelt in these three decades under the three RCPs were not very different than the snowmelt in the historical period (1951-2017). The models projected that snowmelt would increase in low-elevation arid areas and decrease in the higher elevation Tianshan and Altai Mountains (Fig. 13). In Northeast China, the total projected snowmelts in different decades under the three RCPs were all lower than in the historical period. The model projected that snowmelt would increase in the Greater Khingan Range and the

Songliao Plain and decrease in the Lesser Khingan and Changbai mountains. In most areas of the Tibetan Plateau, the model projected a large decrease in snowmelt (Fig. 13). Under RCP2.6, RCP4.5 and RCP8.5, the snowmelt in the Tibetan Plateau in the 2030s (2050s, 2090s) was projected to decrease by 17.7% (18.7%, 16.8%), 20.4% (22.6%, 24.2%), 20.2% (27.0%, 47.3%) compared to the historical period. Southeast China was another area where snowmelt was projected to decrease to a





large degree (Fig. 13). The model projected that the total snowmelt in China would decrease in different decades: under

RCP2.6, RCP4.5 and RCP8.5, the projected decrease in snowmelt in China in the 2030s (2050s, 2090s) was 13.4% (16.3%, 13.8%), 19.1% (19.8%, 22.5%) and 17.1% (24.7%, 42.8%) compared to the historical period.

**Table 3. Changes in mean annual air temperature ($T_a$, °C), precipitation ($P$, mm) and snowmelt ($M$, ×10$^{11}$ m³) in China and its three main stable snow cover regions during various decades under the three representative concentration pathways (RCPs). Historical period: 1951-2017.**

| Periods | China | | | Northern Xinjiang | | | Northeast China | | | Tibetan Plateau | | |
|---|---|---|---|---|---|---|---|---|---|---|---|---|
| | $T_a$ | $P$ | $M$ | $T_a$ | $P$ | $M$ | $T_a$ | $P$ | $M$ | $T_a$ | $P$ | $M$ |
| Historical period | 6.31 | 573 | 2.41 | 5.28 | 166 | 0.18 | 2.25 | 491 | 0.42 | -2.66 | 355 | 1.15 |
| RCP2.6-2030s | 7.54 | 589 | 2.09 | 7.00 | 180 | 0.18 | 3.88 | 530 | 0.42 | -1.56 | 384 | 0.95 |
| RCP2.6-2050s | 7.74 | 609 | 2.02 | 7.31 | 176 | 0.18 | 4.05 | 540 | 0.39 | -1.37 | 392 | 0.94 |
| RCP2.6-2090s | 7.63 | 602 | 2.08 | 6.96 | 183 | 0.19 | 4.02 | 529 | 0.41 | -1.53 | 395 | 0.96 |
| RCP4.5-2030s | 7.65 | 585 | 1.95 | 7.10 | 177 | 0.18 | 4.05 | 519 | 0.39 | -1.48 | 382 | 0.92 |
| RCP4.5-2050s | 8.20 | 601 | 1.93 | 7.79 | 176 | 0.17 | 4.65 | 539 | 0.41 | -0.95 | 400 | 0.89 |
| RCP4.5-2090s | 8.71 | 624 | 1.87 | 8.41 | 189 | 0.18 | 5.39 | 539 | 0.38 | -0.55 | 417 | 0.87 |
| RCP8.5-2030s | 7.78 | 581 | 2.00 | 7.33 | 181 | 0.18 | 4.13 | 514 | 0.41 | -1.31 | 385 | 0.92 |
| RCP8.5-2050s | 8.80 | 610 | 1.82 | 8.62 | 180 | 0.17 | 5.37 | 536 | 0.39 | -0.40 | 406 | 0.84 |
| RCP8.5-2090s | 11.19 | 638 | 1.38 | 11.45 | 188 | 0.15 | 8.31 | 596 | 0.34 | 1.78 | 436 | 0.61 |


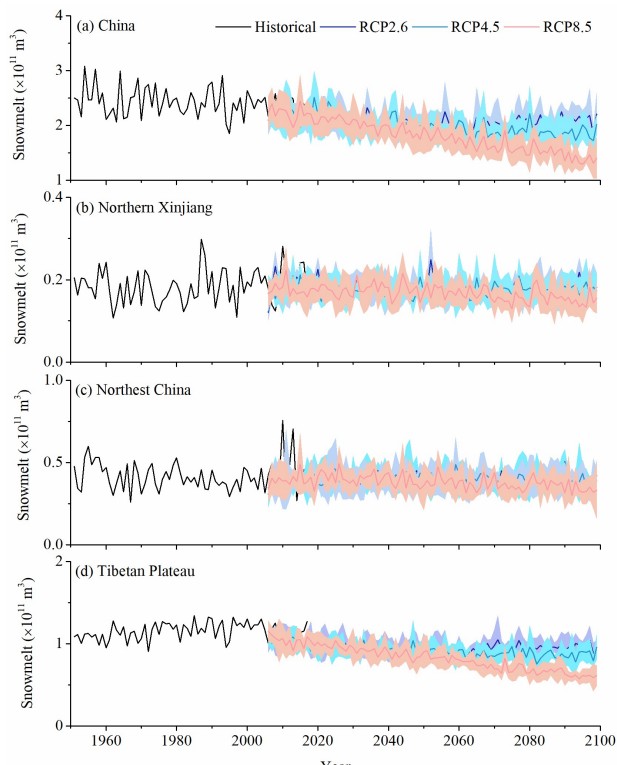

**Figure 12. Projected future changes in snowmelt in China and its three main stable snow cover regions. Historical: 1951-2017; RCP: representative concentration pathway.**


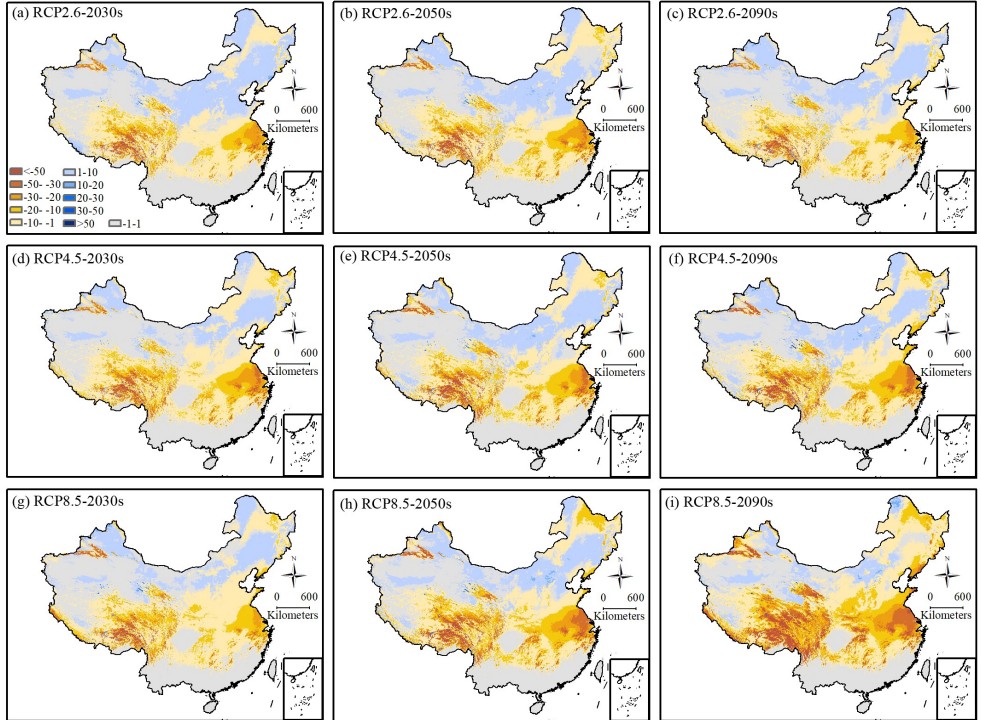

**Figure 13. Spatial distributions of differences (mm) between the projected mean annual snowmelt and the historical period (1951-2017) for the 2030s, 2050s and 2090s under RCP2.6, RCP4.5 and RCP8.5 in China.**

**4.4.2 Snowmelt runoff ratio**

Under the three RCPs, the projected mean annual snowmelt runoff ratios in the third-level basins in different decades were mostly smaller than those in the historical period, except for a few basins in Southern Xinjiang and North China (Fig. 14). In general, the largest decreases in snowmelt runoff ratio in the basins were projected to occur by the 2090s, followed by the 2050s and 2030s. The largest decreases were projected under RCP8.5, followed by RCP4.5 and RCP2.6. Among the three main stable snow cover regions, the snowmelt runoff ratios were projected to decrease the most in basins in the Tibetan

Plateau, followed by basins in Northern Xinjiang and Northeast China. Under RCP8.5, in the 2090s, the projected mean annual snowmelt runoff ratios were lower than those in the historical period in all basins except the three basins near the Taklimakan Desert, and the snowmelt runoff ratios in the Tibetan Plateau and Tianshan Mountains were projected to decrease by more than 5% in most basins and by more than 10% in a few basins relative to the historical period (Fig. 14i).

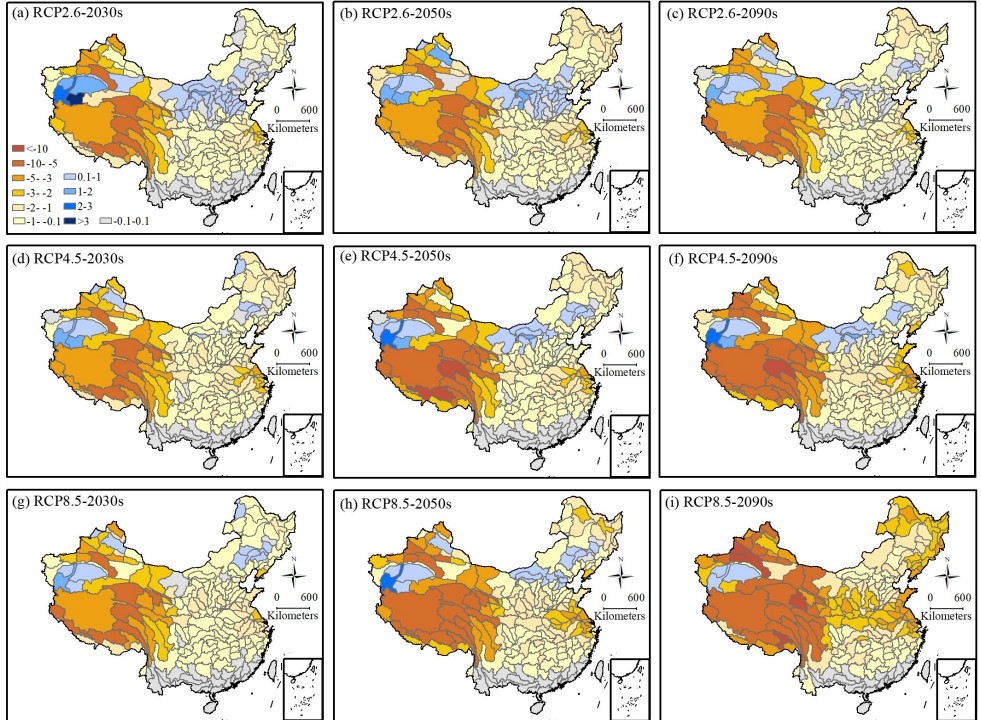

**Figure 14. Spatial distributions of differences (%) between the projected mean annual snowmelt runoff ratio and the historical period (1951-2017) for the 2030s, 2050s and 2090s under RCP2.6, RCP4.5 and RCP8.5 in China.**

## 5 Discussion

### 5.1 Model Evaluation

Snowmelt is difficult to measure directly, and therefore, the model outputs of snowfall, snow depth, snow cover extent and snow water equivalent were selected to verify the model performance. Although the model was solely driven by air temperature and precipitation, the model outputs showed acceptable performance compared with the results from other studies. Han et al. (2021) used rainfall and snowfall temperature thresholds to identify precipitation types in the Lancang River basin in Southwest China, and obtained $R^2$ values between the simulated snowfall and the snowfall observed at three meteorological stations of 0.42, 0.34 and 0.61, respectively; in this study, the $R^2$ values at the same stations were 0.72, 0.46 and 0.39, respectively. Li et al. (2020) used temperature thresholds to calculate the snowfall in the Tianshan Mountains of Central Asia, and obtained a mean $R^2$ value between the simulated snowfall and the snowfall observed at 27 meteorological





stations of 0.61; in this study, the mean $R^2$ value at 50 stations in Xingjiang was 0.39. Zhong et al. (2018) discriminated the precipitation phase based on temperature thresholds in the Songhua River basin, Northeast China, and obtained $R^2$ values between simulated and observed snowfall of less than 0.3 for most meteorological stations; in this study, the $R^2$ was larger than 0.3 for most stations.

Snow depth was the worst-performing model output, this was mainly for the following reasons. (1) The model was performed on a monthly scale, whereas the output data were compared with the observed snow depth measured on the last day of each month, which increased the error. (2) The output data were at the grid scale, whereas the observed snow depth was at the site scale, where the snow properties were not always representative of snow at grid (Sexstone et al., 2020). (3) Snow depth itself is difficult to simulate, and even the snow depth retrieved by remote sensing have been shown to have high uncertainty. Orsolini et al. (2019) found that the mean annual snow depth at 33 meteorological stations in the Tibetan Plateau based on multiple global reanalysis products was 1.38 cm to 11.71 cm, with a mean value of 7.88 cm, while the observed snow depth was 0.23 cm. Furthermore, Bin et al. (2013) evaluated snow depth obtained from five algorithms using AMSR-E passive microwave against ground observations from meteorological stations across China, and found that the $RMSE$ varied from 6.85 cm to 16.79 cm in Xinjiang region, and from 6.21 cm to 18.05 cm in Northeast China. In this study, the $RMSE$ varied from 0.56 cm to 13.32 cm in Xinjiang, and 0.54 cm to 9.00 cm in Northeast China. Additionally, many studies have shown that the retrieved snow depth is more accurate in regions with larger snow depth (e.g. Zhou et al., 2017; Wang and Zheng, 2020); and in this study, the performances in the regions with larger snow depth such as Northern Xinjiang and Northeast China were also much better than those in other regions.

Among the simulated snow properties, the snow cover extent showed the best performance, with $R^2$ and $NSE$ values above 0.80 (Fig. 5). The performance of the snow water equivalent was acceptable in northern Xinjiang and Northeast China but was poor in the Tibetan Plateau (Fig. 6). There are several possible reasons for this difference. (1) The accuracy of the driving data of precipitation and temperature in the Tibetan Plateau was lower than that in other regions (Peng et al., 2019). (2) Due to the sparse distribution of meteorological stations in the Tibetan Plateau and the fact that most of these were located at low elevations, the reliability of the model parameters might be worse in this region than in other regions. (3) The snow water equivalent data used for verification have large uncertainties in the Tibetan Plateau because of its high elevation and complex terrain and climatic conditions (You et al., 2020).

In summary, the verification of snowfall, snow depth, snow cover extent and snow water equivalent suggests that the model was reliable for calculating the snowmelt in China.

## 5.2 Influence of changes in temperature and precipitation on snowmelt

Snowmelt is sensitive to both temperature and precipitation, and the relationship between snowmelt and warming is more complex than monotonic declines (Mankin et al., 2015). In China, climate warming has led to temperature increase, and snowmelt is increasing significantly in some regions (Fig. 7c). The grids with significant changes in snowmelt from 1951 to 2017 were selected to analyse the changes in air temperature, precipitation and snowfall (Fig. 15). In the regions where





snowmelt changed significantly, the temperature generally increased (although not all significantly). Precipitation decreased
(increased) significantly in many regions where snowmelt increased (decreased) significantly. The regions with significant
changes in snowmelt were relatively consistent with the regions with snowfall changes. There were no girds where snowfall
increased significantly and snowmelt decreased significantly. In a few grids, snowfall decreased significantly and the
snowmelt increased significantly. These grids were too few and could be ignored (Table S2). Compared with the total annual

precipitation, the change of snowfall has more influence on snowmelt. Tan et al. (2019) analysed the relationship between
snow cover days and precipitation in China and found that snow cover days were highly correlated with winter precipitation,
but were not correlated with spring precipitation. In global climate models, there is more uncertainty about precipitation than
temperature (Woldemeskel et al., 2016). Many assessments of precipitation from climate models were performed at the
annual scale (e.g. Woldemeskel et al., 2016; Ahmed et al., 2019; Yue et al., 2019). Assessing precipitation from climate

models using multiple time scales or seasons may allow a more accurate prediction of snowfall and snowmelt changes.

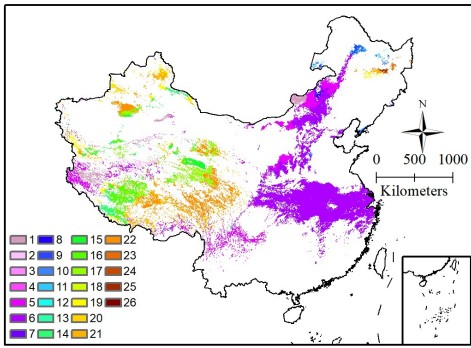

**Figure 15. Trends of the annual snowmelt, precipitation, snowfall and air temperature based on the Mann-Kendall method in China during the 1951-2017 period. The meanings of the numbers in the legend are shown in Table S2.**

As snow cover formation and snowmelt are closely related to the temperature threshold of 0 ℃, temperature change near 0 ℃

is more likely to trigger a drastic change in snowmelt. This may partly explain for the spatial distribution of snowmelt
changes in China. In Northern Xinjiang and Northeast China, the main stable snow cover regions in China, the changes in
snowmelt from 1951 to 2017 were not significant, and the future projections suggest that the changes in snowmelt in these
regions will not be drastic. In these two regions, the snow season temperatures are well below the freezing (Qin et al., 2006),
and the snowmelt trend depends on the snowfall trend rather than the temperature trend. There is no reason for snowmelt to

change significantly when precipitation does not change significantly and temperature is still to remain well below freezing
in those two regions. Therefore, in Northern Xinjiang and Northeast China, snowmelt is unlikely to change significantly
when precipitation does not change significantly and the temperature remains well below freezing. In Southeast China, the
snowmelt decreased significantly from 1951 to 2017, and the model projected that snowmelt will decrease further under
RCP2.6, RCP4.5 and RCP8.5. The climate in this region is warm with average monthly temperatures near 0 ℃ in winter



(Zeng et al., 2016). Precipitation types are sensitive to temperature increase, and under climate change, less precipitation falls as snow and snowmelt decreases. With high elevation and low temperature, the Tibetan Plateau experienced the most complex changes in snowmelt from 1951 to 2017. Temperature significantly increased in the Tibetan Plateau and the annual precipitation increased in most areas of the region (Kuang and Jiao, 2016). This study showed that the snowmelt in the Tibetan Plateau increased significantly from 1951 to 2017. High elevations tend to be colder and therefore can tolerate more

warming (Livneh and Badger, 2020). The increased precipitation in the Tibetan Plateau may have offset the decrease in snowfall caused by the temperature increase, and precipitation may be a more influential control on snowmelt in that region than temperature. The results of CIMP5 models showed that the snowmelt in the Tibetan Plateau may decrease in the future. In this region, further warming could cause the temperature threshold for snow formation to be reached, resulting in less snowfall and snowmelt at medium and low elevations, and the temperature may become a more influential control on

snowmelt.

### 5.3 Impact of snowmelt change on regional water supply

China's three main stable snow cover regions (Northern Xinjiang, Northeast China and the Tibetan Plateau) are located in three different climate areas (TCZ, TMZ and MPZ, respectively). Besides these three regions, the part of Southeast China (mainly the Huaihe River basin and the lower reaches of the Yangtze River) that is in the SMZ also has relatively high

snowmelt (Fig. 7a). In the following, we examine these four regions to shed light on the potential regional water supply problems that may be associated with changes in snowmelt under climate change.

### 5.3.1 Northern Xinjiang

Northern Xinjiang is located in the TCZ, and snowmelt is an important source of fresh water in this region (Chen et al., 2020; Wu et al., 2021). This study showed that between 1951 and 2017 the mean annual snowmelt runoff ratios in the third-level

basins in Northern Xinjiang were above 10% and exceeded 30% in some basins. Although the annual snowmelt showed a significant increasing trend from 1951 to 2017, the increase was mainly in November. Because of the earlier snowmelt time under climate warming, the monthly snowmelt increased in March and decreased significantly in April. The snowmelt runoff ratios in April were generally more than 30%, also showed a significant decreasing trend. March and April are times of plant and crop cultivation, and the water requirements for agricultural irrigation therefore increase sharply in these months. The

shift of the snowmelt time may reduce the water that is available for agriculture because of the mismatch between snowmelt and crop-growing (Notarnicola, 2020), and changes in the snowmelt amount and timing may bring agricultural risks in the Northern Xinjiang.

### 5.3.2 Northeast China

In Northeast China, the runoff in March and April is mainly generated from snowmelt. From 1951 to 2017, snowmelt in

March increased while that in April decreased significantly. Due to its location in the TMZ, which has high levels of





precipitation, the snowmelt runoff ratios in the basins in Northeast China are generally about 10%, and the importance of snowmelt as a water resource in this region may be less than in Northern Xinjiang. However, because of the cold and long winter, snow plays a role as a natural reservoir to store water in winter, and snow melting releases water in spring, which can influence agriculture by affecting soil moisture (Qi et al., 2020). Snowmelt is vital for seasonal water supply in Northeast

China , which is an important region for agricultural production in China. Because of the changes in snowmelt documented in this study, changes in water resource allocation in spring should be considered to meet the demands of spring crop planting.

### 5.3.3 Tibetan Plateau

The Tibetan Plateau is known as the "water tower" of Asia, since many of the continent's major rivers originate there (e.g.
the Yangtze, Yellow, Indus, Mekong, Brahmaputra, Salween and Ganges rivers). The western and northern parts of the Tibetan Plateau are the source of many arid inland rivers such as the Hetian and Heihe rivers. Additionally, the Tibetan Plateau contains arid inland basins (e.g. the Qiangtang inland basin). These waters from the Tibetan Plateau have been sustaining life, agricultural, and industrial water usage for nearly 40% of the world's population (Xu et al., 2008). Due to its high elevation and low temperature, the mean annual snowmelt runoff ratios in the basins of the Tibetan Plateau are
generally greater than 10%. Snowmelt contributes runoff every month, and in some months all of the runoff is contributed by snowmelt. The snowmelt in the Tibetan Plateau showed an increasing trend from 1951 to 2017, and as the temperature is projected to rises further in the future, it is likely that both the snowmelt and snowmelt runoff ratio will decrease in the near future. Other studies have similarly concluded that the snowmelt and snowmelt runoff ratio in the Tibetan Plateau will decline in the future (Qin et al., 2020; Kraaijenbrink et al., 2021). The Tibetan Plateau and its surrounding downstream areas
are projected to experience declines in the share of water from snowmelt and will thus require increases in alternative water supplies.

### 5.3.4 Southeast China

Southeast China is located in the SMZ, where precipitation is relatively high. The snowfall in winter is high, resulting in a large amount of snowmelt in this region. However, due to this region's high precipitation, snowmelt contributes relatively
little to its water resources, with the snowmelt runoff ratios being less than 2% in most basins. Although climate change has caused a significant decrease in snowmelt in this region, and this trend is projected to continue in the future, the impact on water supply is likely to be much smaller than in other regions of China. The increased frequency and intensity of flooding that are projected to be caused by climate change is a major concern for this region (Qin and Lu, 2014).

### 5.4 Uncertainties and limitations

The snowmelt simulation reported in this study was largely driven by the climatology dataset of WorldClim using delta spatial downscaling, and any uncertainties in the dataset were likely to propagate to the snowmelt simulation. Many studies





showed that, although the WorldClim data were adequate for air temperatures, there were large errors in its precipitation data, especially in mountainous areas with complex topography and high elevations (Beck et al., 2020; Bobrowski et al., 2021). Although the downscaled dataset generated by bilinear interpolation method containing detailed topographic information, as

well as the effects of distance to the nearest coast and satellite-derived covariates, was of high quality, there was still a gap between the downscaled data and the observed data, especially in the mountainous areas (Peng et al., 2019; Ding and Peng, 2020). Uncertainties in the CIMP5 models (Woldemeskel et al., 2016) and the downscaling method (Lima et al., 2021) might introduce uncertainties in the assessments of future changes in snowmelt.

As the results of this study were based on simple temperature index model simulations, the uncertainty arising from the

model parameterization needs to be addressed. There were three important parameters in the model: the threshold temperatures for the separation of rainfall and snowfall ($T_{rain}$ and $T_{snow}$), the degree-day factors ($DDF$) and the ratio of snow sublimation to snow accumulation ($k$). Although $T_{rain}$ and $T_{snow}$ were derived from observations at meteorological stations, errors and uncertainties may have been increased when the values were interpolated into the grid. The $DDF$ was estimated using an empirical method that depends on the snow density observed at meteorological stations. Due to the uneven

distribution of meteorological stations, especially few and sparse meteorological stations in the Tibetan Plateau, and the uncertainties introduced by the spatial interpolation of data from meteorological stations (Zhang and Ma, 2018), the errors of these parameters may have led to the uncertainty in the snowmelt calculation. Additionally, snow sublimation is very difficult quantify by measurement or modelling (Stigter et al., 2018). Due to the limited data availability and the complexity of snow sublimation, this study used the ratio of snow sublimation to snow accumulation to simplify the calculation of

sublimation. However, this ratio can vary considerably both spatially and temporally, having been estimated to vary between 0.1 and 90% around the world (Stigter et al., 2018). The simplified calculation of snow sublimation and the failure to consider the monthly variability of this parameter may have increased the uncertainty of the snowmelt calculation.

Moreover, there are many glaciers in West China, and there are differences in the processes and parameters of glacier melting and snow melting (Terink et al., 2015; Armstrong et al., 2018). In this study, snowmelt was not clearly distinguished

in the glacier area, which is another source of uncertainty in the snowmelt calculation.

In addition to snowmelt, the snowmelt runoff ratio was another important research object in this study. We estimated this ratio using snowmelt / (snowmelt + rainfall). However, this method may underestimate the ratio because snowmelt is more effective at generating catchment runoff than rainfall (Li et al., 2017; Jenicek and Ledvinka, 2020).

## 6 Conclusions

A simple temperature index model was used to calculate a time series of monthly snowmelt at a resolution of 0.5 seconds in China for the 1951-2017 period. The mean annual snowmelt in China was $2.41 \times 10^{11}$ m³, and the mean annual snowmelt in Northern Xinjiang, Northeast China and Tibetan Plateau was approximately $0.18 \times 10^{11}$ m³, $0.42 \times 10^{11}$ m³ and $1.15 \times 10^{11}$ m³, respectively. From 1951 to 2017, the annual snowmelt increased significantly in the Tibetan Plateau, and decreased




significantly in the North, Central and Southeast China. There was a decreasing trend in snowmelt in China, although this
trend was not statistically significant. In West China, the mean annual snowmelt runoff ratio was more than 10% in almost
all third-level basins, except for those in the Taklimakan Desert. In basins in North and Northeast China, the snowmelt
runoff ratio was generally more than 5%, whereas in basins in South China it was less than 2%. The Sen's slope showed that
from 1951 to 2017, the annual snowmelt runoff ratio decreased in most third-level basins in China, where this decrease was
significant were mainly distributed in central Inner Mongolia, the southern slopes of the Tianshan Mountains and South
China.

Under RCP2.6, RCP4.5 and RCP8.5, the snowmelt in China showed a significant decreasing trend from 2006 to 2099, and
the projected decrease in snowmelt in the 2030s (2050s, 2090s) was 13.4% (16.3%, 13.8%), 19.1% (19.8%, 22.5%), 17.1%
(24.7%, 42.8%) compared to the historical period (1951-2005), respectively. Among China's three main stable snow cover
regions, the Tibetan Plateau was projected to experience the largest decrease in snowmelt in the future, followed by
Northeast China; the snowmelt in the Northern Xinjiang was not projected to change significantly in the future. Under the
three RCPs, the projected mean annual snowmelt runoff ratios in the third-level basins in different decades were less than
those in the historical period, except for a few basins in Southern Xinjiang and North China, with the largest decrease
projected to occur in the Tibetan Plateau.

This study also investigated the spatial variability of snowmelt changes caused by changes in temperature and precipitation.
It found that in low temperature regions (which can tolerate more warming), the snowmelt change was mainly influenced by
precipitation, whereas in warm regions, snowmelt change was most sensitive to temperature increases. The spatial variability
of snowmelt changes may lead to the regional differences in the impact of snowmelt on water supply.

Although some uncertainties were introduced by the model principles, driving data and parameterization and other factors,
this study is the first attempt to quantify snowmelt and its changes in China. Given the importance of snowmelt, the results
have important implications for future researches on snow-hydrology-climate interactions and contribute to water resource
management planning under climate change.

*Data availability.* The spatial resolution with 0.5 seconds, including monthly minimum, maximum, and mean temperatures
($T_{min}$, $T_{max}$, and $T_a$) and precipitation, are obtained from Network Common Data Form at
https://doi.org/10.5281/zenodo.3114194 for precipitation and https://doi.org/10.5281/zenodo.3185722 for air temperatures.
The air temperature, snowfall, snow depth and snow density observation data are obtained from the China Meteorological
Administration (http://data.cma.cn/). The snow depth dataset derived from passive microwave remote sensing data is
obtained by the National Tibetan Plateau Data Center (http://data.tpdc.ac.cn). The snow water equivalent dataset derived
from passive microwave remote sensing data is obtained by the National Cryosphere Desert Data Center
(https://www.ncdc.ac.cn). The CMIP5 data set is distributed by the Inter-Sectoral Impact Model Intercomparison Project on



its own website (http://www.isi-mip.org). The third-level basins dataset is obtained by the Resource and Environment Science and Data Center (https://www.resdc.cn/).

*Author contributions.* RC initiated the study. YY, RC and GL developed the methodology. YY performed all analyses. YY and ZL prepared the input meteorological data. GL and XW prepared the CIMP5 models data. YY prepared the manuscript with contributions from RC.

*Competing interests.* The authors declare that they have no conflict of interest.

*Acknowledgments.* The authors would like to acknowledge data from the Network Common Data Form, China Meteorological Administration, National Tibetan Plateau Data Center, National Cryosphere Desert Data Center, Resource and Environment Science and Data Center, and Inter-Sectoral Impact Model Intercomparison Project. The authors would like to thank Prof. Xiongfeng Li for providing the snow cover classification in China, and Dr. Chuntan Han for providing the rain/snow separation threshold temperatures in China. This work was carried out with financial support from the National Key Research and Development Project (2019YFC1510505), National Natural Sciences Foundation of China (41690141), Joint Research Project of Three-River Headwaters National Park, Chinese Academy of Sciences and the People's Government of Qinghai Province (LHZX-2020-11), and National Natural Sciences Foundation of China (41901084).

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
