# Peer review of "Trends and variability in snowmelt in China under climate change"

_Hydrology and Earth System Sciences, 2021_

## Author Comment (AC1)

Responses to the comments from Anonymous Reviewer #1

The manuscript presents a country-wide assessment on present and future changes of snowmelt in China. To this end, a simplified temperature-index model is used that simulates several snow properties including snowmelt, snowfall, and snow sublimation. The model is forced with high resolution temperature and precipitation datasets as well as datasets that are needed for the model parametrization, which include PDD, snow density and snow/rain threshold temperatures. The model outputs, snowmelt, snow depth, snow water equivalent, and snow cover extent, are validated on equivalent station-based or grid-based observation datasets. Finally, the model is forced with temperature and precipitation data for 5 CMIP5 models under 3 different RCPs are selected to project future changes of snowmelt in China. The manuscript is well structured and generally well written. However, there are some issues with the grammar in this manuscript since most of the manuscript has been written in a past tense, whereas I would expect that some sections of the manuscript could have been written in a present tense or future tense. The topic the manuscript covers is very interesting. The assessments presented in this paper are novel as well, but I have several major and minor comments that need to be addressed.

Response: We would like to sincerely thank you for taking the time to carefully read and review our manuscript. We have revised the manuscript according to your valuable comments and suggestions. The responses following a point-by-point to the comments are provided below.

**General Comments**

•   My first general comment is on the grammar that has been used while writing the manuscript. Most of the manuscript is written in a past tense. That is fine in some sections, such as the Abstract, but insufficient in those sections where a present or future tense is warranted.

Response: We thank you for this comment. We have revised the tenses of some sentences.

•   To compare future snowmelt with present snowmelt I think it would be more beneficial to use climate "reference" periods instead of comparing future decadal values with the mean values of an entire historical period since the entire period reflects different climatological

characteristics. For example, since the late 1970s/early 1980s the global warming accelerated (Hartmann et al., 2013; doi:10.1017/CBO9781107415324.008), which might have had a different impact on the snow climatology than before the late 1970s. Therefore, I would like to suggest using 1981-2010 as an historical "reference period" and to use 2010-2039, 2040-2069, and 2070-2099 as near-future, mid-future, and far-future "reference" periods, respectively. That might give a better representative view on future snowmelt changes.

Response: Thank you for this comment, and we have learned a lot from it. In the revised manuscript, we evaluate the snowmelt in China over projected climatic scenarios in different time frames i.e., near-future (2010-2039), mid-future (2040-2069), and far-future (2070-2099) in comparison with the reference period (1980-2009). The reviewer suggested using 1981-2010 as the reference period, and 2010-2039 as the near future period. In our opinion, if 2010 has been classified into the near future period, it may be better to use 1980-2009 as the reference period.

• Section 2.2.2 L108: Why did the authors use observational air temperature data from 824 stations, whereas they already mentioned before that they would like to use a high-resolution air temperature dataset to force the model. What is the added value of the observational air temperature data? What is the bias between the observational air temperature data and the high-resolution dataset of Peng et al., 2019 and to what is the difference between PDDs retrieved from the high-resolution dataset and PDDs retrieved from the observational data? Please elaborate on this.

Response: The high-resolution temperature dataset includes the monthly minimum, maximum, and mean temperatures, but no monthly PDD (accumulated positive air temperature) used to calculate snowmelt in this manuscript. The monthly PDD were calculated according to the monthly mean temperature according to Eq. 6.

$$PDD = \begin{cases} 0 & T_a \leq T_1 \\ a \cdot T_a^2 + b \cdot T_a + c & T_1 < T_a < T_2 \\ T_a \cdot D_m & T_a \geq T_2 \end{cases} \tag{6}$$

The value of the observational air temperature data from 824 stations is to determine PDD parameters in Eq. 6, i.e., $T_1$, $T_2$, $a$, $b$, and $c$. Table 1 in the original manuscript (Table 2 in the revised manuscript) shows the parameters required for the calculation of the monthly PDD.

The biases between the observational data and the high-resolution dataset have been discussed in detail in Peng et al. (2019). We have added some sentences in the section "2.2.1 High spatial resolution dataset of monthly air temperatures and precipitation" to introduce the performances of the high-resolution dataset, and the added sentences are as follows: The data were spatially downscaled from the 30′ Climatic Research Unit time series data with the 0.5′ climatology dataset of WorldClim using delta spatial downscaling. Peng et al. (2019) evaluated this dataset using observations collected from 1951 to 2016 by 496 meteorological stations across China, and the average values of the coefficient of determination ($R^2$), mean absolute error *(MAE)*, root mean square error (*RMSE*), and Nash-Sutcliffe efficiency (*NSE*) for the monthly mean temperature at all meteorological stations were 0.996, 0.820 °C, 0.969 °C and 0.981, respectively, and for the monthly precipitation were 0.863, 13.269 mm, 21.941 mm and 0.808, respectively (Peng et al., 2019).

We have presented the statistical analysis between PDDs retrieved from the high-resolution dataset and PDDs retrieved from the observational data in four different climatic zones of China, and the values of $R^2$, *MAE*, *RMSE* and *NSE* were shown in Table 1 in the original manuscript. To better illustrate the accuracy of calculated PDD, we added the statistical analysis at 824 meteorological stations. The added sentences are as follows:

Statistical analysis of the calculated monthly PDD against observed PDD at 824 meteorological stations shows the excellent performance at almost all of the stations (Fig. S5).

[Figure]

Figure S5. Statistical criteria of the calculated monthly *PDD* (accumulated positive air temperature) against observed *PDD* at 824 meteorological stations (a, $R^2$, coefficient of determination; b, *MAE/Mean*, *MAE*, mean absolute error; *Mean*, monthly mean *PDD*; c, *RMSE/Mean*, *RMSE*, root mean square error; d, *NSE*, Nash-Sutcliffe efficiency).

• Section 2.2.4; L135-136: Why did the authors only select these five CMIP5 models? Are the climate conditions in these models representative for the full range of possible conditions in terms of climate change (e.g. cold-dry, warm-wet)? What are the consequences for choosing these models for the outcomes of this study? And finally, it would be beneficial to show a few figures on the projected temperature and precipitation changes in China to put the projected changes of snowmelt in a better context.

Response: We select the five CMIP5 models because these models were downscaled to 0.5° and bias-corrected by the Inter-Sectoral Impact Model Intercomparison Project (ISI-MIP). Detailed information on the bias correction and ISI-MIP can be seen in Hempel et al. (2013) and Warszawski et al. (2014). Many scientists have chosen these five models to predict future climate change around the world (e.g. Zarrin et al., 2021; Gusev et al., 2019; Nasonova et al., 2018; Marx

et al., 2018; Vetter et al., 2016). Many studies have used these five models to predict future climate change in China (e.g. Chang et al., 2020; Yin et al., 2018; Chen et al., 2017; Su et al., 2016; Yuan et al., 2016). We have added two figures on the projected precipitation (Fig. S3) and temperature (Fig. S4) changes in China and the added figures are as follows:

[Figure]

Figure S3. Projected future changes in precipitation in China and its three main stable snow cover regions. Historical: 1951-2017; RCP: representative concentration pathway.

[Figure]

Figure S4. Projected future changes in temperature in China and its three main stable snow cover regions. Historical: 1951-2017; RCP: representative concentration pathway.

Chang, J., Wei, Y., Yuan, X., Liao, H., and Yu, B.: The nonlinear impacts of global warming on regional economic production: an empirical analysis from China, Weather Clim. Soc., 12, 759-769, https://doi.org/10.1175/wcas-d-20-0029.1, 2020.

Chen, J., Gao, C., Zeng, X., Xiong, M., Wang, Y., Jing, C., Krysanova, V., Huang, J., Zhao, N., and Su, B.: Assessing changes of river discharge under global warming of 1.5 ℃ and 2 ℃ in the upper reaches of the

Yangtze River Basin: Approach by using multiple- GCMs and hydrological models, Quatern. Int., 453, 63-73, https://doi.org/10.1016/j.quaint.2017.01.017, 2017.

Gusev, E. M., Nasonova, O. N., Kovalev, E. E., and Ayzel, G. V.: Impact of possible climate change on extreme annual runoff from river basins located in different regions of the globe, Water Resour., 46, S126-S136, https://doi.org/10.1134/s0097807819070108, 2019.

Hempel, S., Frieler, K., Warszawski, L., Schewe, J., and Piontek, F.: A trend-preserving bias correction-the ISI-MIP approach, Earth Syst. Dynam., 4, 219-236, https://doi.org/10.5194/esd-4-219-2013, 2013.

Marx, A., Kumar, R., Thober, S., Rakovec, O., Wanders, N., Zink, M., Wood, E. F., Pan, M., Sheffield, J., and Samaniego, L.: Climate change alters low flows in Europe under global warming of 1.5, 2, and 3 °C, Hydrol. Earth Syst. Sci., 22, 1017-1032, https://doi.org/10.5194/hess-22-1017-2018, 2018.

Nasonova, O. N., Gusev, Y. M., Kovalev, E. E., and Ayzel, G. V.: Climate change impact on streamflow in large-scale river basins: projections and their uncertainties sourced from GCMs and RCP scenarios, Proc. IAHS, 379, 139-144, https://doi.org/10.5194/piahs-379-139-2018, 2018.

Su, B., Huang, J., Zeng, X., Gao, C., and Jiang, T.: Impacts of climate change on streamflow in the upper Yangtze River basin, Clim. Change, 141, 533-546, https://doi.org/10.1007/s10584-016-1852-5, 2016.

Vetter, T., Reinhardt, J., Flörke, M., van Griensven, A., Hattermann, F., Huang, S., Koch, H., Pechlivanidis, I. G., Plötner, S., Seidou, O., Su, B., Vervoort, R. W., and Krysanova, V.: Evaluation of sources of uncertainty in projected hydrological changes under climate change in 12 large-scale river basins, Clim. Change, 141, 419-433, 10.1007/s10584-016-1794-y, 2016.

Warszawski, L., Frieler, K., Huber, V., Piontek, F., Serdeczny, O., and Schewe, J.: The Inter-Sectoral Impact Model Intercomparison Project (ISI-MIP): project framework, P. Natl. Acad. Sci. U. S. A., 111, 3228-3232, https://doi.org/10.1073/pnas.1312330110, 2014.

Yin, Y., Ma, D., and Wu, S.: Climate change risk to forests in China associated with warming, Sci. Rep., 8, 493, https://doi.org/10.1038/s41598-017-18798-6, 2018.

Yuan, X., Sun, X., Lall, U., Mi, Z., He, J., and Wei, Y.: China's socioeconomic risk from extreme events in a changing climate: a hierarchical Bayesian model, Clim. Change, 139, 169-181, https://doi.org/10.1007/s10584-016-1749-3, 2016.

Zarrin, A., Dadashi-Roudbari, A., and Hassani, S.: Historical variability and future changes in seasonal extreme temperature over Iran, Theor. Appl. Climatol., 146, 1227-1248, https://doi.org/10.1007/s00704-021-03795-7, 2021.

- Section 3.1.3; L178: How are the PDDs measured? Or where do the data come from? Why not just simply say that when the temperature is above 0 snow starts to melt and the PDD starts to count? What are the temperature thresholds?

Response: PDD in this study is the monthly accumulated positive air temperature. The monthly measured PDD were calculated from the daily measured positive temperature, which were collected from 824 meteorological stations during the 1951-2017 period and were provided by the China Meteorological Administration.

In this study, we use the temperature index model to calculate snowmelt, and we can't simply say that the snow starts to melt when the temperature is above 0 °C. Temperature index models are based on the assumptions that the temporal variability of incoming solar radiation is adequately represented by the variations of air temperature that the snowmelt during a time interval is proportional to positive air temperatures (Hock, 2003; Jost et al., 2012; Lopez et al., 2020). The input driver of the temperature index model is PDD, not mean temperature. For example, when the monthly mean temperature is 0 °C, there must be some days when the temperature is above 0 °C, so there must be snowmelt in that month. It is not reasonable to assume that melting starts when the temperature is above 0 °C.

The temperature thresholds are the parameters to determine the monthly PDD calculation method in Eq. 6. If the monthly mean temperature is lower than $T_1$, the temperature of each day is considered to be negative and the monthly PDD is 0. If the monthly mean temperature is higher than $T_2$, the temperature of each day is considered to be positive, and monthly PDD is the product of the monthly mean temperature and the number of days.

Hock, R.: Temperature index melt modelling in mountain areas, J. Hydrol., 282, 104-115, https://doi.org/10.1016/s0022-1694(03)00257-9, 2003.

Jost, G., Dan Moore, R., Smith, R., and Gluns, D. R.: Distributed temperature-index snowmelt modelling for forested catchments, J. Hydrol., 420-421, 87-101, https://doi.org/10.1016/j.jhydrol.2011.11.045, 2012.

Lopez, M. G., Vis, M. J. P., Jenicek, M., Griessinger, N., and Seibert, J.: Assessing the degree of detail of temperature-based snow routines for runoff modelling in mountainous areas in central Europe, Hydrol. Earth Syst. Sci., 24, 4441-4461, https://doi.org/10.5194/hess-24-4441-2020, 2020.

- Section 3.1.3; Table 1: The results that are presented here are strange. According to the authors the NSE and R2 are both 1.0, which means perfect. I doubt whether this is the case since the MAE and RMSE indicate there are errors. If the MAE and RMSE were 0 I could have imagined that the R2 or NSE are equal to 1 or close to 1, however that is not the case. Therefore, I guess something must have go wrong in the calculations of R2 and NSE and urge the authors to address this point.

Response: The calculations of *NSE* and $R^2$ are correct (Fig. 2b). *NSE* and $R^2$ are both 1.00 in Table 1 because we retained 2 digits after the decimal point by rounding. We have revised those numbers by retaining 4 digits after the decimal point, and the revised Table 1 (Table 2 in the revised manuscript) is as follow:

Table 2. The parameters required for the calculation of the monthly accumulated positive air temperature (*PDD*) and the statistical analysis between the calculated and measured monthly *PDD* in four different climatic zones of China.

| | $T_1$ | $T_2$ | *a* | *b* | *c* | $R^2$ | *MAE* | *RMSE* | *NSE* |
|---|---|---|---|---|---|---|---|---|---|
| MPZ | -7.99 | 5.79 | 0.79 | 15.37 | 56.38 | 0.9966 | 5.87 | 10.85 | 0.9958 |
| TCZ | -10.85 | 9.89 | 0.52 | 15.29 | 85.38 | 0.9975 | 7.96 | 15.32 | 0.9968 |
| TMZ | -10.41 | 9.51 | 0.52 | 15.45 | 81.43 | 0.9973 | 8.45 | 16.56 | 0.9964 |
| SMZ | -4.05 | 8.56 | 0.22 | 23.12 | 49.63 | 0.9993 | 2.67 | 7.63 | 0.9989 |

Note. MPZ, mountain plateau zone; TMZ, temperate monsoon zone; TCZ, temperate continental zone; SMZ, subtropical monsoon zone; $T_1$, $T_2$, a, b and c, parameters in the equation (6); $R^2$, coefficient of determination; *MAE*, mean absolute error (°C); *RMSE*, root mean square error (°C); *NSE*, Nash-Sutcliffe efficiency.

- Section 4.1.1; L260: NSE = 0.2 is considered to be an unsatisfactory evaluation score, so why did the authors decide to use this value as a threshold? Is there a reason why at several stations the outcomes are unsatisfactory? And maybe the authors can elaborate more on their choice to use the NSE as an evaluation criterium for snowfall. It is more usual to use NSE as an

evaluation index for snowmelt runoff or discharge. For the evaluation criteria, please also check Moriasi et al., 2007: https://pubag.nal.usda.gov/download/9298/PDF

Response: Thank you for the recommended literature and we have learned a lot from it. Moriasi et al. (2007) suggested the hydrologic models in watershed simulations could be judged as "satisfactory" if $NSE > 0.50$, and Moriasi et al. (2015) recommended the same $NSE$ value as a threshold. The $NSE$ thresholds recommended by Moriasi et al. were established by descriptive statistics such as mean, median, minimum, and maximum of model data collected from peer-reviewed articles. There are many factors that affect the $NSE$ value, and even a poor fit may lead to a high $NSE$ value. A value of 0.7 may or may not be indicative of a good fit (McCuen, 2006). Moriasi et al. (2015) also noted that these threshold recommendations may be adjusted based on the quality and quantity of available measured data, spatial and temporal scales, and project scope and magnitude. The recommended thresholds are for flow simulations for watershed-scale models, and they may not be suitable for snowfall evaluation. A more common understanding of $NSE$ is that the closer the value to 1, the better the model performance, while $NSE$ is less than 0, the model is unacceptable (e.g. Legates and Mccabe, 1999; Mccuen et al., 2006; Moriasi et al., 2007; Moriasi et al., 2015; Halefom et al., 2017; Gupta and Kling, 2011; Jackson et al., 2019). The 0.2 mentioned in line 260 is not a threshold value for $NSE$, but only a reference to the value of the legend distribution given in Figure 3d. We have changed "296 (64.7%) had $NSE > 0.2$" to "353 (77.2%) had N$SE > 0.0$" in this line.

The main reasons for the unsatisfactory outcomes of several stations are as follows: 1) the snowfall was simulated on a monthly scale by the temperature thresholds; 2) the output data were at the grid scale, whereas the observed snowfall was at the site scale. In addition to $NSE$, other evaluation criteria such as $R^2$, $MAE$ and $RMSE$ were selected to verify the snowfall output by the model. Considering all evaluation criteria and comparing model performances from other studies on identifying precipitation types (see Section "5.1 Model Evaluation"), we believe that it is reasonable to calculate snowfall on a national spatial scale in this study.

Gupta, H. V. and Kling, H.: On typical range, sensitivity, and normalization of Mean Squared Error and Nash-Sutcliffe Efficiency type metrics, Water Resour. Res., 47, W10601, https://doi.org/10.1029/2011wr010962, 2011.

Halefom, A., Sisay, E., Khare, D., Singh, L., and Worku, T.: Hydrological modeling of urban catchment using semi-distributed model, Model. Earth Syst. Environ., 3, 683-692, https://doi.org/10.1007/s40808-017-0327-7, 2017.

Jackson, E. K., Roberts, W., Nelsen, B., Williams, G. P., Nelson, E. J., and Ames, D. P.: Introductory overview: Error metrics for hydrologic modelling – A review of common practices and an open source library to facilitate use and adoption, Environ. Model. Softw., 119, 32-48, https://doi.org/10.1016/j.envsoft.2019.05.001, 2019.

Legates, D. R. and McCabe, G. J.: Evaluating the use of "goodness-of-fit" Measures in hydrologic and hydroclimatic model validation, Water Resour. Res., 35, 233-241, https://doi.org/10.1029/1998wr900018, 1999.

McCuen, R. H., Knight, Z., and Cutter, A. G.: Evaluation of the Nash-Sutcliffe Efficiency Index, J. Hydrol. Eng., 11, 597-602, https://doi.org/10.1061/(ASCE)1084-0699(2006)11:6(597), 2006.

Moriasi, D. N., Gitau, M. W., Pai, N., and Daggupati, P.: Hydrologic and water quality models: performance measures and evaluation criteria, T. ASABE, 58, 1763-1785, https://doi.org/10.13031/trans.58.10715, 2015.

Moriasi, D. N., Arnold, J. G., Van Liew, M. W., Bingner, R. L., Harmel, R. D., and Veith, T. L.: Model evaluation guidelines for systematic quantification of accuracy in watershed simulations, T. ASABE, 50, 885-900, https://doi.org/10.13031/2013.23153, 2007.

• Section 4.1.2; L269 / Section 4.1.3; L282-283 / Section 4.1.4; L294: The simulated output reported by the model of the authors is on monthly basis, based on monthly mean values. Then it would be more appropriate and representative to compare the simulated snow depth with the monthly means of the observed snow depth instead of selecting the last day of each month. That is not representative. Besides, most likely comparing the simulated values with the observed monthly means will improve the outcomes of the authors presented later in the manuscript.

Response: Thank you for your suggestion. The model was performed on a monthly scale, whereas the snow depth was observed at meteorological stations daily. We calculated the statistical criteria of the simulated snow depth against the monthly means of the observed snow depth, and found that there was no significant change compared with the statistical criteria of the simulated snow depth against the last day's observations. In our opinion, either using the monthly

means of observed snow depths or using the last day's observations to validate the snow depths output by the model is not representative, but that using the last day's observations is more appropriate. The input snowfall data of the model is monthly snowfall, and the input temperature data is monthly PDD. The model calculates the snow depth from the amount of snow remaining after melting. When there is no snow on the last day, it means that all the snow has melted in that month, the snow accumulation is 0, and the snow depth is 0. However, it is very likely that there are a few days in the month that are covered with snow, and the monthly mean snow depth will be greater than 0. It is not appropriate to use the monthly mean snow depth to validate the snow depths output by the model.

**Specific Comments**

• Abstract; L11: 1 km corresponds to 30 arcseconds or 0.5 arcminute. The latter has also been used in the dataset description for the high-resolution temperature and precipitation datasets by Peng et al., 2019.

Response: Thank you for pointing this out. It was our mistake. The units in L11 and elsewhere in the manuscript have been corrected in the revised version.

• Abstract; L16: I recommend the authors to use $m^3$/year instead of $m^3$ to make clear that the authors talk about the mean annual snowmelt instead of the total snowmelt volumes within a period.

Response: Thank you for your comment. We have revised "$m^3$" to "$m^3$ $year^{-1}$" in the revised manuscript.

• Abstract; L20: From the abstract I cannot derive what "third level" stands for. Also, from the manuscript it is not clear what authors mean with "third level". Therefore, I would like to suggest using "subbasin" instead of "third level".

Response: Thank you for your suggestion. In our opinion, using "third-level basins" may be more appropriate than "subbasin" in this manuscript. The hydrologists in China have graded China's

basins into first-level, second-level and third-level basins. We have added a figure to the Supplement, which we hope will help reviewer and readers understand the third-level basins in the manuscript. The added figure is as follows:

[Figure]

Figure S1. The first-level (a) and second-level (b) basins in China.

• Introduction; L39: What do the authors mean with slower snowmelt rates? Please rephrase to increase the clarity of the sentence.

Response: According to study from Musselman et al. (2017), warming climates could lead to slower snowmelt. This is because the less snow accumulates on the ground, the slower the snow melting rate in winter, rather than lasting until summer, when the high temperature will cause the snow to melt quickly.

Musselman, K. N., Clark, M. P., Liu, C., Ikeda, K., and Rasmussen, R.: Slower snowmelt in a warmer world, Nat. Clim. Change, 7, 214-219, https://doi.org/10.1038/nclimate3225, 2017.

• Introduction; L63: Do the authors mean the crop sowing season, crop harvesting season, or the entire crop season with "crop planting season"? I would like to recommend using one of the above-mentioned terms.

Response: We will use "crop sowing season" in the revised manuscript.

• Introduction; L65: Recently a new paper has been published on the Asian water towers by Immerzeel et al., 2020 (https://doi.org/10.1038/s41586-019-1822-y).

Response: Thank you for recommending paper, we have added it to the references.

• Introduction; L76: Here and throughout the manuscript, please check and see my point at Abstract; L11.

Response: We have revised the spatial units throughout the manuscript.

• 2 Data Collection: Since a significant number of datasets is used in this study and the reader can get lost in all the numbers and details, I would like to suggest adding a table that includes the information of the datasets used, such as the variables (e.g., snow density), the source, the measurement periods, and the number of stations, etc.

Response: Thank you for your suggestion. We have added a table in the revised manuscript, and the added table is as follows:

Table 1. Data used in the snowmelt model.

| Variable | Period | Timescale | Spatial resolution | Number of meteorological stations | Source |
|---|---|---|---|---|---|
| Air temperatures* | 1951-2017 | Monthly | 0.5′ | - | Zenodo |
| Precipitation | 1951-2017 | Monthly | 0.5′ | - | Zenodo |
| Snow cover extent | 1979-2018 | Daily | 0.25 ° | - | NTPDC |
| Snow water equivalent | 1980-2020 | Daily | 25 km | - | NCDDC |
| Air temperature | 1951-2017 | Daily | - | 824 | CMA |
| Snowfall | 1961-1979 | Daily | - | 475 | CMA |
| Snow depth | 1951-2009 | Daily | - | 557 | CMA |
| Snow density | 1999-2008 | 5 days | - | 417 | CMA |
| Threshold temperatures | - | Monthly | - | 485 | Han et al. (2010) |

Note. *Minimum, maximum, and mean temperatures; CMA, China Meteorological Administration; NTPDC, National Tibetan Plateau Data Center; NCDDC, National Cryosphere Desert Data Center.

• Section 2.2.1; L100: NetCDF is a data format, data cannot be obtained from NetCDF but are supplied in a NetCDF format. As I understand via the links the data were obtained from Peng et al., 2019, so please refer to them. Also, the link to the precipitation datasets did not work.

Response: Thank you for pointing this out. We have revised the sentence as follows: Data with a high spatial resolution of 0.5′ (approximately 1 km), including the monthly minimum, maximum,

and mean temperatures ($T_{min}$, $T_{max}$, and $T_a$, respectively) and precipitation, were obtained from Zenodo in the Network Common Data Form (https://doi.org/10.5281/zenodo.3114194 for precipitation, Peng, 2019a; https://doi.org/10.5281/zenodo.3185722 for air temperatures, Peng, 2019b).

- Section 2.2; L102: CRU timeseries are supplied on a 0.5 degree grid, not at a 30 arcseconds grids. Please correct.

Response: Corrected.

- Section 2.2; L103: I miss a reference to the WorldClim dataset. Also, more information on this dataset would be beneficial. For instance, what is spatial resolution of this dataset?

Response: According to the paper by Peng et al. (2019), they used four high-resolution datasets at spatial resolutions of 10′, 5′, 2.5′, and 0.5′ from WorldClim v2.0 (http://worldclim.org, last access: 25 April 2019) (Fick and Hijmans, 2017). Data published by Peng used the WorldClim datasets at spatial resolutions of 0.5′. Detailed information on the WorldClim dataset can be seen in Peng et al. (2019).

Fick, S. E. and Hijmans, R. J.: WorldClim 2: new 1km spatial resolution climate surfaces for global land areas, Int. J. Climatol., 37, 4302-4315, https://doi.org/10.1002/joc.5086, 2017.

Peng, S., Ding, Y., Liu, W., and Li, Z.: 1 km monthly temperature and precipitation dataset for China from 1901 to 2017, Earth Syst. Sci. Data, 11, 1931-1946, https://doi.org/10.5194/essd-11-1931-2019, 2019.

- Section 2.2; L102-104: I would split up the sentence into two parts to increase the readability.

Response: We have revised the sentence as follows: The data were spatially downscaled from the 0.5 °Climatic Research Unit time series data with the 0.5′ climatology dataset of WorldClim using delta spatial downscaling. Peng et al. (2019) evaluated this dataset using observations collected from 1951 to 2016 by 496 meteorological stations across China.

- Section 2.2; L104: The observations are collected from 1951 to 2016, but is the latter year not supposed to be 2017?

Response: The data produced by Peng et al. (2019) covers the 1901 to 2017 period, and Peng et al. evaluated the dataset with meteorological stations observations from 1951 to 2016. The data period we used in the study is 1951-2017.

- Section 2.2.2; L114: Is this about the critical temperature defining whether precipitation falls as snow or rain? That there are different threshold makes sense, but it would be good to map this as well for the readers convenience.

Response: Yes, this is about the critical temperature. We have added a figure to the Supplement, and the added figure is as follows:

[Figure]

Figure S2. The threshold temperature (℃) for snowfall (a) and rainfall (b) at 485 meteorological stations in China.

- Section 2.2.2; L116: I think in this context there are more recent data available as well. For example, see Jennings et al. (2018; https://doi.org/10.1038/s41467-018-03629-7) on the spatial variation of the rain-snow temperature threshold across the Northern Hemisphere.

Response: Thank you for providing the literature. The study area of this manuscript is China, and we have collected the threshold temperature parameters based on the observations from the meteorological stations. In this manuscript, we prefer to use the data we collected. In future work, we may use the recommended data.

- Section 2.2.2; L117: Were the threshold temperatures spatially interpolated? If so, via IDW or another method?

Response: The IDW method. We have revised the sentence as follows: The threshold temperatures of each calculated cell were interpolated using the parameters from the meteorological stations via inverse distance weighting (IDW) method.

• Section 2.2.3; L120: Are there no other observations of snowfall data from a later period?

Response: Unfortunately, we did not collect snowfall data after 1980. According to the information we have, since 1980, all meteorological stations only provide data on total precipitation, and no longer provide data on precipitation types.

• Section 2.2.3; L125: Please validate the spatial resolution of the dataset.

Response: Checked.

• Section 2.2.3; L129: How is snow cover derived from snow depth?

Response: If the snow depth in the grid is greater than 0, it means that the grid is covered with snow. If the snow depth in the grid is 0, it means there is no snow in the grid.

• Section 3.1; L151: Does the PDD represent the mean monthly accumulated positive air temperature?

Response: PDD represent the monthly accumulated positive air temperature.

• Section 3.1.1; L159: The threshold temperatures are described, but not presented as a main result, which makes it difficult to imagine what the numbers are. Is it possible for the authors to present those numbers by means of a table or figure, either in the manuscript or the supplementary information?

Response: We have added a figure to the Supplement, which can be found in the previous response (Figure S2 ).

• Section 3.1.2; L160-168: The authors use a method to calculate DDF based on the density of snow and water. The density of snow is based on observations. The question is, however, how do

the authors calculate the future DDF. Do they authors assume the snow density to be constant over time?

Response: In this study, we calculated the DDF based on the snow density, and assumed that the monthly DDF is constant.

• Section 3.1.2; L164: The snow density is variable since it is observed, but the density of water is constant, so please note it here for the readers.

Response: We have revised the sentence as follows: where $\rho_s$ and $\rho_w$ are the density of snow and water (g cm$^{-3}$), respectively. The observed snow density is variable and the density of water is constant.

• Section 3.1.4; L200: Do the authors have a reference to this sublimation method? Or is this a method the authors developed their own? If so, is this method considered to be valid?

Response: Snow sublimation is difficult to quantify by measurement or modelling. Accurate calculation requires a large amount of high-quality meteorological data, which is difficult to achieve in large spatial scale. Therefore, we developed a simple method to estimate the monthly snow sublimation in this study. This method can be considered valid because: 1) the empirical parameters were set according to studies reporting the ratio of snow sublimation to snowfall in China and surrounding areas (Table S1 and Fig. S1 in the original manuscript); 2) the outputs by model were validated by snowfall, snow depth, snow cover extent and snow water equivalent.

• Section 3.2; L216-L218: I don't consider this as a good argument, since many hydrological models use the same methods, the authors also use. The models require temperature and precipitation data + several GIS data as an input, which are mostly available.

Response: The hydrological models mentioned in line 216 mainly refer to watershed hydrological models with runoff generation. Those models can separately simulate the runoff from rainfall and snowfall, and further calculate the snowmelt runoff ratio. The model in this manuscript did not calculate the runoff and used a simple method to estimate the snowmelt runoff ratio. To make it clearer, we have revised the sentence as follows: The watershed hydrological models with runoff processes are generally effective for determining the snowmelt

runoff ratio (Immerzeel et al., 2010; Jenicek and Ledvinka, 2020), however, they are difficult to implement over large regions due to data limitations and large computational requirements.

• Section 3.3; L233: Does n refer to the number of samples within a dataset?

Response: Yes.

• Section 4.1.1; L254-255: Please indicate this info in the data section.

Response: We have moved this sentence to the data section.

• Section 4.1.2; L268: How is the snow depth calculated from the snow accumulation and snow density?

Response: The snow accumulation calculated by the model is in mm. The snow depth can be calculated as follows:

$$snowdepth(mm) = snow\ accumulation \cdot water\ density/snow\ density$$

• Section 4.1.2; L270-271: Please indicate this info in the data section.

Response: We have moved this sentence to the data section.

• Section 4.1.4; L295-296: An alternative solution for dealing with scale differences is to conservatively remap the outcomes to the 25km grid to compare simulated values with observed values. Did the authors consider using a conservate remapping technique to compare the simulated values with the observed values?

Response: Thank you for your suggestion. We did not use a conservate remapping technique because the validation method we chose was valid.

• Section 4.1.4; L298: Here as well as in the other sections where validation outcomes are presented it would be beneficial to add the observed and simulated values as well.

Response: We have added the observed and simulated values in here and other sections. In this section, the added sentence is as follows: The mean monthly snow accumulation in China output

by the model from 1980 to 2017 was $2.55 \times 10^{10}$ m$^3$, while the mean snow water equivalent derived from passive microwave remote on the last day of each month was $1.47 \times 10^{10}$ m$^3$.

• Section 4.1.4; 299-306: I consider this argumentation as insufficient. Firstly, that the microwave remote sensing data have a spatial resolution of 25 km does not mean that relatively small glaciers are not recorded. At least, I think the authors should be able to substantiate why this is the case (e.g. by means of scientific literature) and whether it is a common problem for microwave remote sensing data. Secondly, since the authors decided to use the last day as a monthly observational representative the chance is very likely that the snow water equivalent on that particular day is 0, whereas the mean monthly snow water equivalent would have been different. This might explain as well why there is a discrepancy between the observed data and the simulated data. To my opinion, the authors need to elaborate more on this point.

Response: We believe that the resolution of microwave remote sensing is the main reason why relatively small glaciers have not been recorded. Fig. R1a shows the spatial distribution of the mean snow water equivalent in August 2017 in China, and Fig. R1b shows the spatial distribution of glaciers in China. The mean snow water equivalent of 0 in August means that the snow water equivalent of every day in August is 0. As can be clearly seen in the figure, most of the glacial regions in the Tibetan Plateau have a mean snow water equivalent of 0 in August, indicating that the data retrieved by the microwave remote sensing has not recorded the snow water equivalent of those glaciers. We have consulted Lingmei Jiang, the producer of snow water equivalent data, and Tao Che, the producer of snow depth data, both of whom are experts in microwave remote sensing, and they both agree with us. Regarding the second point of this comment, since the model determines the monthly snow accumulation based on the amount of snow remaining after melting, we prefer to use the snow water equivalent on the last day of the month to validate the snow accumulation.

[Figure]

Figure R1. Spatial distribution of the mean snow water equivalent (SWE) in August 2017 in China (a) and spatial distribution of glaciers in China (b).

- Section 4.2.1; L333: Here and throughout the manuscript, many readers have most likely no idea where most of the geographical areas are located. Therefore, I would like to recommend adding those locations to a map or to give an indication where these areas are located, such as West China or via some lat-lon coordinate.

Response: Thank you for your comment. In the Supplement, we have added a figure about the geographical distribution in the Tibetan Plateau (Fig. S8 in the revised manuscript), and we hope this figure is helpful to readers. The added figure is as follows:

[Figure]

Figure S8. The location of the Qaidam Basin, Qiangtang Plateau and Mountains on the Tibetan Plateau.

- Section 4.2.2; L358-359: I think this is a result of global warming that combined, causes a larger fraction of rainfall relative to snowfall during summertime and an earlier onset of snowmelt during spring.

Response: We agree.

- Section 4.2.2; L359: Can the authors support there finding by indicating the increase in temperature that is measured in the regions?

Response: In the context of global warming, the increase in temperature in the vast majority of the world has already been confirmed. According to temperature data provided by Peng et al. (2019), the temperature generally increased throughout China from 1951-2017, although not all grids were significant. Fig. 15 in the original manuscript can provide the information of temperature change in the areas with significant change of snowmelt.

- Section 4.3.1; L362: What are third-level basins? Are these basins of a third order? How are these defined? Is not better to mention them as subbasins?

Response: We have added a figure to the Supplement, which we hope will help reviewer and readers understand the third-level basins in the manuscript. The added figure can be found in the previous response (Fig. S1).

- Section 4.3.1; L366: Due to heavy rainfall and low snowmelt?

Response: Although snowfall accounts for a small proportion of precipitation, due to the heavy precipitation, the amount of snowmelt in some areas of South China is not low (Figure 7a in the original manuscript). We believe the low snowmelt runoff ratios in basins in South China are due to the heavy rainfall.

- Section 4.3.2; Line 386: I guess it should be Section 4.3.2 instead of 4.2.2

Response: Thank you for pointing this out. We have revised it.

- Section 4.3.2; L391: Is it Southeastern China or the southeastern part of the Tibetan Plateau?

Response: It's the southeastern part of the Tibetan Plateau. We have revised it.

- Section 4.3.2; L408: Is it significant or not significant. Considering the way how the sentence is phrased I would say non-significant. Please check and rephrase if necessary.

Response: It's not significant. We have revised the sentence as follows: ……the Mann-Kendall test showed that these monthly increasing trends in third-level basins were rarely significant.

• Section 5.1; L454-465: What are the big differences and why are the results of this paper different/better/worse than the results in the other studies?

Response: The results of this study are not significantly different from those of other studies, being slightly better in some places and slightly worse in others. The reasons for these differences are: 1) the input precipitation and temperature data are different, and 2) the threshold temperatures used to determine precipitation types are different. For example, Li et al. (2020) used the datasets on daily temperature and precipitation from the Asian Precipitation-Highly Resolved Observational Data Integration Towards Evaluation (APHRODITE) and the National Oceanic and Atmospheric Administration (NOAA) Climate Prediction Center (CPC), and they used temperature or wet-bulb temperature and relative humidity to define thresholds.

• Section 5.1; L468: The authors indicate their self that they should have used another method. What can the authors do to improve their results?

Response: As we mentioned earlier, the model was performed on a monthly scale, whereas the snow depth was observed at meteorological stations daily. It is difficult to find a suitable method to validate the simulated monthly-scale snow depth by the observed daily-scale snow depth. Based on the model principles and our understanding, using the last day's observations to validate the simulated monthly snow depth is the best option.

• Section 5.1; L466-479: What could be a potential reason for the under-performing snow depth? Could undercatch be a reason?

Response: We have stated three possible reasons in the manuscript. The reviewer mentioned that undercatch may be a reason. Does the reviewer mean gauge undercatch? According to our understanding, gauge undercatch is more about snowfall, not snow depth. We cannot answer with certainty whether undercatch is a reason, but it could be.

- Section 5.1; L484-487: I think point 2 and 3 are related to each other. Due to the complex terrain and harsh climate conditions, the number of stations is limited, and the distribution is sparse.

Response: Points 2 and 3 are to explain why the performance of the snow water equivalent in the Tibetan Plateau was poorer than those in other regions. Point 2 and Point 3 are indeed related, but they are two different reasons. The second reason is that the reliability of the model parameters might be worse in the Tibetan Plateau than in other regions. The third reason is that the snow water equivalent data used for verification have large uncertainties in the Tibetan Plateau.

- Section 5.2; L502: Is there a specific reason for the higher correlation during wintertime? Is the variability of winter precipitation larger than the variability of spring precipitation?

Response: This result comes from Tan et al. (2019). We think the reason is that snowfall mainly occurs in winter. Tan et al. (2019) only analyzed the change trend of precipitation, so we cannot answer the question about the variability of precipitation.

Tan, X., Wu, Z., Mu, X., Gao, P., Zhao, G., Sun, W., and Gu, C.: Spatiotemporal changes in snow cover over China during 1960-2013, Atmos. Res., 218, 183-194, https://doi.org/10.1016/j.atmosres.2018.11.018, 2019.

- Section 5.2; L523: Are the annual precipitation increases related to the changes in monsoon rainfalls or to changes in the westerly driven precipitation?

Response: According to the review paper by Kuang and Jiao (2016), there may be different mechanisms of precipitation changes in the Tibetan Plateau, and changes in monsoonal circulation may be an important reason.

Kuang, X. and Jiao, J. J.: Review on climate change on the Tibetan Plateau during the last half century, J. Geophys. Res. Atmos., 121, 3979-4007, https://doi.org/10.1002/2015jd024728, 2016.

- Section 5.2; L524: High elevations are more sensitive to warming --> Elevation-dependent warming. I am not sure what the authors mean with "tolerate", but I guess it should be opposite.

Response: We agree that high elevations are more sensitive to warming, and we think this is not in conflict with the fact that high elevations can tolerate more warming. For example, suppose there are two sites with a temperature of -5 ℃ at the high elevation site and -1 ℃ at the low elevation sites. The high elevation site is more sensitive to warming. The temperature at the high elevation site will increase by 4 ℃ and the temperature at the low elevation site will increase by 2 ℃. With a warming climate, the temperature at the high elevation site will be -1 ℃, the type of precipitation will still be snowfall and the snow will still not melt. However, at the low elevation site, the temperature will reach 1 ℃, the snowfall will turn to rainfall and the snow will melt.

• Section 5.3.2; L550-551: What are the authors trying to say with the high levels of precipitation in this region? As far I know most of the precipitation falls here during the East Asian summer monsoon, which is in summer, whereas the snowmelt season is in spring. That means the high levels of precipitation cannot be a good explanation for the snowmelt runoff ratios in this region.

Response: Thank you for pointing this out. What we wanted to say is that there is more precipitation in this region than in Northern Xinjiang. To avoid misunderstanding, we have revised this sentence as follows: The snowmelt runoff ratios in the basins in Northeast China are generally about 10%, and the importance of snowmelt as a water resource in this region may be less than in Northern Xinjiang.

• Section 5.3.3; L565: I think the authors cannot simply state that in some months all the runoff is contributed by snowmelt, since the authors have not considered the contribution of glacier meltwater to total runoff. For this reason, the authors need to elaborate more on this point, or I recommend them to rephrase the sentence.

Response: We have revised this sentence as follows: Snowmelt contributes runoff every month, and in some months, there is no rainfall to generate runoff.

**Technical Comments**

- Introduction; L38: "contribute" instead of "contributes"

Response: Changed.

- Introduction; L39: here and throughout the manuscript: "an earlier onset of snowmelt" instead of "earlier snowmelt times"

Response: Changed.

- Introduction; L44: "operations" instead of "operation"

Response: Changed.

- Introduction; L45: here and throughout the manuscript: "physically-based" instead of "physically based"

Response: Changed.

- Introduction; L46: I would use "simplified" instead of "simpler"

Response: Changed.

- Introduction; L52: "variations of air temperature and that the snowmelt" instead of "variations of air temperature that the snowmelt"

Response: Changed.

- Introduction; L61: "snowmelt" instead of "snow meltwater"

Response: Changed.

- Introduction; L64: "Snowmelt is also an important hydrological process on the Tibetan Plateau" instead of "Snow melting is aslo an important hydrological process in the Tibetan Plateau"

Response: Changed.

- Introduction; L65: "and is considered as the Asian water towers" instead of "and considered as the asian water towers".

Response: Changed.

- Introduction; L65: "Further, snowmelt is an important" instead of "Snowmelt is also an important"

Response: Changed.

- Introduction; L77: "considers" instead of "considered"

Response: Changed.

- Introduction; L78: "are" instead of "were"

Response: Changed.

- Introduction; L79: "China as well as in its three main" instead of "China and in its three main"

Response: Changed.

- Introduction; L80: "during 1951-2017" instead of "in the 1951-2017 period"

Response: Changed.

- Section 2.2; L102: "temperature" instead of "temperatures"

Response: Changed.

- Section 2.2.3; L119: I recommend merging the two sentences here into one. For example: "The observational snowfall (snow depth) data used to validate the model were collected from 475 (557) meteorological stations in China (Fig.1d (1a)) during the 1961-1979 (1951-2009) period and were provided by the China Meteorological Administration"

Response: Changed.

- Section 2.2.3; L123: "from" instead of "from by"

Response: Changed.

- Section 2.2.4; L139-142: This sentence is long and, particularly, the first part of the sentence needs to be rephrased to increase its readability and clarity.

Response: We have revised the sentence as follows: In this study, we use the delta downscaling method to determine the monthly future meteorological data (2006-2099) based on the high-spatial-resolution temperature and precipitation dataset and the simulations of the five CIMP5 models during the historical period (1951-2005).

- Section 3.1.2; L164: "mm °C$^{-1}$ day$^{-1}$" instead of "cm °C$^{-1}$ day$^{-1}$"

Response: In fact, the unit of DDF in Eq. 4 is indeed cm °C$^{-1}$ day$^{-1}$, not mm °C$^{-1}$ day$^{-1}$. In order to be consistent with the unit of DDF in Eq. 1, we have changed Eq. 4 and Eq.5 as follows:

$$DDF = 11(\rho_s/\rho_w) \tag{4}$$
$$DDF = 10.4(\rho_s/\rho_w) - 0.7 \tag{5}$$

- Section 3.1.4; L193: "methods" instead of "method"

Response: Changed.

- Section 3.3; L240: "where the β sign reflects whether a trend is negative or positive" instead of "where β sign reflects data trend reflection"

Response: Changed.

- Section 4.1.1; L259: I guess the authors made a mistake here. I guess it should be 57.5%.

Response: Thank you for pointing this out. We have revised it.

- Section 4.1.1; L260: "accounting for 60.0%" can be removed.

Response: Removed.

- Section 4.2.1; L316: repeated word, i.e., "the area with with"

Response: We have deleted a "with".

- Section 4.2.1; L322: "Plateau becomes the main region of snowmelt until May. In summer, there is no snowfall in most of China and snowmelt" instead of "Plateau became the main region of snowmelt until May. In summer, there was no snowfall in most of China and snowmelt"

Response: Changed.

- Section 4.2.2; L346: "In Southeast China" instead of "Southeast China"

Response: Changed.

- Section 4.2.2; L355: "but" instead of "while"

Response: Changed.

- Section 4.2.2; L356: "might imply" instead of "implied"

Response: Changed.

- Section 4.2.2; L357: "at the Tibetan" instead of "in the Tibetan"

Response: Changed.

- Section 4.4.1; L415: "are shown" instead of "were shown"

Response: Changed.

- Section 4.4.1; L425: "17.1% (24.7%, 42.8%), respectively, compared to the historical period." Instead of "17.1% (24.7%, 42.8%) compared to the historical period."

Response: In the revised manuscript, we use 1980-2009 as the historical "reference period", and 2010-2039, 2040-2069 and 2070-2099 as near-future, mid-future, and far-future periods, respectively. The revised sentence is as follows: …11.7% (24.8%, 36.5%), respectively, compared to the reference period.

• Section 4.4.2; L445-449: Very long sentence. I think it is better to split up the sentence in two parts.

Response: We have revised this sentence as follows: Under RCP8.5, the projected mean annual snowmelt runoff ratios in the far-future were lower than those in the reference period in all basins except the three basins near the Taklimakan Desert and one basin in central Inner Mongolia. Under RCP8.5, relative to the reference period, the snowmelt runoff ratios in the Tibetan Plateau and Tianshan Mountains were projected to decrease by more than 5% in most basins and by more than 10% in a few basins in the far-future.

• Section 5.2; L498: "grid cells" instead of "grids"

Response: Changed.

• Section 5.2; L513: "freezing point" instead of "freezing"

Response: Changed.

• Section 5.3.1; L546: "and therefore introduce agricultural risks Northern Xinjiang." Instead of "and changes in the snowmelt amount and timing may bring agricultural risks in the Northern Xinjiang."

Response: Changed.

• Section 5.3.3; L565: "contributes to runoff" instead of "contributes runoff"

Response: Changed.

---

## Author Comment (AC2)

Responses to the comments from Anonymous Reviewer #2

As the key source of freshwater, snowmelt water resource in China has never been quantified on a national scale. This study used a simple temperature index model to calculate the snowmelt in China. The model is shown to perform acceptably well in China when the outputs were validated by snowfall, snow depth, snow cover extent and snow water equivalent. The results of this paper have important significance for understanding the distribution and variation of snowmelt in China. The simple model in this paper is interesting and the description of the model is comprehensive, and I think the method provides useful guidance for calculating snowmelt water resources outside China. In general, I think this work is valuable and of interest to the great community, and the manuscript is well written and worthy of publication in HESS. My comments are listed as follows:

Response: We would like to sincerely thank you for the valuable comments and suggestions to improve our manuscript. We have revised the manuscript according to your helpful comments. The responses following a point-by-point to the comments are provided below.

1. Line 11: the spatial resolution, 0.5 seconds? Shouldn't it be 0.5 minutes? Please check it throughout the manuscript.

Response: Thank you for pointing this out. It was our mistake. The units in Line 11 and elsewhere in the manuscript have been corrected in the revised version.

2. Line 16: change the unit "$m^3$" to "$m^3$ $year^{-1}$", and revise it throughout the manuscript.

Response: Thank you for your comment. We will use $m^3$ $year^{-1}$ instead of $m^3$ in the revised manuscript.

3. Line 19: Should it be "snowmelt water resource"? or "snowmelt time", "snowmelt rate"? "snowmelt" isn't clear, I think.

Response: Snowmelt is water produced when snow melts. In this sentence, we think the appropriate word is snowmelt.

4. Line 38: "contributes" to "contribute".

Response: Thanks, we have revised it.

5. Line 61: "snow meltwater" to "snowmelt".

Response: Changed.

6. Line 64: change "aslo" to "also".

Response: Changed.

7. Line 93: Figure 1a showed the mean snow depth from 557 meteorological stations in China. The mean snow depth is little significant, and it is better expressed by accumulated snow depth or the maximum snow depth. Figure 1c. Please use 3 as superscript in $cm^3$.

Response: Thank you for this comment. We have revised the the mean snow depth to the maximum snow depth in Figure 1a, and we have revised the unit in the Figure 1c. The revised figure is as follows:

[Figure]

Figure 1. The three main stable snow cover regions and the mean snow depth in China (1951-2009) (a); China's five climatic zones (MPZ, mountain plateau zone; TMZ, temperate monsoon zone; TCZ, temperate continental zone; SMZ, subtropical monsoon zone) and mean annual air temperature (1951-2017) (b); the snow cover classification and mean monthly snow density in China (1999-2008) (c); the third-level basins and mean annual snowfall in China (1961-1979) (d).

8.  Line 95: "mean snow density in China" is monthly or yearly? It should be introduced clearly.

Response: It is monthly. We have revised it.

9.  Line 101: The data link "https://doi.org/10.5281/zenodo.3114194forprecipitation" can not be connected.

Response: The data link is https://doi.org/10.5281/zenodo.3114194.

10. Line 115: The threshold temperature in China in this study should be shown in this manuscript by figure or table, or partly shown, I suggest.

Response: Thank you for your suggestion. We have added a figure to the Supplement, and the added figure is as follows:

[Figure]

Figure S2. The threshold temperature (°C) for snowfall (a) and rainfall (b) at 485 meteorological stations in China.

11. Line 117: What is the interpolated method?

Response: The interpolated method is IDW method. We have revised the sentence as follows: The threshold temperatures of each calculated cell were interpolated using the parameters from the meteorological stations via inverse distance weighting (IDW) method.

12. Line 124: Please delete "by".

Response: Deleted.

13. Line 129: The original dataset is snow depth, and the authors used this dataset to verify the snow cover extent. Please explain that.

Response: It is difficult to observe the snow cover extent on the ground, and space remote sensing constitutes an efficient observation technique. The snow cover extent can be generated from the snow depth dataset, and we use this dataset to validate snow cover extent output by the model.

14. Line 135: downscaling was finished by yourself or others? It should be elaborated in detail.

Response: The downscaling was done by ourselves. The L139-142 in the original manuscript introduces the downscaling method, and we have revised the sentence as follows: In this study, we use the delta downscaling method to determine the monthly future meteorological data (2006-2099) based on the high-spatial-resolution temperature and precipitation dataset and the simulations of the five CIMP5 models during the historical period (1951-2005).

15. Line 151: The unit of DDF is mm $°C^{-1}$ $day^{-1}$, but the unit in equation (4) (Line 164) is cm $°C^{-1}$ $day^{-1}$. Please check.

Response: Both units are correct. To make the two units consistent, we changed Eq. 4 and Eq.5 as follows:

$$DDF = 11(\rho_s/\rho_w) \tag{4}$$
$$DDF = 10.4(\rho_s/\rho_w) - 0.7 \tag{5}$$

16. Line 182: *NSE* equals one is not understandable, *RMSE* is not small for the temperature value. And the table does not reflect the monthly difference, but it has said monthly parameter in the title.

Response: *NSE* equals one in Table 1 because we retained 2 digits after the decimal point by rounding. *RMSE* in the Table 1 are not the statistical analysis between the calculated and measured monthly mean temperature, but the statistical analysis between the calculated and measured monthly accumulated positive air temperature (PDD). The values of *RMSE* may be not small for the mean temperature value, but they are very small for the monthly PDD (Fig. 2b). The parameters in Table 1 have no seasonal differences and they are used to calculate monthly PDD.

We have revised the numbers by retaining 4 digits after the decimal point, and the revised Table 1 (Table 2 in the revised manuscript) is as follow:

Table 2. The parameters required for the calculation of the monthly accumulated positive air temperature (*PDD*) and the statistical analysis between the calculated and measured monthly *PDD* in four different climatic zones of China.

|  | $T_1$ | $T_2$ | *a* | *b* | *c* | $R^2$ | *MAE* | *RMSE* | *NSE* |
|------|--------|-------|------|-------|-------|--------|------|-------|--------|
| MPZ | -7.99 | 5.79 | 0.79 | 15.37 | 56.38 | 0.9966 | 5.87 | 10.85 | 0.9958 |
| TCZ | -10.85 | 9.89 | 0.52 | 15.29 | 85.38 | 0.9975 | 7.96 | 15.32 | 0.9968 |
| TMZ | -10.41 | 9.51 | 0.52 | 15.45 | 81.43 | 0.9973 | 8.45 | 16.56 | 0.9964 |
| SMZ | -4.05 | 8.56 | 0.22 | 23.12 | 49.63 | 0.9993 | 2.67 | 7.63 | 0.9989 |

Note. MPZ, mountain plateau zone; TMZ, temperate monsoon zone; TCZ, temperate continental zone; SMZ, subtropical monsoon zone; $T_1$, $T_2$, a, b and c, parameters in the equation (6); $R^2$, coefficient of determination; *MAE*, mean absolute error (°C); *RMSE*, root mean square error (°C); *NSE*, Nash-Sutcliffe efficiency.

17. Line 259: 263 (5.7%). 5.7%? please check.

Response: Thank you for pointing this out. It's 57.5%. We have revised it.

18. Lines 268-276: The number of meteorological stations used for snow depth verification was 264, far fewer than the 557 stations in Figure 1a. Why choose so few meteorological stations for verification?

Response: We have collected observational snow depth data from 557 meteorological stations, however, at some meteorological stations, short snow cover duration and shallow snow result in very little data on snow depth. When performing snow depth validation, meteorological stations with little data were not selected, and data from 264 stations were finally selected for validation. According to Fig. 1a and Fig. 4, most of the meteorological stations that have not participated in the verification are in central and southern China where the climate is relatively warm.

Meteorological stations located in the three main stable snow cover regions are rarely excluded from verification.

19. Line 316: Delete the extra word "with".

Response: Deleted.

20. Line 327: It is better to cite Fig.7b before Fig.8 in the manuscript.

Response: In line 315 in the original manuscript, we have cited Fig. 7a for the first time. In line 319, we have cited Fig. 8 for the first time. Fig. 7 is cited before Fig. 8, and we do not think it is necessary to cite every sub-figure in Fig. 7 before Fig. 8.

21. Line 387: "4.2.2" to "4.3.2".

Response: Changed.

22. Line 415: "were shown" to "are shown".

Response: Changed.

23. Line 340: How is the density used when using the model from point to surface in China? Do you use the average value for the five typical regions from different sites or other methods? It should be explained in detail.

Response: Sorry, we can't understand this comment. The line 340 in the manuscript is the caption of Fig.8, and we do not see any relationship between the caption and this comment. And there is no "five typical regions" in our manuscript. If the reviewer is asking how we use snow density, we can answer that. We use the snow density to calculate the DDF values at the meteorological stations separately and then interpolate those to each calculated cell via IDW.

24. Lines 409-451: 4.4 Future changes of snowmelt under different climate scenarios. The historical period is from 1951 to 2017, while the future periods are different decades, namely the 2030s, 2050s, and 2090s. The comparison periods are inconsistent. I suggest changing

"historical period" to "reference period", and setting the period of the reference period to be the same as those of the future comparison periods.

Response: Thank you for this comment. In the revised manuscript, we evaluate the snowmelt in China over projected climatic scenarios in different time frames i.e., near-future (2010-2039), mid-future (2040-2069), and far-future (2070-2099) in comparison with the reference period (1980-2009).

25. Why did you select the 2030s, 2050s, and 2090s? It should be introduced clearly.

Response: In the original manuscript, the 2030s, 2050s, and 2090s were selected to represent the near-future, mid-future, and far-future periods, respectively, to analyze the possible impact of future climate change on snowmelt. According to the comments from the Reviewer #1, in the revised manuscript, we use 2010-2039, 2040-2069, and 2070-2099 as near-future, mid-future, and far-future periods, respectively.

---

## Author Response (ED1)

Dear Editor,

Thank you for giving us the opportunity to revise our manuscript. We greatly appreciate you for your efforts and time on our manuscript, and we also thank you and two reviewers for providing helpful and constructive comments on our manuscript. Following those comments, we have revised the manuscript carefully. In the following, we provide detailed responses to your and reviewers' comments.

Thank you very much for your consideration, and best regards,

Yong Yang (on behalf of all co-authors)

**Responses to the Editor**

1. With regards to the reference period, I suggest you use the 30-year period 1981-2010. This has been a common practice recommended by WMO and also been used in the IPCC reports. Please see the link from WMO: https://public.wmo.int/en/media/news/updated-30-year-reference-period-reflects-changing-climate. You could consider using 2011-2040, 2041-2070, and 2071-2100 as near-future, mid-future, and far-future. It would be actually more useful if you could use the 30-year time period centred on the Specific Warming Levels from e.g. 2 degree to 4 degrees.

Response: We thank you for this comment, and we have learned a lot from it. In the revised manuscript, we evaluate the snowmelt in China over projected climatic scenarios in different time frames i.e., near-future (2011-2040), mid-future (2041-2070), and far-future (2071-2099) in comparison with the reference period (1981-2010). Some readers may be interested in snowmelt changes under different levels of future warming (i.e. 2 °C and 4 °C with respect to the pre-industrial period). However, we think this manuscript is already quite long (36 pages), and that adding the analyses on different levels of warming could make the manuscript too lengthy. Fig. S4 (Fig. 12) in the revised manuscript shows projected future changes in temperature (snowmelt) in China and its three main stable snow cover regions. Table 4 in the revised

manuscript shows not only the snowmelt in reference period and different future periods, but also the temperature in different periods. We think that readers can get some information from Fig. S4, Fig. 12 and Table 4 about changes in snowmelt under different warming levels.

2. I agree that the 5 GCMs selected by the ISI-MIP fast track project have been widely used in various research papers. You responded with "Many scientists have chosen these five models to predict future climate change around the world", but this does not really address the question from the reviewer: "Are the climate conditions in these models representative for the full range of possible conditions in terms of climate change (e.g. cold-dry, warm-wet)?" Please demonstrate or cite references that these 5 GCMs are representative of the possible future climate conditions in China out of the large ensemble of ca. 23 CMIP5 GCMs.

Response: According to the article introducing ISI-MIP published by Warszawski et al. (2014), these five CMIP5 models were selected because they could span the space of global mean temperature change and relative precipitation changes. We think these five CMIP5 models can represent the full range of possible conditions in terms of climate change. These five models are widely used in climate change research in China (e.g. Li et al., 2016; Su et al., 2016; Yuan et al., 2016; Chen et al., 2017; Yuan et al., 2017; Yin et al., 2018; Chang et al., 2020). We have added the sentences in the section "2.2.4 Climate projections and downscaling", and the added sentences are as follows: The Inter-Sectoral Impact Model Intercomparison Project (ISI-MIP) selected these five CMIP5 models to span the space of global mean temperature change and relative precipitation changes (Warszawski et al., 2014). Aforementioned climate projections have been bias-corrected downscaled to a grid with a resolution of 0.5 ° by ISI-MIP (Hempel et al., 2013), and widely utilized in climate change research in China (e.g. Li et al., 2016; Yuan et al., 2017; Chang et al., 2020).

Chang, J., Wei, Y., Yuan, X., Liao, H., and Yu, B.: The nonlinear impacts of global warming on regional economic production: an empirical analysis from China, Weather Clim. Soc., 12, 759-769, https://doi.org/10.1175/wcas-d-20-0029.1, 2020.

Chen, J., Gao, C., Zeng, X., Xiong, M., Wang, Y., Jing, C., Krysanova, V., Huang, J., Zhao, N., and Su, B.: Assessing changes of river discharge under global warming of 1.5 ℃ and 2 ℃ in the upper reaches of the

Yangtze River Basin: Approach by using multiple- GCMs and hydrological models, Quatern. Int., 453, 63-73, https://doi.org/10.1016/j.quaint.2017.01.017, 2017.

Hempel, S., Frieler, K., Warszawski, L., Schewe, J., and Piontek, F.: A trend-preserving bias correction-the ISI-MIP approach, Earth Syst. Dynam., 4, 219-236, https://doi.org/10.5194/esd-4-219-2013, 2013.

Li, J., Chen, Y. D., Zhang, L., Zhang, Q., and Chiew, F. H. S.: Future changes in floods and water availability across China: linkage with changing climate and uncertainties, J. Hydrometeorol., 17, 1295-1314, https://doi.org/10.1175/jhm-d-15-0074.1, 2016.

Su, B., Huang, J., Zeng, X., Gao, C., and Jiang, T.: Impacts of climate change on streamflow in the upper Yangtze River basin, Clim. Change, 141, 533-546, https://doi.org/10.1007/s10584-016-1852-5, 2016.

Warszawski, L., Frieler, K., Huber, V., Piontek, F., Serdeczny, O., and Schewe, J.: The Inter-Sectoral Impact Model Intercomparison Project (ISI-MIP): project framework, P. Natl. Acad. Sci. U. S. A., 111, 3228-3232, https://doi.org/10.1073/pnas.1312330110, 2014.

Yin, Y., Ma, D., and Wu, S.: Climate change risk to forests in China associated with warming, Sci. Rep., 8, 493, https://doi.org/10.1038/s41598-017-18798-6, 2018.

Yuan, X., Sun, X., Lall, U., Mi, Z., He, J., and Wei, Y.: China's socioeconomic risk from extreme events in a changing climate: a hierarchical Bayesian model, Clim. Change, 139, 169-181, https://doi.org/10.1007/s10584-016-1749-3, 2016.

Yuan, Z., Yan, D., Yang, Z., Yin, J., Zhang, C., and Yuan, Y.: Projection of surface water resources in the context of climate change in typical regions of China, Hydrolog. Sci. J., 62, 283-293, https://doi.org/10.1080/02626667.2016.1222531, 2017.

The manuscript presents a country-wide assessment on present and future changes of snowmelt in China. To this end, a simplified temperature-index model is used that simulates several snow properties including snowmelt, snowfall, and snow sublimation. The model is forced with high resolution temperature and precipitation datasets as well as datasets that are needed for the model parametrization, which include PDD, snow density and snow/rain threshold temperatures. The model outputs, snowmelt, snow depth, snow water equivalent, and snow cover extent, are validated on equivalent station-based or grid-based observation datasets. Finally, the model is forced with temperature and precipitation data for 5 CMIP5 models under 3 different RCPs are selected to project future changes of snowmelt in China. The manuscript is well structured and generally well written. However, there are some issues with the grammar in this manuscript since most of the manuscript has been written in a past tense, whereas I would expect that some sections of the manuscript could have been written in a present tense or future tense. The topic the manuscript covers is very interesting. The assessments presented in this paper are novel as well, but I have several major and minor comments that need to be addressed.

Response: We would like to sincerely thank you for taking the time to carefully read and review our manuscript. We have revised the manuscript according to your valuable comments and suggestions. The responses following a point-by-point to the comments are provided below.

**General Comments**

• My first general comment is on the grammar that has been used while writing the manuscript. Most of the manuscript is written in a past tense. That is fine in some sections, such as the Abstract, but insufficient in those sections where a present or future tense is warranted.

Response: We thank you for this comment. We have revised the tenses of some sentences.

• To compare future snowmelt with present snowmelt I think it would be more beneficial to use climate "reference" periods instead of comparing future decadal values with the mean values of an entire historical period since the entire period reflects different climatological

characteristics. For example, since the late 1970s/early 1980s the global warming accelerated (Hartmann et al., 2013; doi:10.1017/CBO9781107415324.008), which might have had a different impact on the snow climatology than before the late 1970s. Therefore, I would like to suggest using 1981-2010 as an historical "reference period" and to use 2010-2039, 2040-2069, and 2070-2099 as near-future, mid-future, and far-future "reference" periods, respectively. That might give a better representative view on future snowmelt changes.

Response: Thank you for this comment, and we have learned a lot from it. In the open discussion, we responded that we would choose 1980-2009 as the reference period, and choose 2010-2039, 2040-2069, and 2070-2099 as the near-future, mid-future and far-future, respectively. The editor suggested that we use 1981-2010 as the reference period. Therefore, in the revised manuscript, we evaluate the snowmelt in China over projected climatic scenarios in different time frames i.e., near-future (2011-2040), mid-future (2041-2070), and far-future (2071-2099) in comparison with the reference period (1981-2010).

• Section 2.2.2 L108: Why did the authors use observational air temperature data from 824 stations, whereas they already mentioned before that they would like to use a high-resolution air temperature dataset to force the model. What is the added value of the observational air temperature data? What is the bias between the observational air temperature data and the high-resolution dataset of Peng et al., 2019 and to what is the difference between PDDs retrieved from the high-resolution dataset and PDDs retrieved from the observational data? Please elaborate on this.

Response: The high-resolution temperature dataset includes the monthly minimum, maximum, and mean temperatures, but no monthly PDD (accumulated positive air temperature) used to calculate snowmelt in this manuscript. The monthly PDD were calculated according to the monthly mean temperature according to Eq. 6.

$$PDD = \begin{cases} 0 & T_a \leq T_1 \\ a \cdot T_a^2 + b \cdot T_a + c & T_1 < T_a < T_2 \\ T_a \cdot D_m & T_a \geq T_2 \end{cases} \qquad (6)$$

The value of the observational air temperature data from 824 stations is to determine PDD parameters in Eq. 6, i.e., $T_1$, $T_2$, $a$, $b$, and $c$. Table 1 in the original manuscript (Table 2 in the revised manuscript) shows the parameters required for the calculation of the monthly PDD.

The biases between the observational data and the high-resolution dataset have been discussed in detail in Peng et al. (2019). We have added some sentences in the section "2.2.1 High spatial resolution dataset of monthly air temperatures and precipitation" to introduce the performances of the high-resolution dataset, and the added sentences are as follows: The data were spatially downscaled from the 30′ Climatic Research Unit time series data with the 0.5′ climatology dataset of WorldClim using delta spatial downscaling. Peng et al. (2019) evaluated this dataset using observations collected from 1951 to 2016 by 496 meteorological stations across China, and the average values of the coefficient of determination ($R^2$), mean absolute error *(MAE)*, root mean square error (*RMSE*), and Nash-Sutcliffe efficiency (*NSE*) for the monthly mean temperature at all meteorological stations are 0.996, 0.820 ℃, 0.969 ℃ and 0.981, respectively, and for the monthly precipitation are 0.863, 13.269 mm, 21.941 mm and 0.808, respectively (Peng et al., 2019).

We have presented the statistical analysis between PDDs retrieved from the high-resolution dataset and PDDs retrieved from the observational data in four different climatic zones of China, and the values of $R^2$, *MAE*, *RMSE* and *NSE* were shown in Table 1 in the original manuscript. To better illustrate the accuracy of calculated PDD, we added the statistical analysis at 824 meteorological stations. The added sentences are as follows:

Statistical analysis of the calculated monthly PDD against observed PDD at 824 meteorological stations shows the excellent performance at almost all of the stations (Fig. S5).

[Figure]

Figure S5. Statistical criteria of the calculated monthly *PDD* (accumulated positive air temperature) against observed *PDD* at 824 meteorological stations (a, $R^2$, coefficient of determination; b, *MAE/Mean*, *MAE*, mean absolute error; *Mean*, monthly mean *PDD*; c, *RMSE/Mean*, *RMSE*, root mean square error; d, *NSE*, Nash-Sutcliffe efficiency).

• Section 2.2.4; L135-136: Why did the authors only select these five CMIP5 models? Are the climate conditions in these models representative for the full range of possible conditions in terms of climate change (e.g. cold-dry, warm-wet)? What are the consequences for choosing these models for the outcomes of this study? And finally, it would be beneficial to show a few figures on the projected temperature and precipitation changes in China to put the projected changes of snowmelt in a better context.

Response: We select the five CMIP5 models because these models were downscaled to 0.5 ° and bias-corrected by the Inter-Sectoral Impact Model Intercomparison Project (ISI-MIP). According to the article introducing ISI-MIP published by Warszawski et al. (2014), these five CMIP5 models were selected because they could span the space of global mean temperature change and relative precipitation changes. We think these five CMIP5 models can represent the full range of

possible conditions in terms of climate change. Detailed information on the bias correction and ISI-MIP can be seen in Hempel et al. (2013) and Warszawski et al. (2014). Many scientists have chosen these five models to predict future climate change around the world (e.g. Zarrin et al., 2021; Gusev et al., 2019; Nasonova et al., 2018; Marx et al., 2018; Vetter et al., 2016). Many studies have used these five models to predict future climate change in China (e.g. Li et al., 2016; Su et al., 2016; Yuan et al., 2016; Chen et al., 2017; Yuan et al., 2017; Yin et al., 2018; Chang et al., 2020). We have added the sentences in the section "2.2.4 Climate projections and downscaling", and the added sentences are as follows: The Inter-Sectoral Impact Model Intercomparison Project (ISI-MIP) selected these five CMIP5 models to span the space of global mean temperature change and relative precipitation changes (Warszawski et al., 2014). Aforementioned climate projections have been bias-corrected downscaled to a grid with a resolution of 0.5 °by ISI-MIP (Hempel et al., 2013), and widely utilized in climate change research in China (e.g. Li et al., 2016; Yuan et al., 2017; Chang et al., 2020).

Chang, J., Wei, Y., Yuan, X., Liao, H., and Yu, B.: The nonlinear impacts of global warming on regional economic production: an empirical analysis from China, Weather Clim. Soc., 12, 759-769, https://doi.org/10.1175/wcas-d-20-0029.1, 2020.

Chen, J., Gao, C., Zeng, X., Xiong, M., Wang, Y., Jing, C., Krysanova, V., Huang, J., Zhao, N., and Su, B.: Assessing changes of river discharge under global warming of 1.5 ℃ and 2 ℃ in the upper reaches of the Yangtze River Basin: Approach by using multiple- GCMs and hydrological models, Quatern. Int., 453, 63-73, https://doi.org/10.1016/j.quaint.2017.01.017, 2017.

Gusev, E. M., Nasonova, O. N., Kovalev, E. E., and Ayzel, G. V.: Impact of possible climate change on extreme annual runoff from river basins located in different regions of the globe, Water Resour., 46, S126-S136, https://doi.org/10.1134/s0097807819070108, 2019.

Hempel, S., Frieler, K., Warszawski, L., Schewe, J., and Piontek, F.: A trend-preserving bias correction-the ISI-MIP approach, Earth Syst. Dynam., 4, 219-236, https://doi.org/10.5194/esd-4-219-2013, 2013.

Li, J., Chen, Y. D., Zhang, L., Zhang, Q., and Chiew, F. H. S.: Future changes in floods and water availability across China: linkage with changing climate and uncertainties, J. Hydrometeorol., 17, 1295-1314, https://doi.org/10.1175/jhm-d-15-0074.1, 2016.

Marx, A., Kumar, R., Thober, S., Rakovec, O., Wanders, N., Zink, M., Wood, E. F., Pan, M., Sheffield, J., and Samaniego, L.: Climate change alters low flows in Europe under global warming of 1.5, 2, and 3 °C, Hydrol. Earth Syst. Sci., 22, 1017-1032, https://doi.org/10.5194/hess-22-1017-2018, 2018.

Nasonova, O. N., Gusev, Y. M., Kovalev, E. E., and Ayzel, G. V.: Climate change impact on streamflow in large-scale river basins: projections and their uncertainties sourced from GCMs and RCP scenarios, Proc. IAHS, 379, 139-144, https://doi.org/10.5194/piahs-379-139-2018, 2018.

Su, B., Huang, J., Zeng, X., Gao, C., and Jiang, T.: Impacts of climate change on streamflow in the upper Yangtze River basin, Clim. Change, 141, 533-546, https://doi.org/10.1007/s10584-016-1852-5, 2016.

Vetter, T., Reinhardt, J., Flörke, M., van Griensven, A., Hattermann, F., Huang, S., Koch, H., Pechlivanidis, I. G., Plötner, S., Seidou, O., Su, B., Vervoort, R. W., and Krysanova, V.: Evaluation of sources of uncertainty in projected hydrological changes under climate change in 12 large-scale river basins, Clim. Change, 141, 419-433, 10.1007/s10584-016-1794-y, 2016.

Warszawski, L., Frieler, K., Huber, V., Piontek, F., Serdeczny, O., and Schewe, J.: The Inter-Sectoral Impact Model Intercomparison Project (ISI-MIP): project framework, P. Natl. Acad. Sci. U. S. A., 111, 3228-3232, https://doi.org/10.1073/pnas.1312330110, 2014.

Yin, Y., Ma, D., and Wu, S.: Climate change risk to forests in China associated with warming, Sci. Rep., 8, 493, https://doi.org/10.1038/s41598-017-18798-6, 2018.

Yuan, X., Sun, X., Lall, U., Mi, Z., He, J., and Wei, Y.: China's socioeconomic risk from extreme events in a changing climate: a hierarchical Bayesian model, Clim. Change, 139, 169-181, https://doi.org/10.1007/s10584-016-1749-3, 2016.

Yuan, Z., Yan, D., Yang, Z., Yin, J., Zhang, C., and Yuan, Y.: Projection of surface water resources in the context of climate change in typical regions of China, Hydrolog. Sci. J., 62, 283-293, https://doi.org/10.1080/02626667.2016.1222531, 2017.

Zarrin, A., Dadashi-Roudbari, A., and Hassani, S.: Historical variability and future changes in seasonal extreme temperature over Iran, Theor. Appl. Climatol., 146, 1227-1248, https://doi.org/10.1007/s00704-021-03795-7, 2021.

We have added two figures on the projected precipitation (Fig. S3) and temperature (Fig. S4) changes in China and the added figures are as follows:

[Figure]

Figure S3. Projected future changes in precipitation in China and its three main stable snow cover regions. Historical: 1951-2017; RCP: representative concentration pathway.

[Figure]

Figure S4. Projected future changes in temperature in China and its three main stable snow cover regions. Historical: 1951-2017; RCP: representative concentration pathway.

- Section 3.1.3; L178: How are the PDDs measured? Or where do the data come from? Why not just simply say that when the temperature is above 0 snow starts to melt and the PDD starts to count? What are the temperature thresholds?

Response: PDD in this study is the monthly accumulated positive air temperature. The monthly measured PDD were calculated from the daily measured positive temperature, which were

collected from 824 meteorological stations during the 1951-2017 period and were provided by the China Meteorological Administration.

In this study, we use the temperature index model to calculate snowmelt, and we can't simply say that the snow starts to melt when the temperature is above 0 °C. Temperature index models are based on the assumptions that the temporal variability of incoming solar radiation is adequately represented by the variations of air temperature that the snowmelt during a time interval is proportional to positive air temperatures (Hock, 2003; Jost et al., 2012; Lopez et al., 2020). The input driver of the temperature index model is PDD, not mean temperature. For example, when the monthly mean temperature is 0 °C, there must be some days when the temperature is above 0 °C, so there must be snowmelt in that month. It is not reasonable to assume that melting starts when the temperature is above 0 °C.

The temperature thresholds are the parameters to determine the monthly PDD calculation method in Eq. 6. If the monthly mean temperature is lower than $T_1$, the temperature of each day is considered to be negative and the monthly PDD is 0. If the monthly mean temperature is higher than $T_2$, the temperature of each day is considered to be positive, and monthly PDD is the product of the monthly mean temperature and the number of days.

Hock, R.: Temperature index melt modelling in mountain areas, J. Hydrol., 282, 104-115, https://doi.org/10.1016/s0022-1694(03)00257-9, 2003.

Jost, G., Dan Moore, R., Smith, R., and Gluns, D. R.: Distributed temperature-index snowmelt modelling for forested catchments, J. Hydrol., 420-421, 87-101, https://doi.org/10.1016/j.jhydrol.2011.11.045, 2012.

Lopez, M. G., Vis, M. J. P., Jenicek, M., Griessinger, N., and Seibert, J.: Assessing the degree of detail of temperature-based snow routines for runoff modelling in mountainous areas in central Europe, Hydrol. Earth Syst. Sci., 24, 4441-4461, https://doi.org/10.5194/hess-24-4441-2020, 2020.

• Section 3.1.3; Table 1: The results that are presented here are strange. According to the authors the NSE and R2 are both 1.0, which means perfect. I doubt whether this is the case since the MAE and RMSE indicate there are errors. If the MAE and RMSE were 0 I could have imagined that the R2 or NSE are equal to 1 or close to 1, however that is not the case. Therefore,

I guess something must have go wrong in the calculations of R2 and NSE and urge the authors to address this point.

Response: The calculations of *NSE* and $R^2$ are correct (Fig. 2b). *NSE* and $R^2$ are both 1.00 in Table 1 because we retained 2 digits after the decimal point by rounding. We have revised those numbers by retaining 4 digits after the decimal point, and the revised Table 1 (Table 2 in the revised manuscript) is as follow:

Table 2. The parameters required for the calculation of the monthly accumulated positive air temperature (*PDD*) and the statistical analysis between the calculated and measured monthly *PDD* in four different climatic zones of China.

|  | $T_1$ | $T_2$ | *a* | *b* | *c* | $R^2$ | *MAE* | *RMSE* | *NSE* |
|---|---|---|---|---|---|---|---|---|---|
| MPZ | -7.99 | 5.79 | 0.79 | 15.37 | 56.38 | 0.9966 | 5.87 | 10.85 | 0.9958 |
| TCZ | -10.85 | 9.89 | 0.52 | 15.29 | 85.38 | 0.9975 | 7.96 | 15.32 | 0.9968 |
| TMZ | -10.41 | 9.51 | 0.52 | 15.45 | 81.43 | 0.9973 | 8.45 | 16.56 | 0.9964 |
| SMZ | -4.05 | 8.56 | 0.22 | 23.12 | 49.63 | 0.9993 | 2.67 | 7.63 | 0.9989 |

Note. MPZ, mountain plateau zone; TMZ, temperate monsoon zone; TCZ, temperate continental zone; SMZ, subtropical monsoon zone; $T_1$, $T_2$, a, b and c, parameters in the equation (6); $R^2$, coefficient of determination; *MAE*, mean absolute error (°C); *RMSE*, root mean square error (°C); *NSE*, Nash-Sutcliffe efficiency.

• Section 4.1.1; L260: NSE = 0.2 is considered to be an unsatisfactory evaluation score, so why did the authors decide to use this value as a threshold? Is there a reason why at several stations the outcomes are unsatisfactory? And maybe the authors can elaborate more on their choice to use the NSE as an evaluation criterium for snowfall. It is more usual to use NSE as an evaluation index for snowmelt runoff or discharge. For the evaluation criteria, please also check Moriasi et al., 2007: https://pubag.nal.usda.gov/download/9298/PDF

Response: Thank you for the recommended literature and we have learned a lot from it. Moriasi et al. (2007) suggested the hydrologic models in watershed simulations could be judged as "satisfactory" if *NSE* > 0.50, and Moriasi et al. (2015) recommended the same *NSE* value as a

threshold. The *NSE* thresholds recommended by Moriasi et al. were established by descriptive statistics such as mean, median, minimum, and maximum of model data collected from peer-reviewed articles. There are many factors that affect the *NSE* value, and even a poor fit may lead to a high *NSE* value. A value of 0.7 may or may not be indicative of a good fit (McCuen, 2006). Moriasi et al. (2015) also noted that these threshold recommendations may be adjusted based on the quality and quantity of available measured data, spatial and temporal scales, and project scope and magnitude. The recommended thresholds are for flow simulations for watershed-scale models, and they may not be suitable for snowfall evaluation. A more common understanding of *NSE* is that the closer the value to 1, the better the model performance, while *NSE* is less than 0, the model is unacceptable  (e.g. Legates and Mccabe, 1999; Mccuen et al., 2006; Moriasi et al., 2007; Moriasi et al., 2015; Halefom et al., 2017; Gupta and Kling, 2011; Jackson et al., 2019). The 0.2 mentioned in line 260 is not a threshold value for *NSE*, but only a reference to the value of the legend distribution given in Figure 3d. We have changed "296 (64.7%) had *NSE* > 0.2" to "353 (77.2%) have N*SE* > 0.0" in this line.

The main reasons for the unsatisfactory outcomes of several stations are as follows: 1) the snowfall was simulated on a monthly scale by the temperature thresholds; 2) the output data were at the grid scale, whereas the observed snowfall was at the site scale. In addition to *NSE*, other evaluation criteria such as $R^2$, *MAE* and *RMSE* were selected to verify the snowfall output by the model. Considering all evaluation criteria and comparing model performances from other studies on identifying precipitation types (see Section "5.1 Model Evaluation"), we believe that it is reasonable to calculate snowfall on a national spatial scale in this study.

Gupta, H. V. and Kling, H.: On typical range, sensitivity, and normalization of Mean Squared Error and Nash-Sutcliffe Efficiency type metrics, Water Resour. Res., 47, W10601, https://doi.org/10.1029/2011wr010962, 2011.

Halefom, A., Sisay, E., Khare, D., Singh, L., and Worku, T.: Hydrological modeling of urban catchment using semi-distributed model, Model. Earth Syst. Environ., 3, 683-692, https://doi.org/10.1007/s40808-017-0327-7, 2017.

Jackson, E. K., Roberts, W., Nelsen, B., Williams, G. P., Nelson, E. J., and Ames, D. P.: Introductory overview: Error metrics for hydrologic modelling – A review of common practices and an open source

library to facilitate use and adoption, Environ. Model. Softw., 119, 32-48, https://doi.org/10.1016/j.envsoft.2019.05.001, 2019.

Legates, D. R. and McCabe, G. J.: Evaluating the use of "goodness-of-fit" Measures in hydrologic and hydroclimatic model validation, Water Resour. Res., 35, 233-241, https://doi.org/10.1029/1998wr900018, 1999.

McCuen, R. H., Knight, Z., and Cutter, A. G.: Evaluation of the Nash-Sutcliffe Efficiency Index, J. Hydrol. Eng., 11, 597-602, https://doi.org/10.1061/(ASCE)1084-0699(2006)11:6(597), 2006.

Moriasi, D. N., Gitau, M. W., Pai, N., and Daggupati, P.: Hydrologic and water quality models: performance measures and evaluation criteria, T. ASABE, 58, 1763-1785, https://doi.org/10.13031/trans.58.10715, 2015.

Moriasi, D. N., Arnold, J. G., Van Liew, M. W., Bingner, R. L., Harmel, R. D., and Veith, T. L.: Model evaluation guidelines for systematic quantification of accuracy in watershed simulations, T. ASABE, 50, 885-900, https://doi.org/10.13031/2013.23153, 2007.

• Section 4.1.2; L269 / Section 4.1.3; L282-283 / Section 4.1.4; L294: The simulated output reported by the model of the authors is on monthly basis, based on monthly mean values. Then it would be more appropriate and representative to compare the simulated snow depth with the monthly means of the observed snow depth instead of selecting the last day of each month. That is not representative. Besides, most likely comparing the simulated values with the observed monthly means will improve the outcomes of the authors presented later in the manuscript.

Response: Thank you for your suggestion. The model was performed on a monthly scale, whereas the snow depth was observed at meteorological stations daily. We calculated the statistical criteria of the simulated snow depth against the monthly means of the observed snow depth, and found that there was no significant change compared with the statistical criteria of the simulated snow depth against the last day's observations. In our opinion, either using the monthly means of observed snow depths or using the last day's observations to validate the snow depths output by the model is not representative, but that using the last day's observations is more appropriate. The input snowfall data of the model is monthly snowfall, and the input temperature data is monthly PDD. The model calculates the snow depth from the amount of snow remaining after melting. When there is no snow on the last day, it means that all the snow has melted in that

month, the snow accumulation is 0, and the snow depth is 0. However, it is very likely that there are a few days in the month that are covered with snow, and the monthly mean snow depth will be greater than 0. It is not appropriate to use the monthly mean snow depth to validate the snow depths output by the model.

**Specific Comments**

• Abstract; L11: 1 km corresponds to 30 arcseconds or 0.5 arcminute. The latter has also been used in the dataset description for the high-resolution temperature and precipitation datasets by Peng et al., 2019.

Response: Thank you for pointing this out. It was our mistake. The units in L11 and elsewhere in the manuscript have been corrected in the revised version.

• Abstract; L16: I recommend the authors to use $m^3$/year instead of $m^3$ to make clear that the authors talk about the mean annual snowmelt instead of the total snowmelt volumes within a period.

Response: Thank you for your comment. We have revised "$m^3$" to "$m^3$ year$^{-1}$" in the revised manuscript.

• Abstract; L20: From the abstract I cannot derive what "third level" stands for. Also, from the manuscript it is not clear what authors mean with "third level". Therefore, I would like to suggest using "subbasin" instead of "third level".

Response: Thank you for your suggestion. In our opinion, using "third-level basins" may be more appropriate than "subbasin" in this manuscript. The hydrologists in China have graded China's basins into first-level, second-level and third-level basins. We have added a figure to the Supplement, which we hope will help reviewer and readers understand the third-level basins in the manuscript. The added figure is as follows:

[Figure]

[Figure]

Figure S1. The first-level (a) and second-level (b) basins in China.

• Introduction; L39: What do the authors mean with slower snowmelt rates? Please rephrase to increase the clarity of the sentence.

Response: According to study from Musselman et al. (2017), warming climates could lead to slower snowmelt. This is because the less snow accumulates on the ground, the slower the snow melting rate in winter, rather than lasting until summer, when the high temperature will cause the snow to melt quickly.

Musselman, K. N., Clark, M. P., Liu, C., Ikeda, K., and Rasmussen, R.: Slower snowmelt in a warmer world, Nat. Clim. Change, 7, 214-219, https://doi.org/10.1038/nclimate3225, 2017.

• Introduction; L63: Do the authors mean the crop sowing season, crop harvesting season, or the entire crop season with "crop planting season"? I would like to recommend using one of the above-mentioned terms.

Response: We will use "crop sowing season" in the revised manuscript.

• Introduction; L65: Recently a new paper has been published on the Asian water towers by Immerzeel et al., 2020 (https://doi.org/10.1038/s41586-019-1822-y).

Response: Thank you for recommending paper, we have added it to the references.

• Introduction; L76: Here and throughout the manuscript, please check and see my point at Abstract; L11.

Response: We have revised the spatial units throughout the manuscript.

- 2 Data Collection: Since a significant number of datasets is used in this study and the reader can get lost in all the numbers and details, I would like to suggest adding a table that includes the information of the datasets used, such as the variables (e.g., snow density), the source, the measurement periods, and the number of stations, etc.

Response: Thank you for your suggestion. We have added a table in the revised manuscript, and the added table is as follows:

Table 1. Data used in the snowmelt model.

| Variable | Period | Timescale | Spatial resolution | Number of meteorological stations | Source |
|---|---|---|---|---|---|
| Air temperatures* | 1951-2017 | Monthly | 0.5′ | - | Zenodo |
| Precipitation | 1951-2017 | Monthly | 0.5′ | - | Zenodo |
| Snow cover extent | 1979-2018 | Daily | 0.25 ° | - | NTPDC |
| Snow water equivalent | 1980-2020 | Daily | 25 km | - | NCDDC |
| Air temperature | 1951-2017 | Daily | - | 824 | CMA |
| Snowfall | 1961-1979 | Daily | - | 475 | CMA |
| Snow depth | 1951-2009 | Daily | - | 557 | CMA |
| Snow density | 1999-2008 | 5 days | - | 417 | CMA |
| Threshold temperatures | - | Monthly | - | 485 | Han et al. (2010) |

Note. *Minimum, maximum, and mean temperatures; CMA, China Meteorological Administration; NTPDC, National Tibetan Plateau Data Center; NCDDC, National Cryosphere Desert Data Center.

- Section 2.2.1; L100: NetCDF is a data format, data cannot be obtained from NetCDF but are supplied in a NetCDF format. As I understand via the links the data were obtained from Peng et al., 2019, so please refer to them. Also, the link to the precipitation datasets did not work.

Response: Thank you for pointing this out. We have revised the sentence as follows: Data with a high spatial resolution of 0.5′ (approximately 1 km), including the monthly minimum, maximum, and mean temperatures ($T_{min}$, $T_{max}$, and $T_a$, respectively) and precipitation, were obtained from Zenodo in the Network Common Data Form (https://doi.org/10.5281/zenodo.3114194 for

• Section 2.2; L102: CRU timeseries are supplied on a 0.5 degree grid, not at a 30 arcseconds grids. Please correct.

Response: Corrected.

• Section 2.2; L103: I miss a reference to the WorldClim dataset. Also, more information on this dataset would be beneficial. For instance, what is spatial resolution of this dataset?

Response: According to the paper by Peng et al. (2019), they used four high-resolution datasets at spatial resolutions of 10′, 5′, 2.5′, and 0.5′ from WorldClim v2.0 (http://worldclim.org, last access: 25 April 2019) (Fick and Hijmans, 2017). Data published by Peng used the WorldClim datasets at spatial resolutions of 0.5′. Detailed information on the WorldClim dataset can be seen in Peng et al. (2019).

Fick, S. E. and Hijmans, R. J.: WorldClim 2: new 1km spatial resolution climate surfaces for global land areas, Int. J. Climatol., 37, 4302-4315, https://doi.org/10.1002/joc.5086, 2017.

Peng, S., Ding, Y., Liu, W., and Li, Z.: 1 km monthly temperature and precipitation dataset for China from 1901 to 2017, Earth Syst. Sci. Data, 11, 1931-1946, https://doi.org/10.5194/essd-11-1931-2019, 2019.

• Section 2.2; L102-104: I would split up the sentence into two parts to increase the readability.

Response: We have revised the sentence as follows: The data were spatially downscaled from the 0.5 °Climatic Research Unit time series data with the 0.5′ climatology dataset of WorldClim using delta spatial downscaling. Peng et al. (2019) evaluated this dataset using observations collected from 1951 to 2016 by 496 meteorological stations across China.

• Section 2.2; L104: The observations are collected from 1951 to 2016, but is the latter year not supposed to be 2017?

Response: The data produced by Peng et al. (2019) covers the 1901 to 2017 period, and Peng et al. evaluated the dataset with meteorological stations observations from 1951 to 2016. The data period we used in the study is 1951-2017.

• Section 2.2.2; L114: Is this about the critical temperature defining whether precipitation falls as snow or rain? That there are different threshold makes sense, but it would be good to map this as well for the readers convenience.

Response: Yes, this is about the critical temperature. We have added a figure to the Supplement, and the added figure is as follows:

[Figure]

Figure S2. The threshold temperature (℃) for snowfall (a) and rainfall (b) at 485 meteorological stations in China.

• Section 2.2.2; L116: I think in this context there are more recent data available as well. For example, see Jennings et al. (2018; https://doi.org/10.1038/s41467-018-03629-7) on the spatial variation of the rain-snow temperature threshold across the Northern Hemisphere.

Response: Thank you for providing the literature. The study area of this manuscript is China, and we have collected the threshold temperature parameters based on the observations from the meteorological stations. In this manuscript, we prefer to use the data we collected. In future work, we may use the recommended data.

• Section 2.2.2; L117: Were the threshold temperatures spatially interpolated? If so, via IDW or another method?

Response: The IDW method. We have revised the sentence as follows: The threshold temperatures of each calculated cell were interpolated using the parameters from the meteorological stations via inverse distance weighting (IDW) method.

• Section 2.2.3; L120: Are there no other observations of snowfall data from a later period?

Response: Unfortunately, we did not collect snowfall data after 1980. According to the information we have, since 1980, all meteorological stations only provide data on total precipitation, and no longer provide data on precipitation types.

• Section 2.2.3; L125: Please validate the spatial resolution of the dataset.

Response: Checked.

• Section 2.2.3; L129: How is snow cover derived from snow depth?

Response: If the snow depth in the grid is greater than 0, it means that the grid is covered with snow. If the snow depth in the grid is 0, it means there is no snow in the grid.

• Section 3.1; L151: Does the PDD represent the mean monthly accumulated positive air temperature?

Response: PDD represent the monthly accumulated positive air temperature.

• Section 3.1.1; L159: The threshold temperatures are described, but not presented as a main result, which makes it difficult to imagine what the numbers are. Is it possible for the authors to present those numbers by means of a table or figure, either in the manuscript or the supplementary information?

Response: We have added a figure to the Supplement, which can be found in the previous response (Figure S2).

• Section 3.1.2; L160-168: The authors use a method to calculate DDF based on the density of snow and water. The density of snow is based on observations. The question is, however, how do

the authors calculate the future DDF. Do they authors assume the snow density to be constant over time?

Response: In this study, we calculated the DDF based on the snow density, and assumed that the monthly DDF is constant.

• Section 3.1.2; L164: The snow density is variable since it is observed, but the density of water is constant, so please note it here for the readers.

Response: We have revised the sentence as follows: where $\rho_s$ and $\rho_w$ are the density of snow and water (g cm$^{-3}$), respectively. The observed snow density is variable and the density of water is constant.

• Section 3.1.4; L200: Do the authors have a reference to this sublimation method? Or is this a method the authors developed their own? If so, is this method considered to be valid?

Response: Snow sublimation is difficult to quantify by measurement or modelling. Accurate calculation requires a large amount of high-quality meteorological data, which is difficult to achieve in large spatial scale. Therefore, we developed a simple method to estimate the monthly snow sublimation in this study. This method can be considered valid because: 1) the empirical parameters were set according to studies reporting the ratio of snow sublimation to snowfall in China and surrounding areas (Table S1 and Fig. S1 in the original manuscript); 2) the outputs by model were validated by snowfall, snow depth, snow cover extent and snow water equivalent.

• Section 3.2; L216-L218: I don't consider this as a good argument, since many hydrological models use the same methods, the authors also use. The models require temperature and precipitation data + several GIS data as an input, which are mostly available.

Response: The hydrological models mentioned in line 216 mainly refer to watershed hydrological models with runoff generation. Those models can separately simulate the runoff from rainfall and snowfall, and further calculate the snowmelt runoff ratio. The model in this manuscript did not calculate the runoff and used a simple method to estimate the snowmelt runoff ratio. To make it clearer, we have revised the sentence as follows: The watershed hydrological models with runoff processes are generally effective for determining the snowmelt

runoff ratio (Immerzeel et al., 2010; Jenicek and Ledvinka, 2020), however, they are difficult to implement over large regions due to data limitations and large computational requirements.

- Section 3.3; L233: Does n refer to the number of samples within a dataset?

Response: Yes.

- Section 4.1.1; L254-255: Please indicate this info in the data section.

Response: We have moved this sentence to the data section.

- Section 4.1.2; L268: How is the snow depth calculated from the snow accumulation and snow density?

Response: The snow accumulation calculated by the model is in mm. The snow depth can be calculated as follows:

$$snowdepth(mm) = snow\ accumulation \cdot water\ density/snow\ density$$

- Section 4.1.2; L270-271: Please indicate this info in the data section.

Response: We have moved this sentence to the data section.

- Section 4.1.4; L295-296: An alternative solution for dealing with scale differences is to conservatively remap the outcomes to the 25km grid to compare simulated values with observed values. Did the authors consider using a conservate remapping technique to compare the simulated values with the observed values?

Response: Thank you for your suggestion. We did not use a conservate remapping technique because the validation method we chose was valid.

- Section 4.1.4; L298: Here as well as in the other sections where validation outcomes are presented it would be beneficial to add the observed and simulated values as well.

Response: We have added the observed and simulated values in here and other sections. In this section, the added sentence is as follows: The mean monthly snow accumulation in China output

by the model from 1980 to 2017 is $2.55 \times 10^{10}$ m$^3$, while the mean snow water equivalent derived from passive microwave remote on the last day of each month is $1.47 \times 10^{10}$ m$^3$.

• Section 4.1.4; 299-306: I consider this argumentation as insufficient. Firstly, that the microwave remote sensing data have a spatial resolution of 25 km does not mean that relatively small glaciers are not recorded. At least, I think the authors should be able to substantiate why this is the case (e.g. by means of scientific literature) and whether it is a common problem for microwave remote sensing data. Secondly, since the authors decided to use the last day as a monthly observational representative the chance is very likely that the snow water equivalent on that particular day is 0, whereas the mean monthly snow water equivalent would have been different. This might explain as well why there is a discrepancy between the observed data and the simulated data. To my opinion, the authors need to elaborate more on this point.

Response: We believe that the resolution of microwave remote sensing is the main reason why relatively small glaciers have not been recorded. Fig. R1a shows the spatial distribution of the mean snow water equivalent in August 2017 in China, and Fig. R1b shows the spatial distribution of glaciers in China. The mean snow water equivalent of 0 in August means that the snow water equivalent of every day in August is 0. As can be clearly seen in the figure, most of the glacial regions in the Tibetan Plateau have a mean snow water equivalent of 0 in August, indicating that the data retrieved by the microwave remote sensing has not recorded the snow water equivalent of those glaciers. We have consulted Lingmei Jiang, the producer of snow water equivalent data, and Tao Che, the producer of snow depth data, both of whom are experts in microwave remote sensing, and they both agree with us. Regarding the second point of this comment, since the model determines the monthly snow accumulation based on the amount of snow remaining after melting, we prefer to use the snow water equivalent on the last day of the month to validate the snow accumulation.

[Figure]

Figure R1. Spatial distribution of the mean snow water equivalent (SWE) in August 2017 in China (a) and spatial distribution of glaciers in China (b).

• Section 4.2.1; L333: Here and throughout the manuscript, many readers have most likely no idea where most of the geographical areas are located. Therefore, I would like to recommend adding those locations to a map or to give an indication where these areas are located, such as West China or via some lat-lon coordinate.

Response: Thank you for your comment. In the Supplement, we have added a figure about the geographical distribution in the Tibetan Plateau (Fig. S8 in the revised manuscript), and we hope this figure is helpful to readers. The added figure is as follows:

[Figure]

Figure S8. The location of the Qaidam Basin, Qiangtang Plateau and Mountains on the Tibetan Plateau.

• Section 4.2.2; L358-359: I think this is a result of global warming that combined, causes a larger fraction of rainfall relative to snowfall during summertime and an earlier onset of snowmelt during spring.

• Section 4.2.2; L359: Can the authors support there finding by indicating the increase in temperature that is measured in the regions?

Response: In the context of global warming, the increase in temperature in the vast majority of the world has already been confirmed. According to temperature data provided by Peng et al. (2019), the temperature generally increased throughout China from 1951-2017, although not all grids were significant. Fig. 15 in the original manuscript can provide the information of temperature change in the areas with significant change of snowmelt.

• Section 4.3.1; L362: What are third-level basins? Are these basins of a third order? How are these defined? Is not better to mention them as subbasins?

Response: We have added a figure to the Supplement, which we hope will help reviewer and readers understand the third-level basins in the manuscript. The added figure can be found in the previous response (Fig. S1).

• Section 4.3.1; L366: Due to heavy rainfall and low snowmelt?

Response: Although snowfall accounts for a small proportion of precipitation, due to the heavy precipitation, the amount of snowmelt in some areas of South China is not low (Figure 7a in the original manuscript). We believe the low snowmelt runoff ratios in basins in South China are due to the heavy rainfall.

• Section 4.3.2; Line 386: I guess it should be Section 4.3.2 instead of 4.2.2

Response: Thank you for pointing this out. We have revised it.

• Section 4.3.2; L391: Is it Southeastern China or the southeastern part of the Tibetan Plateau?

Response: It's the southeastern part of the Tibetan Plateau. We have revised it.

• Section 4.3.2; L408: Is it significant or not significant. Considering the way how the sentence is phrased I would say non-significant. Please check and rephrase if necessary.

Response: It's not significant. We have revised the sentence as follows: ……the Mann-Kendall test shows that these monthly increasing trends in third-level basins are rarely significant.

• Section 5.1; L454-465: What are the big differences and why are the results of this paper different/better/worse than the results in the other studies?

Response: The results of this study are not significantly different from those of other studies, being slightly better in some places and slightly worse in others. The reasons for these differences are: 1) the input precipitation and temperature data are different, and 2) the threshold temperatures used to determine precipitation types are different. For example, Li et al. (2020) used the datasets on daily temperature and precipitation from the Asian Precipitation-Highly Resolved Observational Data Integration Towards Evaluation (APHRODITE) and the National Oceanic and Atmospheric Administration (NOAA) Climate Prediction Center (CPC), and they used temperature or wet-bulb temperature and relative humidity to define thresholds.

• Section 5.1; L468: The authors indicate their self that they should have used another method. What can the authors do to improve their results?

Response: As we mentioned earlier, the model was performed on a monthly scale, whereas the snow depth was observed at meteorological stations daily. It is difficult to find a suitable method to validate the simulated monthly-scale snow depth by the observed daily-scale snow depth. Based on the model principles and our understanding, using the last day's observations to validate the simulated monthly snow depth is the best option.

• Section 5.1; L466-479: What could be a potential reason for the under-performing snow depth? Could undercatch be a reason?

Response: We have stated three possible reasons in the manuscript. The reviewer mentioned that undercatch may be a reason. Does the reviewer mean gauge undercatch? According to our understanding, gauge undercatch is more about snowfall, not snow depth. We cannot answer with certainty whether undercatch is a reason, but it could be.

- Section 5.1; L484-487: I think point 2 and 3 are related to each other. Due to the complex terrain and harsh climate conditions, the number of stations is limited, and the distribution is sparse.

Response: Points 2 and 3 are to explain why the performance of the snow water equivalent in the Tibetan Plateau was poorer than those in other regions. Point 2 and Point 3 are indeed related, but they are two different reasons. The second reason is that the reliability of the model parameters might be worse in the Tibetan Plateau than in other regions. The third reason is that the snow water equivalent data used for verification have large uncertainties in the Tibetan Plateau.

- Section 5.2; L502: Is there a specific reason for the higher correlation during wintertime? Is the variability of winter precipitation larger than the variability of spring precipitation?

Response: This result comes from Tan et al. (2019). We think the reason is that snowfall mainly occurs in winter. Tan et al. (2019) only analyzed the change trend of precipitation, so we cannot answer the question about the variability of precipitation.

Tan, X., Wu, Z., Mu, X., Gao, P., Zhao, G., Sun, W., and Gu, C.: Spatiotemporal changes in snow cover over China during 1960-2013, Atmos. Res., 218, 183-194, https://doi.org/10.1016/j.atmosres.2018.11.018, 2019.

- Section 5.2; L523: Are the annual precipitation increases related to the changes in monsoon rainfalls or to changes in the westerly driven precipitation?

Response: According to the review paper by Kuang and Jiao (2016), there may be different mechanisms of precipitation changes in the Tibetan Plateau, and changes in monsoonal circulation may be an important reason.

Kuang, X. and Jiao, J. J.: Review on climate change on the Tibetan Plateau during the last half century, J. Geophys. Res. Atmos., 121, 3979-4007, https://doi.org/10.1002/2015jd024728, 2016.

- Section 5.2; L524: High elevations are more sensitive to warming --> Elevation-dependent warming. I am not sure what the authors mean with "tolerate", but I guess it should be opposite.

Response: We agree that high elevations are more sensitive to warming, and we think this is not in conflict with the fact that high elevations can tolerate more warming. For example, suppose there are two sites with a temperature of -5 ℃ at the high elevation site and -1 ℃ at the low elevation sites. The high elevation site is more sensitive to warming. The temperature at the high elevation site will increase by 4 ℃ and the temperature at the low elevation site will increase by 2 ℃. With a warming climate, the temperature at the high elevation site will be -1 ℃, the type of precipitation will still be snowfall and the snow will still not melt. However, at the low elevation site, the temperature will reach 1 ℃, the snowfall will turn to rainfall and the snow will melt.

• Section 5.3.2; L550-551: What are the authors trying to say with the high levels of precipitation in this region? As far I know most of the precipitation falls here during the East Asian summer monsoon, which is in summer, whereas the snowmelt season is in spring. That means the high levels of precipitation cannot be a good explanation for the snowmelt runoff ratios in this region.

Response: Thank you for pointing this out. What we wanted to say is that there is more precipitation in this region than in Northern Xinjiang. To avoid misunderstanding, we have revised this sentence as follows: The snowmelt runoff ratios in the basins in Northeast China are generally about 10%, and the importance of snowmelt as a water resource in this region may be less than in Northern Xinjiang.

• Section 5.3.3; L565: I think the authors cannot simply state that in some months all the runoff is contributed by snowmelt, since the authors have not considered the contribution of glacier meltwater to total runoff. For this reason, the authors need to elaborate more on this point, or I recommend them to rephrase the sentence.

Response: We have revised this sentence as follows: Snowmelt contributes runoff every month, and in some months, there is no rainfall to generate runoff.

**Technical Comments**

- Introduction; L38: "contribute" instead of "contributes"

Response: Changed.

- Introduction; L39: here and throughout the manuscript: "an earlier onset of snowmelt" instead of "earlier snowmelt times"

Response: Changed.

- Introduction; L44: "operations" instead of "operation"

Response: Changed.

- Introduction; L45: here and throughout the manuscript: "physically-based" instead of "physically based"

Response: Changed.

- Introduction; L46: I would use "simplified" instead of "simpler"

Response: Changed.

- Introduction; L52: "variations of air temperature and that the snowmelt" instead of "variations of air temperature that the snowmelt"

Response: Changed.

- Introduction; L61: "snowmelt" instead of "snow meltwater"

Response: Changed.

- Introduction; L64: "Snowmelt is also an important hydrological process on the Tibetan Plateau" instead of "Snow melting is aslo an important hydrological process in the Tibetan Plateau"

Response: Changed.

- Introduction; L65: "and is considered as the Asian water towers" instead of "and considered as the asian water towers".

Response: Changed.

- Introduction; L65: "Further, snowmelt is an important" instead of "Snowmelt is also an important"

Response: Changed.

- Introduction; L77: "considers" instead of "considered"

Response: Changed.

- Introduction; L78: "are" instead of "were"

Response: Changed.

- Introduction; L79: "China as well as in its three main" instead of "China and in its three main"

Response: Changed.

- Introduction; L80: "during 1951-2017" instead of "in the 1951-2017 period"

Response: Changed.

- Section 2.2; L102: "temperature" instead of "temperatures"

Response: Changed.

- Section 2.2.3; L119: I recommend merging the two sentences here into one. For example: "The observational snowfall (snow depth) data used to validate the model were collected from 475 (557) meteorological stations in China (Fig.1d (1a)) during the 1961-1979 (1951-2009) period and were provided by the China Meteorological Administration"

Response: Changed.

- Section 2.2.3; L123: "from" instead of "from by"

Response: Changed.

- Section 2.2.4; L139-142: This sentence is long and, particularly, the first part of the sentence needs to be rephrased to increase its readability and clarity.

Response: We have revised the sentence as follows: In this study, we use the delta downscaling method to determine the monthly future meteorological data (2006-2099) based on the high-spatial-resolution temperature and precipitation dataset and the simulations of the five CIMP5 models during the historical period (1951-2005).

- Section 3.1.2; L164: "mm °C$^{-1}$ day$^{-1}$" instead of "cm °C$^{-1}$ day$^{-1}$"

Response: In fact, the unit of DDF in Eq. 4 is indeed cm °C$^{-1}$ day$^{-1}$, not mm °C$^{-1}$ day$^{-1}$. In order to be consistent with the unit of DDF in Eq. 1, we have changed Eq. 4 and Eq.5 as follows:

$$DDF = 11(\rho_s/\rho_w) \tag{4}$$
$$DDF = 10.4(\rho_s/\rho_w) - 0.7 \tag{5}$$

- Section 3.1.4; L193: "methods" instead of "method"

Response: Changed.

- Section 3.3; L240: "where the $\beta$ sign reflects whether a trend is negative or positive" instead of "where $\beta$ sign reflects data trend reflection"

Response: Changed.

- Section 4.1.1; L259: I guess the authors made a mistake here. I guess it should be 57.5%.

Response: Thank you for pointing this out. We have revised it.

- Section 4.1.1; L260: "accounting for 60.0%" can be removed.

Response: Removed.

• Section 4.2.1; L316: repeated word, i.e., "the area with with"

Response: We have deleted a "with".

• Section 4.2.1; L322: "Plateau becomes the main region of snowmelt until May. In summer, there is no snowfall in most of China and snowmelt" instead of "Plateau became the main region of snowmelt until May. In summer, there was no snowfall in most of China and snowmelt"

Response: Changed.

• Section 4.2.2; L346: "In Southeast China" instead of "Southeast China"

Response: Changed.

• Section 4.2.2; L355: "but" instead of "while"

Response: Changed.

• Section 4.2.2; L356: "might imply" instead of "implied"

Response: Changed.

• Section 4.2.2; L357: "at the Tibetan" instead of "in the Tibetan"

Response: Changed.

• Section 4.4.1; L415: "are shown" instead of "were shown"

Response: Changed.

• Section 4.4.1; L425: "17.1% (24.7%, 42.8%), respectively, compared to the historical period." Instead of "17.1% (24.7%, 42.8%) compared to the historical period."

Response: In the revised manuscript, we use 1981-2010 as the historical "reference period", and 2011-2040, 2041-2070 and 2071-2099 as near-future, mid-future, and far-future periods, respectively. The revised sentence is as follows: …11.7% (24.8%, 36.5%), respectively, compared to the reference period.

- Section 4.4.2; L445-449: Very long sentence. I think it is better to split up the sentence in two parts.

Response: We have revised this sentence as follows: Under RCP8.5, the projected mean annual snowmelt runoff ratios in the far-future are lower than those in the reference period in all basins except the three basins near the Taklimakan Desert and one basin in central Inner Mongolia. Under RCP8.5, relative to the reference period, the snowmelt runoff ratios in the Tibetan Plateau and Tianshan Mountains are projected to decrease by more than 5% in most basins and by more than 10% in a few basins in the far-future.

- Section 5.2; L498: "grid cells" instead of "grids"

Response: Changed.

- Section 5.2; L513: "freezing point" instead of "freezing"

Response: Changed.

- Section 5.3.1; L546: "and therefore introduce agricultural risks Northern Xinjiang." Instead of "and changes in the snowmelt amount and timing may bring agricultural risks in the Northern Xinjiang."

Response: Changed.

- Section 5.3.3; L565: "contributes to runoff" instead of "contributes runoff"

Response: Changed.

As the key source of freshwater, snowmelt water resource in China has never been quantified on a national scale. This study used a simple temperature index model to calculate the snowmelt in China. The model is shown to perform acceptably well in China when the outputs were validated by snowfall, snow depth, snow cover extent and snow water equivalent. The results of this paper have important significance for understanding the distribution and variation of snowmelt in China. The simple model in this paper is interesting and the description of the model is comprehensive, and I think the method provides useful guidance for calculating snowmelt water resources outside China. In general, I think this work is valuable and of interest to the great community, and the manuscript is well written and worthy of publication in HESS. My comments are listed as follows:

Response: We would like to sincerely thank you for the valuable comments and suggestions to improve our manuscript. We have revised the manuscript according to your helpful comments. The responses following a point-by-point to the comments are provided below.

1. Line 11: the spatial resolution, 0.5 seconds? Shouldn't it be 0.5 minutes? Please check it throughout the manuscript.

Response: Thank you for pointing this out. It was our mistake. The units in Line 11 and elsewhere in the manuscript have been corrected in the revised version.

2. Line 16: change the unit "$m^3$" to "$m^3$ year$^{-1}$", and revise it throughout the manuscript.

Response: Thank you for your comment. We will use $m^3$ year$^{-1}$ instead of $m^3$ in the revised manuscript.

3. Line 19: Should it be "snowmelt water resource"? or "snowmelt time", "snowmelt rate"? "snowmelt" isn't clear, I think.

Response: Snowmelt is water produced when snow melts. In this sentence, we think the appropriate word is snowmelt.

4.  Line 38: "contributes" to "contribute".

Response: Thanks, we have revised it.

5.  Line 61: "snow meltwater" to "snowmelt".

Response: Changed.

6.  Line 64: change "aslo" to "also".

Response: Changed.

7.  Line 93: Figure 1a showed the mean snow depth from 557 meteorological stations in China. The mean snow depth is little significant, and it is better expressed by accumulated snow depth or the maximum snow depth. Figure 1c. Please use 3 as superscript in $cm^3$.

Response: Thank you for this comment. We have revised the the mean snow depth to the maximum snow depth in Figure 1a, and we have revised the unit in the Figure 1c. The revised figure is as follows:

[Figure]

Figure 1. The three main stable snow cover regions and the mean snow depth in China (1951-2009) (a); China's five climatic zones (MPZ, mountain plateau zone; TMZ, temperate monsoon zone; TCZ, temperate continental zone; SMZ, subtropical monsoon zone) and mean annual air temperature (1951-2017) (b); the snow cover classification and mean monthly snow density in China (1999-2008) (c); the third-level basins and mean annual snowfall in China (1961-1979) (d).

8.  Line 95: "mean snow density in China" is monthly or yearly? It should be introduced clearly.

Response: It is monthly. We have revised it.

9.  Line 101: The data link "https://doi.org/10.5281/zenodo.3114194forprecipitation" can not be connected.

Response: The data link is https://doi.org/10.5281/zenodo.3114194.

10. Line 115: The threshold temperature in China in this study should be shown in this manuscript by figure or table, or partly shown, I suggest.

Response: Thank you for your suggestion. We have added a figure to the Supplement, and the added figure is as follows:

[Figure]

Figure S2. The threshold temperature (℃) for snowfall (a) and rainfall (b) at 485 meteorological stations in China.

11. Line 117: What is the interpolated method?

Response: The interpolated method is IDW method. We have revised the sentence as follows: The threshold temperatures of each calculated cell were interpolated using the parameters from the meteorological stations via inverse distance weighting (IDW) method.

12. Line 124: Please delete "by".

Response: Deleted.

13. Line 129: The original dataset is snow depth, and the authors used this dataset to verify the snow cover extent. Please explain that.

Response: It is difficult to observe the snow cover extent on the ground, and space remote sensing constitutes an efficient observation technique. The snow cover extent can be generated from the snow depth dataset, and we use this dataset to validate snow cover extent output by the model.

14. Line 135: downscaling was finished by yourself or others? It should be elaborated in detail.

Response: The downscaling was done by ourselves. The L139-142 in the original manuscript introduces the downscaling method, and we have revised the sentence as follows: In this study, we use the delta downscaling method to determine the monthly future meteorological data (2006-2099) based on the high-spatial-resolution temperature and precipitation dataset and the simulations of the five CIMP5 models during the historical period (1951-2005).

15. Line 151: The unit of DDF is mm °C$^{-1}$ day$^{-1}$, but the unit in equation (4) (Line 164) is cm °C$^{-1}$ day$^{-1}$. Please check.

Response: Both units are correct. To make the two units consistent, we changed Eq. 4 and Eq.5 as follows:

$$DDF = 11(\rho_s/\rho_w) \tag{4}$$
$$DDF = 10.4(\rho_s/\rho_w) - 0.7 \tag{5}$$

16. Line 182: *NSE* equals one is not understandable, *RMSE* is not small for the temperature value. And the table does not reflect the monthly difference, but it has said monthly parameter in the title.

Response: *NSE* equals one in Table 1 because we retained 2 digits after the decimal point by rounding. *RMSE* in the Table 1 are not the statistical analysis between the calculated and measured monthly mean temperature, but the statistical analysis between the calculated and measured monthly accumulated positive air temperature (PDD). The values of *RMSE* may be not small for the mean temperature value, but they are very small for the monthly PDD (Fig. 2b). The parameters in Table 1 have no seasonal differences and they are used to calculate monthly PDD.

We have revised the numbers by retaining 4 digits after the decimal point, and the revised Table 1 (Table 2 in the revised manuscript) is as follow:

Table 2. The parameters required for the calculation of the monthly accumulated positive air temperature (*PDD*) and the statistical analysis between the calculated and measured monthly *PDD* in four different climatic zones of China.

| | $T_1$ | $T_2$ | *a* | *b* | *c* | $R^2$ | *MAE* | *RMSE* | *NSE* |
|---|---|---|---|---|---|---|---|---|---|
| MPZ | -7.99 | 5.79 | 0.79 | 15.37 | 56.38 | 0.9966 | 5.87 | 10.85 | 0.9958 |
| TCZ | -10.85 | 9.89 | 0.52 | 15.29 | 85.38 | 0.9975 | 7.96 | 15.32 | 0.9968 |
| TMZ | -10.41 | 9.51 | 0.52 | 15.45 | 81.43 | 0.9973 | 8.45 | 16.56 | 0.9964 |
| SMZ | -4.05 | 8.56 | 0.22 | 23.12 | 49.63 | 0.9993 | 2.67 | 7.63 | 0.9989 |

Note. MPZ, mountain plateau zone; TMZ, temperate monsoon zone; TCZ, temperate continental zone; SMZ, subtropical monsoon zone; $T_1$, $T_2$, a, b and c, parameters in the equation (6); $R^2$, coefficient of determination; *MAE*, mean absolute error (°C); *RMSE*, root mean square error (°C); *NSE*, Nash-Sutcliffe efficiency.

17. Line 259: 263 (5.7%). 5.7%? please check.

Response: Thank you for pointing this out. It's 57.5%. We have revised it.

18. Lines 268-276: The number of meteorological stations used for snow depth verification was 264, far fewer than the 557 stations in Figure 1a. Why choose so few meteorological stations for verification?

Response: We have collected observational snow depth data from 557 meteorological stations, however, at some meteorological stations, short snow cover duration and shallow snow result in very little data on snow depth. When performing snow depth validation, meteorological stations with little data were not selected, and data from 264 stations were finally selected for validation. According to Fig. 1a and Fig. 4, most of the meteorological stations that have not participated in the verification are in central and southern China where the climate is relatively warm.

Meteorological stations located in the three main stable snow cover regions are rarely excluded from verification.

19. Line 316: Delete the extra word "with".
Response: Deleted.

20. Line 327: It is better to cite Fig.7b before Fig.8 in the manuscript.
Response: In line 315 in the original manuscript, we have cited Fig. 7a for the first time. In line 319, we have cited Fig. 8 for the first time. Fig. 7 is cited before Fig. 8, and we do not think it is necessary to cite every sub-figure in Fig. 7 before Fig. 8.

21. Line 387: "4.2.2" to "4.3.2".
Response: Changed.

22. Line 415: "were shown" to "are shown".
Response: Changed.

23. Line 340: How is the density used when using the model from point to surface in China? Do you use the average value for the five typical regions from different sites or other methods? It should be explained in detail.
Response: Sorry, we can't understand this comment. The line 340 in the manuscript is the caption of Fig.8, and we do not see any relationship between the caption and this comment. And there is no "five typical regions" in our manuscript. If the reviewer is asking how we use snow density, we can answer that. We use the snow density to calculate the DDF values at the meteorological stations separately and then interpolate those to each calculated cell via IDW.

24. Lines 409-451: 4.4 Future changes of snowmelt under different climate scenarios. The historical period is from 1951 to 2017, while the future periods are different decades, namely the 2030s, 2050s, and 2090s. The comparison periods are inconsistent. I suggest changing

"historical period" to "reference period", and setting the period of the reference period to be the same as those of the future comparison periods.

Response: Thank you for this comment. In the revised manuscript, we evaluate the snowmelt in China over projected climatic scenarios in different time frames i.e., near-future (2011-2040), mid-future (2041-2070), and far-future (2071-2099) in comparison with the reference period (1981-2010).

25. Why did you select the 2030s, 2050s, and 2090s? It should be introduced clearly.

Response: In the original manuscript, the 2030s, 2050s, and 2090s were selected to represent the near-future, mid-future, and far-future periods, respectively, to analyse the possible impact of future climate change on snowmelt. According to the comments from the Reviewer #1 and the Editor, in the revised manuscript, we use 2011-2040, 2041-2070, and 2071-2099 as near-future, mid-future, and far-future periods, respectively.

---

## Author Response (AR2)

Dear Editor,

Thank you for giving us the opportunity to revise our manuscript. We are pleased to submit the revised version of our manuscript. In the following, we provide the response to your comment.

**Comment:**

The point made by reviewer#1 is valid. Many of the HMs use similar degree-day method to calculate snowmelt as what you used in your study. There is no need to argue HMs are not capable of producing snowmelt due to data limitation and computational demand, because you are focusing on snow only while HMs aim to simulation river discharge and snowmelt is only an intermediate output. Please consider to rephrase or delete the sentence.

Response:

We thank you for this comment, and we have deleted the sentence.

Thank you very much for your consideration, and best regards,

Yong Yang (on behalf of all co-authors)